

# ChinAllomeTree 1.0: China's normalized tree biomass equation dataset

Yunjian Luo [1, 2, 4], Xiaoke Wang [2, 3], Zhiyun Ouyang [2], Fei Lu [2], Liguo Feng [1], Jun Tao [1]

[1] College of Horticulture and Plant Protection, Yangzhou University, Yangzhou 225009, China

5   [2] State Key Laboratory of Urban and Regional Ecology, Research Center for Eco-Environmental Sciences, Chinese Academy of Sciences, Beijing 100085, China

[3] University of Chinese Academy of Sciences, Beijing 100049, China

[4] Joint International Research Laboratory of Agriculture and Agri-Product Safety, Ministry of Education, Yangzhou 225009, China

10   *Correspondence to*: Xiaoke Wang (wangxk@rcees.ac.cn) and Yunjian Luo (yjluo@yzu.edu.cn)

**Abstract.** The tree biomass equation, which is also called the tree allometric equation, is the most commonly used method to estimate tree and forest biomass at various spatial-temporal scales because of its high accuracy, efficiency and conciseness. For decades, many tree biomass equations have been reported in diverse types of literature (e.g., journals, books and reports). These scattered equations are being compiled, and tree biomass equation datasets are currently available for many geographical regions (e.g., Europe, North America and Sub-Saharan Africa) and countries (e.g., Australia, Indonesia and Mexico) except for in an important region of the world, Eastern Asia, specifically China. Therefore, in this study, we carried out an extensive survey and critical review of the literature (from 1978-2013) on biomass equations conducted in China and developed China's normalized tree biomass equation dataset (ChinAllomeTree version 1.0). This dataset consists of 5,924 biomass component equations for nearly 200 species and their associated background information (e.g., geographical location, climate and stand description), showing sound geographical, climatic and forest vegetation coverages across China. The dataset is freely available at https://doi.pangaea.de/10.1594/PANGAEA.895244 for noncommercial scientific applications, which fills an important regional gap in global biomass datasets and provides key parameters for biomass estimation in forest inventory and carbon accounting in China.



# 1 Introduction

Forests are the dominant terrestrial ecosystem in the world, occupying 30.6% of the global land area ($1.3 \times 10^{10}$ hectare) (FAO, 2016). An important challenge that has been faced by ecologists and foresters for decades is how to enhance the accuracy, consistency and efficiency of tree and forest biomass estimations at various spatial-temporal scales, which is

essential to understanding forest carbon cycling and implementing forest carbon-offset activities (Bustamante et al., 2014; Chave et al., 2014). However, the current estimates still have considerable uncertainties (Pan et al., 2013) due in large part to the limited geographical coverage of the estimation methods and their related parameters. The biomass equation is one of the most commonly used methods because of its high accuracy, efficiency and conciseness (Chave et al., 2014; Paul et al., 2016), but this method still faces a shortage of localized parameters.

The tree biomass equation, which is also called the tree allometric equation, refers to the quantitative relationship between tree biomass (or its components such as stem, branch, leaf and root) and one or more dendrometric variables (e.g., tree diameter and height). From the International Biological Programme (IBP) on, many tree biomass equations have been developed at specific sites for many tree species and forest types. Currently, many national, regional and global efforts are underway to compile and evaluate these scattered biomass equations (Henry et al., 2015). This work is of great

significance to goals such as (1) facilitating the identification of the gaps in the coverage of the equations, (2) testing and comparing existing equations with new ones, (3) developing generalized biomass equations (Forrester et al., 2017; Jenkins et al., 2003), (4) validating and calibrating process-based models and remotely sensed biomass estimates (Rojas-Garcú et al., 2015) and integrating these models with remotely sensed data (e.g., tree height and crown breadth) (Jucker et al., 2016), and (5) elucidating and confirming the generality of plant allometric scaling laws (Návar, 2009; Pilli et al., 2006).

To date, tree biomass equation datasets have been created for geographical regions such as Europe (Zianis et al., 2005; Forrester et al., 2017), Latin America (Návar, 2009), North America (Ter-Mikaelian & Korzukhin, 1997; Jenkins et al., 2004), Southeast Asia (Yuen et al., 2016) and Sub-Saharan Africa (Henry et al., 2011) and for countries such as Australia (Eamus et al., 2000; Keith et al., 2000), Indonesia (Anitha et al., 2015) and Mexico (Rojas-Garcú et al., 2015). However,



one important region of the world is highlighted as an area where a biomass equation dataset is clearly lacking, Eastern

Asia, especially China (Henry et al., 2015).

China covers most of the world's terrestrial biomes and environmental gradients and has a series of forest types

ranging from tropical rainforest to boreal forest. It is said that 'if you study China, you'll know the world' (Fang et al.,

2012). In the late 1970s, studies to measure tree biomass and develop biomass equations were initiated in China (Pan et al.,

1978, Zhang and Feng, 1979). Subsequently, many studies have expanded to nearly all climatic zones and forest types in

China (Luo et al., 2014). Some biomass equation datasets have been built for specific regions (e.g., northeastern China,

Chen and Zhu, 1989; Xishuangbanna Forest Region and Hainan Island, Yuen et al., 2016), forest types (e.g.,

*Cunninghamia lanceolata* forest, Zhang et al., 2013; *Larix* forest, Wang et al., 2005; *Populus* forest, Liang et al., 2006),

and specific time periods (e.g., from 1978 to 1996, Feng et al., 1999). More importantly, these datasets employed different

screening criteria for data inclusion. After our *Ecology* data paper on forest biomass and its allocation (Luo et al., 2014;

related dataset is freely accessible at http://www.esapubs.org/archive/ecol/E095/177/), therefore, it is appropriate to

develop a normalized tree biomass equation dataset for China from a broad literature survey.

## 2 Materials and methods

### 2.1 Literature retrieval

Concerning tree biomass equations in China (excluding Taiwan Province in our study), we made a great effort to

collect the available literature (journals, books and reports) between 1978 and 2013. Using a series of keywords (biomass,

allometry, allometric, relationship, equation, model, and function) with logical operators, studies were retrieved from

national libraries (National Digital Library of China, and China Forestry Digital Library), online literature databases (Web

of Science, China Knowledge Resource Integrated Database, and China Science and Technology Journal Database), and

reference lists from our *Ecology* data paper (Luo et al., 2014) and existing equation compilations (Feng et al., 1999; Wang

et al., 2005; Liang et al., 2006; Xiang et al., 2011; Zhang et al., 2013). During the literature survey, no a priori criteria (e.g.,

tree species, tree age, site condition, measurement method, and statistical technique) were applied.

**2.2 Data collection**

A critical review of the collected literature was conducted to obtain reliable biomass equations using the following criteria:

**(1)    Scope:** Equations for inclusion were restricted to those for both forest-grown trees and open-grown trees. However, equations for mangrove trees and recently disturbed trees (e.g., coppicing, pruning, fire, and insect pests) were not included.

**(2)    Measurement method:** A robust measurement method should cover the appropriate survey period (during the growing season, especially for deciduous trees), plot setting and tree biomass (the oven-dried mass) measurements (cf.
Feng et al., 1999). Generally, plot areas were not less than 100 $m^2$ for boreal and temperate forests, 400 $m^2$ for subtropical forests, and 1000 $m^2$ for tropical forests. To develop biomass equations, at least three sample trees should be selected to determine the tree biomass and its components (e.g., stem, branch, foliage, and root) by destructive harvesting and weighing. The division of tree biomass components can be summarized as shown in Fig. 1, although the number of biomass components varied with the different purposes of the investigations.

● **Aboveground biomass:** The biomass of at least three aboveground tree components (stem, branch, leaf, or their whole subcomponents) should be determined. If any of the three components or their subcomponent biomass was not measured, the aboveground biomass and relevant biomass (e.g., tree crown and total aboveground biomass) were considered to be inadequate.

● **Belowground biomass:** The quality of total belowground biomass was evaluated from three aspects. (1) The
total belowground biomass should be the total biomass of the entire root system (i.e., root crown and different root diameters), which was determined by using either the full excavation method for the entire root system or a hybrid of the full excavation method for the root system (excluding fine roots) and the soil pit method (or soil coring method) for fine roots. (2) The excavation area was larger than or equal to the average tree area covered,



and the excavation depth reached the maximum depth where roots were nearly absent, which was more than at least 50 cm (Mokany et al., 2006). (3) Fine roots are usually classified as <2~5 mm in root diameter (Finér et al., 2011). Fine roots play significant roles in the water and nutrient uptake of trees but contribute little to the total belowground biomass (Mokany et al., 2006). However, if the minimum measured root diameter is >5 mm, the total belowground biomass may be significantly underestimated.

**(3)** **Equation building:** Biomass equations should be developed using robust regression methods (e.g., ordinary least squares, maximum likelihood and Bayesian techniques), explicit equation forms (e.g., power, exponential and linear equations) and valid equation evaluations.

- **Predictor variables:** The predictor variables for the biomass equations were limited to the tree diameter at a certain height (e.g., basal diameter and diameter at breast height), tree height, and various combinations of them. These variables were used mostly because other variables (e.g., stand density, site index, and soil type) were highly related to local conditions and thus reduced the robustness and generality of the biomass equations.

- **Equation forms:** If two or more equation forms with the same predictor variable(s) were used to build the equations, the regression results of only one equation form were selected. More specifically, if the differences (<0.05) in the coefficients of determination ($R^2$) or correlation coefficient ($R$) were small among all equation forms, the priority order of equation forms for inclusion was power, exponential, and others (e.g., polynomial and hyperbolic); if not, the equation form with the highest $R^2$ (or $R$) was selected. Moreover, for studies that had original data rather than equations, equations were fitted using these original data and two typical allometric models: $W = a \, D_x^{\,b}$ and $W = a \, (D_x^{\,2}H)^b$, where W is tree biomass (kg) or its components, $D_x$ and H are tree diameter at $x$ height (cm) and tree height (m), and $a$ and $b$ are equation coefficients.

- **Equation evaluation:** The goodness-of-fit of regression equations should be evaluated, where the statistical measures $R^2$ and $R$ are commonly used in studies in China. Other goodness-of-fit measures except $R^2$ and $R$ were not included in our dataset, largely because diverse forms of the error estimates were employed across studies. In addition, several correction factors are proposed to correct the systematic bias in the biomass estimates by using

log-linearized equations (Clifford et al., 2013); thus, they were collected for log-linearized equations if available.

**(4)** **Quality checking:** Robust measurement methods and reliable equation building methods should be adopted in the original studies. Biomass equations for inclusion were checked and were even corrected using original biomass data if available. With increasing tree sizes (diameter and height), the biomass equations did not show unreasonable ranges of tree biomass or biomass allocations. The tree biomass and its allocations were regarded as acceptable if they fell within the biomass and allocation ranges of the averaged trees by forest type and age class (Luo et al., 2013). When the biomass or biomass allocation of the trees that were generated by an equation was outside the abovementioned empirical ranges, the equation was considered questionable and then rechecked to evaluate its inclusion in our dataset.

The biomass equations that met the above criteria were compiled to develop China's tree biomass equations dataset (or ChinAllomeTree for short), which consists of a biomass equation sheet and a general information sheet. The former sheet includes a biomass component, predictor variable, equation form, equation coefficients, goodness-of-fit statistics (e.g., correlation coefficient and coefficient of determination) and applicable ranges (i.e., the value ranges of predictor variables). The latter sheet stores the supporting information for the equations, including the geographical location (e.g., latitude, longitude and altitude), climate (mean annual temperature (MAT) and mean annual precipitation (MAP)), stand description (e.g., forest type, dominant species and stand origin), and target tree species (group) that are used to develop the biomass equations (e.g., stand age, stand density, and equations included). The detailed variables and their descriptions in the dataset are summarized in Table 1.

## 2.3 Estimation of missing data

Not all original studies reported the geographical location, climate data (MAT and MAP), or applicable ranges of biomass equations. These missing data were estimated as follows:

**(1)** **Geographical location:** Google Earth was used to estimate the geographical centers of the study sites in the original studies without geographical location descriptions in the form of latitude, longitude and altitude.

**(2)** **Climate data:** MAT and MAP data were extracted using geographic coordinates from WorldClim version 1.4

(http://worldclim.org/current), which is a 30 arc-second (~1 km at the Equator) resolution global climate database

(Hijmans et al., 2005).

**(3)** **Applicable ranges of biomass equations:** Empirical biomass equations were built based on sample trees with

limited ranges of tree size (diameter and height). When these equations are applied beyond the ranges for which they were

5  developed, the reliability of the biomass estimates is often questionable (Henry et al., 2011). The size ranges of the sample

trees were not always given in the original studies, and it was not possible to access the raw data used for equation building.

According to the amount and reliability of the information in the original studies, five methods (Table 2) were used to

obtain the applicable ranges for the biomass equations. However, some applicable ranges were finely calibrated under the

rule 'tree biomass increases with increasing tree size'.

## 3 Results

From 518 references during the period 1978~2013 (Appendix B), 759 studies and 5,924 biomass equations from these

studies were compiled in China's tree biomass equation dataset (ChinAllomeTree version 1.0). Temporal changes in the

number of studies showed (Fig. 2) a continuously increasing trend from 1978 to 1990, while a decreasing trend was found

during the period 1991~2002. Since 2002, there has been a generally increasing trend. Studies from 1978 to 1990, 1991 to

15  2002 and 2003 to 2013 contributed 27.4%, 34.0% and 38.6% of the total studies, respectively. These studies were carried

out in 359 sites, showing sound geographical coverage (18.6~52.4 °in latitude, 76.8~130.7 °in longitude and 2~4588 m in

altitude) across China (Fig. 3a) and broad climatic gradients (-5.6~24.6 ℃ in MAT and 39~2500 mm in MAP),

representing all biomes from desert to tropical rainforest (Fig. 3b).

These compiled studies and equations varied greatly with forest type, stand origin and tree species (Fig. 4; Table 3).

20  The studied forests were categorized into five types: deciduous coniferous forest, evergreen coniferous forest, deciduous

broadleaved forest, evergreen broadleaved forest, and coniferous and broadleaved mixed forest. Among the five forest

types, evergreen coniferous forest had the most studies and equations (45.7% and 38.7% of the total studies and equations),



followed by deciduous broadleaved forest (22.9% and 24.1%), evergreen broadleaved forest (17.5% and 21.0%),

deciduous coniferous forest (10.4% and 9.5%), and coniferous-broadleaved mixed forest (3.4% and 6.7%) (Fig. 4a). For

stand origins, 77.2% and 68.8% of the total studies and equations focused on planted forests (Fig. 4b). Apart from mixed

species, there were 5,488 equations specific to 197 species (Table 3). However, only 63 species were in more than two

studies, occupying 80.5% of the total species-specific equations. The five most commonly studied species were

*Cunninghamia lanceolata* (*n*=130), *Pinus massoniana* (*n*=60), *Pinus tabuliformis* (*n*=46), *Pinus koraiensis* (*n*=32) and

*Larix principis-rupprechtii* (*n*=30), which had 706, 365, 395, 218 and 235 equations, respectively.

Compared with the aboveground sector, the belowground sector was not always measured. Many studies (*n*=177) did

not (properly) address the belowground sector, accounting for 23.3% of the total studies. Equations for stem biomass and

its subcomponents contributed 27.1% to the total 5,924 equations, while branch biomass and its subcomponents

contributed 20.1%, leaf biomass and its subcomponents contributed 19.3%, aboveground biomass contributed 6.1%,

belowground biomass and its subcomponents contributed 18.3%, and total tree biomass 7.8% (Table 3). However, only

1.2% of the equations were for other biomass components, such as flower and fruit biomass and tree crown biomass.

Of the 5,924 equations, 43.5% were based on a single predictor (diameter or height), and 56.5% were based on two

predictors (diameter and height) or their combinations (Fig. 5a). The diameter at breast height was the most frequently

used predictor in the biomass equations (96.8%), whereas tree diameter at other heights rather than breast height was used

in 185 equations (3.1%). Moreover, only 9 equations (0.2%) employed tree height as a single predictor. In total, 29

equation forms were applied to develop the quantitative relationships of tree biomass with tree diameter and/or height,

which were categorized into five types: power equation, log-linear equation, linear/polynomial equation, exponential

equation and hyperbolic equation (Table A2). The power equation was the most frequently used type (3,948 equations,

accounting for 66.6% of equations), followed by the log-linear equation (1,438, 24.3%), linear/polynomial equation (432,

7.3%), exponential equation (85, 1.4%) and hyperbolic equation (21, 0.4%) (Fig. 5b).

A considerable proportion (20.1%) of the total 5,924 equations did not specify the sample size (i.e., the number of

trees harvested to develop the equations) (Fig. 6a). The sample size varied from 3 and 420 trees, where the most common

sample sizes were between 6 and 25 trees, accounting for 74.5% of the 4,734 equations with specific sample sizes. For the

applicable ranges of equations, 2,790 out of the 5,924 equations had clear applicable ranges in the original studies. As is

often the case, there was a great bias towards the smaller diameter classes (Fig. 6b) and height classes (Fig. 6c). From the

5,856 equations with available diameter ranges, the maximums and ranges (max-min) of tree diameter varied between 1.6

cm and 150.0 cm and between 1.0 cm and 130.0 cm, where 74.4% and 86.2% of the equations had maximums and range

less than 30 cm, respectively. From the 3,336 equations with available height ranges, the maximums and ranges of the

height ranged from 1.2 m to 66.8 m and 0.6 m to 51.5 m, and most of them (73.7% and 94.1%) were less than 20 m.

**4 Data availability**

This version of China's tree biomass equation dataset (ChinAllomeTree version 1.0) was developed from studies that

were published from 1978~2013. Data collection is ongoing, and the dataset will be updated as additional data are

collected and verified. The dataset is freely available at https://doi.pangaea.de/10.1594/PANGAEA.895244 for

noncommercial scientific applications, but the free availability of the dataset does not constitute permission to reproduce or

publish it.

**5 Conclusion and outlook**

In this study, we developed a normalized tree biomass equation dataset (ChinAllomeTree version 1.0) based on an

extensive literature survey, which covered broad geographical, climatic and forest vegetation gradients across China. Our

dataset provides a major expansion in comparison to the biomass equation datasets currently available for China (Chen and

Zhu, 1989; Feng et al., 1999; Liang et al., 2006; Wang et al., 2005; Yuen et al., 2016; Zhang et al., 2013) and thus fills an

important regional gap relevant to global datasets (Henry et al., 2015). Our dataset also lays a solid data foundation for the

estimation of tree and forest biomass as well as general laws for plant allometric scaling. Moreover, this work highlights

five limitations and identifies the potential for future biomass equation research in China, as follows:



(1) There are still important gaps, and new equations, particularly for natural forests and most noncommercial tree species, are needed.

(2) To some extent, transparent and consistent protocols for tree biomass measurements, especially for the belowground sector, were lacking among studies. Moreover, belowground biomass was not measured or was measured inadequately in many studies.

(3) Component-wise biomass equations were always fitted without paying much attention to the additivity of biomass component equations in practice. To date, various model specification and parameter estimation methods have been proposed to ensure additivity, for example, seemly unrelated regression (Dong et al., 2015).

(4) The complete reports on biomass equations should cover the regression method, sample size, equation evaluation (e.g., $R^2$, error estimates of equations, standard errors of equation coefficients, and correction factors for log-linearized equations) and applicable ranges. However, these reports are often incomplete in current studies, largely due to the lack of uniform report standards.

(5) Limited sample trees with relatively narrow ranges of tree diameter and height were selected from small biotic (e.g., stand age and tree species) and abiotic (e.g., climate and soil) gradients. Additionally, large trees were often ignored in sampling campaigns. These limitations limit the applicability of the biomass equations. To overcome these drawbacks, further research is required to evaluate the quality and performance of these equations and develop generic biomass equations over broader ranges of abiotic and biotic conditions.

**Author contribution**

XW, ZO and YL originated, conceived and designed the work; YL, XW and FL developed and analyzed the equation dataset; all authors contributed to the writing of the manuscript.

**Competing interests**

The authors declare that they have no conflict of interest.



**Acknowledgments**

This study was supported by National Natural Science Foundation of China (31500388), National Key R&D Program of China (2018YFC0507303), Open Foundation of State Key Laboratory of Urban and Regional Ecology of China (SKLURE 2016-2-3), and China Postdoctoral Science Foundation (2016M601144 and 2017T100112).

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



**Tables**

Table 1: Summary of variable information in the dataset. '/' in the dataset denotes values that are not applicable for a variable, and 'na' does values that are not available for a variable.

| Variable | Description | Unit | Type |
|---|---|---|---|
| (1) General information sheet | | | |
| ID | Identification number of each study. | / | Integer |
| Province | Province location of study site. | / | String |
| Study site | Locality name of study site. | / | String |
| Latitude | Latitudes of study sites are either directly from original studies, or are extracted from Google Earth. | ° | Float |
| Longitude | Longitudes of study sites are either directly from original studies, or are extracted from Google Earth. | ° | Float |
| Altitude | Altitudes of study sites are either directly from original studies, or are extracted from Google Earth. | m | Integer |
| MAT | Mean annual temperatures of study sites are either directly from original studies, or are extracted from a 30 arc-seconds resolution global climate database WorldClim version 1.4 (Hijmans *et al*., 2005). | ℃ | Float |
| MAP | Mean annual precipitations of study sites are either directly from original studies, or are extracted from WorldClim version 1.4. | mm | Float |
| Forest type | Forest community characterized by the same tree genera, or if not genera, by ecological similarities (e.g. life form and biotope). | / | String |
| Dominant species | Dominant tree species of a forest type. In some forest types, there are | / | String |



| Variable | Description | Unit | Type |
|---|---|---|---|
| | two or more co-dominant tree species, and then the first four co-dominant species are listed at most. | | |
| Stand origin | Forests are classified by stand origin into natural and planted forests (including aerially seeded forest). | / | String |
| Miscellaneous information | Other information not mentioned in front columns such as site index and human disturbances (e.g. fertilization and selective logging), if available. | / | String |
| Tree species (group) | A tree species (group) that biomass equations are developed for. When equations are developed for two or more tree species, species name is specified as either a particular tree group (e.g. deciduous broadleaved trees) or 'generalized' according to the descriptions in original studies. | / | String |
| Stand age | The age of a natural forest is defined as age since germination, and the age of a planted forest is done as age since planting. | year | Float |
| Stand density | The number of trees per unit area. | trees/ha | Integer |
| Equations included | Equation numbers in 'Biomass equation sheet'. | / | Integer |
| Sources | Source of the data | / | String |
| (2) Biomass equation sheet | | | |
| Equation number | Identification number of each equation. | / | Integer |
| Biomass component | A tree biomass component divided in a certain way. $\Phi$, $s$ and $d$ denote root diameter, excavation area and excavation depth, respectively. | kg | String |
| Predictor variable | One or more dendrometric variables, i.e., tree diameter in cm and | cm; m | String |



| Variable | Description | Unit | Type |
|---|---|---|---|
| | height in m. | | |
| Equation form | It is used to develop a quantitative relationship between a biomass component (W) and predictor variable(s). | / | String |
| Equation coefficients | Equation coefficients consist of values of parameters $a$, $b$, $c$ and $d$, but not all four parameters are used in equations. | / | Float |
| Goodness-of-fit statistics | Goodness-of-fit statistics consist of $n$, $R^2$, $R$ and $CF$: | | |
| | (1) $n$: The number of harvested trees for developing biomass equation, although it is not always available in studies. | / | Integer |
| | (2) $R^2$: Coefficient of determination, a measure of goodness-of-fit. | / | Float |
| | (3) $R$: Correlation coefficient, another measure of goodness-of-fit. | / | Float |
| | (4) $CF$: Correction factor, it is for a log-linearized equation to correct the systematic bias in biomass estimates introduced by log-transformation, if available. Baskerville's CF (Baskerville, 1972) and Snowdon's CF (Snowdon, 1991) were employed by original studies, where the latter is marked with 'λ' in our dataset. | / | Float |
| Applicable ranges | Applicable ranges of equations consist of three parts: | | |
| | (1) Method: Method for determining value ranges (minimum, maximum) of predictor variables, whose descriptions are given in Table 2. | / | String |
| | (2) Diameter: Diameter ranges (minimum, maximum) | cm | Float |
| | (3) Height: Height ranges (minimum, maximum), if height is used as a predictor variable. | m | Float |



**Table 2: Methods for determining applicable ranges of biomass equations ***

| Method | Description |
|--------|-------------|
| I | Original studies presented tree diameter and height ranges (minimum, maximum) of harvested trees in the form of text, tables or figures. For former two forms, applicable ranges (diameter and height ranges) of biomass equations are determined directly, while for figures (e.g. biomass-diameter relationship and height-diameter relationship), they are extracted by using software GetData Graph Digitizer v.2.24. |
| II | When stand structures (or ranges) of diameter and height are available in original studies, they are considered as applicable ranges, although they may exceed actual ranges for equation building. |
| III | When mean and standard deviation (SD) of tree diameter and height are available, applicable ranges are estimated as (mean-2 SD, mean+2 SD), nearly covering 95% of normal stand distributions of tree diameter and height. |
| IV | When only mean values of tree diameter were provided without other statistics (e.g. SD), a rule of thumb is that diameter ranges are roughly estimated as (mean $\times 0.5$, mean $\times 1.5$). |
| V | When the above situations do not occur, applicable ranges of biomass equations are roughly estimated by using ones under similar phylogeny, age and growing environments. However, applicable ranges of some equations are not still obtained because of limited data. |

* According to the amount and reliability of information in original studies, five methods are employed in priority order: I > II > III > IV > V. Concerning those biomass equations with diameter and height as predictor variables, when only diameter ranges are determined, height ranges are estimated from: (1) biomass ranges, which are from original studies or could be calculated by using diameter-based equations if equations based on both diameter and height are available; or (2) height-diameter relationships (height-diameter curves or height/diameter ratios), which are from original studies, or are developed by using raw data of diameter and height within original studies or by using mean diameter and height data from Luo et al. (2013) (Table A1).





**Table 3: The number of retained biomass equations by tree species and biomass component. '–' denotes that there are no equations for a biomass component (group), and mixed species in Column 'Species name' does two or more tree species that equations are developed for. Abbreviations: SBs, stem biomass subcomponents (stem wood and bark); SB, stem biomass; BBs, branch biomass subcomponents (e.g. different aged branches); BB, branch biomass; LBs, leaf biomass subcomponents (different aged leaves); LB, leaf biomass; FF, flower and fruit biomass; CB, tree crown biomass (BB+LB); AW, aboveground woody biomass (SB+BB); AG, aboveground biomass (SB+BB+LB+FF); BGs, belowground biomass subcomponents (e.g. different diameter roots); BG, belowground biomass; TB, tree biomass (AG+BG).**

| No. | Species name | Number of studies | SBs | SB | BBs | BB | LBs | LB | FF | CB | AW | AG | BGs | BG | TB | Total |
|---|---|---|---|---|---|---|---|---|---|---|---|---|---|---|---|---|
| 1 | *Abies fabri* | 3 | 4 | 2 | – | 4 | – | 4 | – | – | – | 3 | – | 4 | 2 | 23 |
| 2 | *Abies georgei* | 1 | 2 | – | – | 1 | – | 1 | – | – | – | – | 2 | – | – | 6 |
| 3 | *Abies georgei* var. *smithii* | 1 | 4 | – | – | 2 | – | 2 | – | – | – | 2 | – | 2 | 2 | 14 |
| 4 | *Abies nephrolepis* | 1 | 4 | – | – | 2 | – | 2 | – | – | – | – | – | 2 | 2 | 12 |
| 5 | *Acacia auriculiformis* | 5 | 6 | 3 | – | 6 | – | 6 | – | – | – | 3 | – | 5 | 2 | 31 |
| 6 | *Acacia confusa* | 1 | – | 1 | – | 1 | – | 1 | – | – | – | 1 | – | – | – | 4 |
| 7 | *Acacia dealbata* | 3 | 2 | 3 | – | 4 | – | 4 | 1 | – | – | 1 | – | 4 | 4 | 23 |
| 8 | *Acacia mangium* | 6 | 6 | 5 | – | 8 | – | 8 | – | – | – | 4 | – | 3 | – | 34 |
| 9 | *Acacia mearnsii* | 1 | 1 | 1 | – | 1 | – | 1 | – | – | – | 1 | – | – | – | 5 |
| 10 | *Acer mandshuricum* | 1 | – | 1 | 2 | – | – | 1 | – | – | – | – | – | – | – | 4 |
| 11 | *Acer mono* | 7 | – | 9 | 10 | 5 | – | 9 | – | – | – | 3 | 2 | 5 | 3 | 46 |
| 12 | *Acer truncatum* | 1 | 2 | – | – | 1 | – | 1 | – | – | – | – | – | 1 | – | 5 |
| 13 | *Ailanthus altissima* | 1 | – | 1 | – | 1 | – | 1 | – | – | – | – | – | 1 | 1 | 5 |
| 14 | *Alniphyllum fortunei* | 2 | – | 2 | – | 2 | – | 2 | – | – | – | – | – | 2 | – | 8 |
| 15 | *Alnus cremastogyne* | 4 | 2 | 4 | – | 5 | – | 5 | – | – | – | 3 | – | 3 | 3 | 25 |

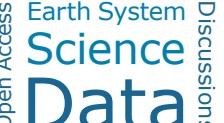

| No. | Species name | Number of studies | The number of biomass equations | | | | | | | | | | | | | |
|---|---|---|---|---|---|---|---|---|---|---|---|---|---|---|---|---|
| | | | SBs | SB | BBs | BB | LBs | LB | FF | CB | AW | AG | BGs | BG | TB | Total |
| 16 | *Alnus sibirica* | 4 | 8 | 1 | – | 5 | – | 5 | – | – | – | – | – | 5 | 4 | 28 |
| 17 | *Amygdalus persica* | 1 | – | 1 | – | 1 | – | 1 | – | – | – | – | – | 1 | 1 | 5 |
| 18 | *Aporusa yunnanensis* | 1 | – | 1 | – | 1 | – | 1 | – | – | – | – | – | 1 | 1 | 5 |
| 19 | *Azadirachta indica* | 1 | – | 2 | – | 2 | – | 2 | – | – | – | 2 | – | 2 | 2 | 12 |
| 20 | *Betula albosinensis* | 1 | 2 | – | – | 1 | – | 1 | 1 | – | – | – | – | 1 | – | 6 |
| 21 | *Betula alnoides* | 4 | 2 | 6 | – | 7 | – | 7 | – | – | – | – | – | 7 | 6 | 35 |
| 22 | *Betula costata* | 2 | 2 | 1 | 2 | 1 | – | 2 | – | – | – | 1 | – | – | – | 9 |
| 23 | *Betula dahurica* | 2 | – | 2 | – | 2 | – | 2 | – | – | – | 1 | 2 | 1 | 1 | 11 |
| 24 | *Betula luminifera* | 3 | 2 | 3 | – | 4 | – | 4 | – | – | – | 2 | – | 2 | 1 | 18 |
| 25 | *Betula platyphylla* | 18 | 26 | 8 | 8 | 19 | – | 21 | – | – | – | 4 | 2 | 16 | 11 | 115 |
| 26 | *Camellia oleifera* | 1 | 2 | – | 2 | – | – | 1 | – | – | – | – | 3 | – | 1 | 9 |
| 27 | *Caryota ochlandra* | 1 | – | 1 | – | 1 | – | 1 | – | – | – | – | – | 1 | – | 4 |
| 28 | *Castanopsis echidnocarpa* | 2 | – | 2 | – | 2 | – | 2 | – | – | – | – | – | 2 | – | 8 |
| 29 | *Castanopsis eyrei* | 1 | 4 | – | 4 | – | – | 2 | – | – | – | 2 | – | – | – | 12 |
| 30 | *Castanopsis fargesii* | 4 | 2 | 2 | 2 | 2 | – | 3 | – | – | – | 2 | – | 2 | – | 15 |
| 31 | *Castanopsis fissa* | 2 | 2 | 1 | – | 2 | – | 2 | – | – | – | – | – | 2 | 1 | 10 |
| 32 | *Castanopsis hystrix* | 2 | 2 | 1 | – | 2 | – | 2 | – | – | – | – | 2 | 1 | – | 10 |
| 33 | *Castanopsis kawakamii* | 1 | – | 1 | – | 1 | – | 1 | – | – | – | – | – | – | – | 3 |
| 34 | *Castanopsis orthacantha* | 1 | – | 1 | – | 1 | – | 1 | – | – | – | – | – | 1 | 1 | 5 |

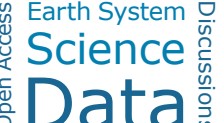



| No. | Species name | Number of studies | The number of biomass equations | | | | | | | | | | | | | |
|---|---|---|---|---|---|---|---|---|---|---|---|---|---|---|---|---|
| | | | SBs | SB | BBs | BB | LBs | LB | FF | CB | AW | AG | BGs | BG | TB | Total |
| 35 | *Castanopsis rufescens* | 1 | – | 1 | – | 1 | – | 1 | – | – | – | 1 | – | – | – | 4 |
| 36 | *Castanopsis sclerophylla* | 1 | – | – | – | – | – | – | – | – | – | – | – | 2 | – | 2 |
| 37 | *Casuarina equisetifolia* | 5 | 8 | 2 | – | 6 | – | 6 | – | – | – | – | – | 3 | – | 25 |
| 38 | *Celtis philippensis* | 1 | – | 1 | – | 1 | – | 1 | – | – | – | – | – | 1 | 1 | 5 |
| 39 | *Cercidiphyllum japonicum* | 1 | 4 | – | – | 2 | – | 2 | – | – | – | 2 | 4 | 2 | 2 | 18 |
| 40 | *Choerospondias axillaris* | 1 | – | 1 | – | 1 | – | 1 | – | – | – | – | – | 1 | 1 | 5 |
| 41 | *Cinnamomum bodinieri* | 2 | – | 2 | – | 2 | – | 2 | – | – | – | – | – | 2 | 2 | 10 |
| 42 | *Cinnamomum camphora* | 6 | 10 | 3 | – | 8 | – | 8 | – | – | – | 2 | 6 | 8 | 6 | 51 |
| 43 | *Citrus reticulata* | 1 | – | 1 | – | 1 | – | 1 | – | – | – | – | – | 1 | 1 | 5 |
| 44 | *Cleidion brevipetiolatum* | 1 | – | 1 | – | 1 | – | 1 | – | – | – | – | – | 1 | 1 | 5 |
| 45 | *Cleistanthus sumatranus* | 1 | – | 1 | – | 1 | – | 1 | – | – | – | – | – | 1 | 1 | 5 |
| 46 | *Cryptocarya chinensis* | 1 | – | 1 | – | 1 | – | 1 | – | – | – | – | – | 1 | 1 | 5 |
| 47 | *Cryptocarya concinna* | 1 | – | 1 | – | 1 | – | 1 | – | – | – | – | – | 1 | 1 | 5 |
| 48 | *Cryptomeria fortunei* | 4 | 6 | 4 | – | 5 | – | 5 | – | 2 | – | 4 | – | 7 | 5 | 38 |
| 49 | *Cryptomeria japonica* | 2 | 4 | – | – | 2 | – | 2 | – | – | – | – | – | 2 | 2 | 12 |
| 50 | *Cunninghamia lanceolata* | 130 | 152 | 70 | 2 | 140 | 1 | 141 | 4 | 4 | – | 31 | 25 | 106 | 30 | 706 |
| 51 | *Cupressus funebris* | 4 | 2 | 3 | – | 4 | – | 4 | – | – | – | 2 | – | 4 | 3 | 22 |
| 52 | *Cupressus lusitanica* | 1 | – | 1 | – | 1 | – | 1 | – | – | – | – | – | 1 | 1 | 5 |
| 53 | *Cyclobalanopsis delavayi* | 1 | – | 1 | – | 1 | – | 1 | – | – | – | – | – | 1 | 1 | 5 |

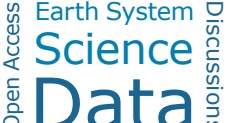

| No. | Species name | Number of studies | The number of biomass equations | | | | | | | | | | | | | |
| --- | --- | --- | --- | --- | --- | --- | --- | --- | --- | --- | --- | --- | --- | --- | --- | --- |
| | | | SBs | SB | BBs | BB | LBs | LB | FF | CB | AW | AG | BGs | BG | TB | Total |
| 54 | *Cyclobalanopsis glauca* | 8 | 6 | 9 | 6 | 9 | – | 12 | 3 | – | – | 11 | 11 | 6 | 6 | 79 |
| 55 | *Elaeocarpus decipiens* | 1 | – | 1 | – | 1 | – | 1 | – | – | – | – | – | 1 | 1 | 5 |
| 56 | *Elaeocarpus sylvestris* | 2 | 4 | – | – | 2 | – | 2 | – | – | – | – | – | 2 | 1 | 11 |
| 57 | *Engelhardtia roxburghiana* | 1 | – | 1 | – | 1 | – | 1 | – | – | – | – | – | 1 | – | 4 |
| 58 | *Erythrophleum fordii* | 1 | 2 | – | – | 1 | – | 1 | – | – | – | – | – | 1 | – | 5 |
| 59 | *Eucalyptus camaldulensis* | 2 | – | 2 | – | 2 | – | 2 | – | – | – | 2 | – | 2 | – | 10 |
| 60 | *Eucalyptus citriodora* | 1 | 4 | – | – | 2 | – | 2 | – | – | – | 2 | 4 | 2 | 2 | 18 |
| 61 | *Eucalyptus exserta* | 2 | 2 | 1 | – | 2 | – | 2 | 1 | – | – | – | – | 2 | – | 10 |
| 62 | *Eucalyptus globulus* | 1 | 2 | – | – | 1 | – | 1 | – | – | – | – | – | 3 | – | 7 |
| 63 | *Eucalyptus grandis × E. urophylla* | 4 | 8 | – | – | 4 | – | 4 | 3 | – | – | – | – | 1 | – | 20 |
| 64 | *Eucalyptus leizhouensis No.1* | 2 | 8 | – | – | 4 | – | 4 | – | – | – | 4 | 4 | 4 | 4 | 32 |
| 65 | *Eucalyptus urophylla* | 8 | 24 | – | – | 12 | – | 12 | – | – | – | 8 | – | 4 | – | 60 |
| 66 | *Eucalyptus urophylla × E. grandis* | 7 | 16 | – | – | 8 | – | 8 | – | – | – | 2 | 4 | 7 | 2 | 47 |
| 67 | *Eucommia ulmoides* | 6 | 12 | 1 | – | 7 | – | 7 | – | – | – | 2 | – | 7 | 5 | 41 |
| 68 | *Fagus engleriana* | 1 | – | 2 | – | 2 | – | 2 | – | – | – | – | – | 2 | – | 8 |
| 69 | *Ficus microcarpa* | 1 | – | 1 | – | 1 | – | 1 | – | – | – | – | – | 1 | – | 4 |



| No. | Species name | Number of studies | The number of biomass equations | | | | | | | | | | | | | |
|---|---|---|---|---|---|---|---|---|---|---|---|---|---|---|---|---|
| | | | SBs | SB | BBs | BB | LBs | LB | FF | CB | AW | AG | BGs | BG | TB | Total |
| 70 | *Fokienia hodginsii* | 3 | 6 | 2 | – | 5 | – | 5 | – | 2 | – | – | – | 3 | 3 | 26 |
| 71 | *Fraxinus mandschurica* | 7 | – | 9 | 10 | 5 | – | 9 | – | – | – | 3 | 2 | 3 | 3 | 44 |
| 72 | *Fraxinus rhynchophylla* | 1 | – | 2 | – | 2 | – | 2 | – | – | – | – | – | 2 | – | 8 |
| 73 | *Ginkgo biloba* | 1 | – | 1 | – | 1 | – | 1 | – | – | – | – | – | 1 | 1 | 5 |
| 74 | *Gordonia acuminata* | 2 | – | 2 | – | 2 | – | 2 | – | – | – | – | – | 2 | – | 8 |
| 75 | *Hevea brasiliensis* | 8 | – | 12 | – | 12 | – | 12 | – | – | – | 7 | 4 | 7 | 7 | 61 |
| 76 | *Idesia polycarpa* | 1 | – | 2 | – | – | – | – | – | 2 | – | 2 | – | – | – | 6 |
| 77 | *Juglans mandshurica* | 3 | 2 | 3 | 2 | 4 | – | 4 | – | – | – | 1 | 2 | 2 | 1 | 21 |
| 78 | *Keteleeria davidiana* | 1 | – | 2 | – | 2 | – | 2 | – | – | – | 2 | – | 2 | 2 | 12 |
| 79 | *Koelreuteria bipinnata* var. *integrifoliola* | 1 | 2 | – | – | 1 | – | 1 | – | – | – | – | – | 1 | 1 | 6 |
| 80 | *Koelreuteria paniculata* | 1 | – | 1 | – | 1 | – | 1 | – | – | – | – | – | 1 | 1 | 5 |
| 81 | *Larix chinensis* | 2 | – | 2 | – | 2 | – | 2 | – | – | – | – | – | 1 | 1 | 8 |
| 82 | *Larix gmelinii* | 27 | 30 | 17 | – | 32 | 2 | 32 | – | – | – | 10 | 2 | 22 | 10 | 157 |
| 83 | *Larix kaempferi* | 7 | 10 | 6 | – | 11 | – | 11 | – | – | – | 3 | – | 11 | 9 | 61 |
| 84 | *Larix mastersiana* | 1 | 4 | – | – | 2 | – | 2 | – | – | – | – | – | 2 | 2 | 12 |
| 85 | *Larix olgensis* | 8 | 10 | 6 | – | 10 | – | 10 | – | 1 | – | 5 | – | 8 | 6 | 56 |
| 86 | *Larix principis-rupprechtii* | 30 | 32 | 27 | 6 | 41 | – | 43 | – | – | – | 20 | – | 38 | 28 | 235 |
| 87 | *Lasiococca comberi* | 1 | – | 1 | – | 1 | – | 1 | – | – | – | – | – | 1 | 1 | 5 |





| No. | Species name | Number of studies | The number of biomass equations | | | | | | | | | | | | | |
|---|---|---|---|---|---|---|---|---|---|---|---|---|---|---|---|---|
| | | | SBs | SB | BBs | BB | LBs | LB | FF | CB | AW | AG | BGs | BG | TB | Total |
| 88 | *Ligustrum lucidum* | 2 | – | 2 | – | 2 | – | 2 | – | – | – | – | – | 2 | 2 | 10 |
| 89 | *Liquidambar formosana* | 3 | 2 | 2 | – | 3 | – | 3 | – | – | – | – | – | 3 | – | 13 |
| 90 | *Liriodendron chinense* | 2 | 4 | 1 | – | 3 | – | 3 | – | – | – | 3 | – | 2 | 2 | 18 |
| 91 | *Lithocarpus craibianus* | 1 | – | 1 | – | 1 | – | 1 | – | – | – | – | – | 1 | 1 | 5 |
| 92 | *Lithocarpus glaber* | 2 | 4 | 2 | 4 | 2 | – | 4 | 2 | – | – | 4 | 4 | 2 | 2 | 30 |
| 93 | *Lithocarpus xylocarpus* | 1 | – | 1 | – | 1 | – | 1 | – | – | – | 1 | – | – | – | 4 |
| 94 | *Litsea cubeba* | 1 | – | 1 | – | 1 | – | 1 | 1 | – | – | – | – | 1 | 1 | 6 |
| 95 | *Litsea pungens* | 1 | – | 1 | – | 1 | – | 1 | – | – | – | – | – | 1 | – | 4 |
| 96 | *Macaranga denticulata* | 1 | – | 1 | – | 1 | – | 1 | – | – | – | – | – | 1 | – | 4 |
| 97 | *Machilus pauhoi* | 1 | 2 | – | – | 1 | – | 1 | – | – | – | 1 | – | – | – | 5 |
| 98 | *Machilus viridis* | 1 | – | 1 | – | 1 | – | 1 | – | – | – | 1 | – | – | – | 4 |
| 99 | *Magnolia officinalis* | 2 | 6 | – | – | 3 | – | 3 | – | – | – | 3 | 2 | 1 | 1 | 19 |
| 100 | *Magnolia officinalis* subsp. *biloba* | 1 | 2 | – | – | 1 | – | 1 | – | – | – | – | – | 1 | 1 | 6 |
| 101 | *Mallotus paniculatus* | 3 | – | 3 | – | 3 | – | 3 | – | – | – | – | – | 3 | – | 12 |
| 102 | *Malus pumila* | 1 | – | 1 | – | 1 | – | 1 | 1 | – | – | – | – | 1 | – | 5 |
| 103 | *Manglietia glauca* | 1 | 2 | – | – | 1 | – | 1 | – | – | – | – | 4 | – | – | 8 |
| 104 | *Manglietia hainanensis* | 1 | 4 | – | – | 2 | – | 2 | – | – | – | 2 | – | – | – | 10 |
| 105 | *Manglietia insignis* | 1 | – | 1 | – | 1 | – | 1 | – | – | – | 1 | – | – | – | 4 |
| 106 | *Metasequoia* | 9 | 4 | 11 | – | 13 | – | 13 | – | – | – | 8 | – | 8 | 8 | 65 |

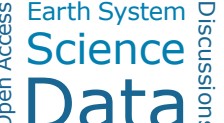


| No. | Species name | Number of studies | The number of biomass equations | | | | | | | | | | | | | |
|---|---|---|---|---|---|---|---|---|---|---|---|---|---|---|---|---|
| | | | SBs | SB | BBs | BB | LBs | LB | FF | CB | AW | AG | BGs | BG | TB | Total |
| | *glyptostroboides* | | | | | | | | | | | | | | | |
| 107 | *Michelia hedyosperma* | 1 | 2 | – | – | 1 | – | 1 | – | – | – | 1 | – | 1 | 1 | 7 |
| 108 | *Michelia macclurei* | 3 | 2 | 2 | – | 3 | – | 3 | – | – | – | – | – | 3 | 1 | 14 |
| 109 | *Millettia laptobotrya* | 1 | – | 1 | – | 1 | – | 1 | – | – | – | – | – | 1 | – | 4 |
| 110 | *Mytilaria laosensis* | 3 | 6 | 1 | – | 4 | – | 4 | – | – | – | – | 4 | 2 | 2 | 23 |
| 111 | *Ormosia hosiei* | 1 | – | 1 | – | 1 | – | 1 | – | – | – | – | – | 1 | – | 4 |
| 112 | *Ormosia xylocarpa* | 1 | – | 1 | – | 1 | – | 1 | – | – | – | – | – | 1 | – | 4 |
| 113 | *Paramichelia baillonii* | 1 | 2 | – | – | 1 | – | 1 | – | – | – | 1 | – | 1 | 1 | 7 |
| 114 | *Parashorea chinensis* | 1 | 2 | – | – | 1 | – | 1 | – | – | – | 1 | 4 | 1 | 1 | 11 |
| 115 | *Paulownia elongata* | 7 | 2 | 10 | – | 11 | – | 11 | 3 | 8 | – | 9 | – | 11 | 10 | 75 |
| 116 | *Paulownia tomentosa ×* *P. fortunei* | 1 | – | 1 | – | 1 | – | 1 | – | – | – | 1 | – | – | – | 4 |
| 117 | *Phellodendron amurense* | 2 | – | 3 | 2 | 3 | – | 3 | – | – | – | 1 | 2 | 1 | 1 | 16 |
| 118 | *Phellodendron chinense* | 3 | 4 | 1 | 2 | 2 | – | 3 | – | – | – | – | 2 | 2 | 3 | 19 |
| 119 | *Phoebe bournei* | 1 | 2 | – | – | 1 | – | 1 | – | – | – | 1 | 5 | 1 | – | 11 |
| 120 | *Phoebe zhennan* | 2 | – | 2 | – | 2 | – | 2 | – | – | – | 1 | – | 2 | 2 | 11 |
| 121 | *Picea asperata* | 2 | 2 | 2 | – | 3 | – | 3 | – | – | – | 2 | – | 3 | 2 | 17 |
| 122 | *Picea brachytyla* var. *complanata* | 1 | 2 | – | – | 1 | – | 1 | – | – | – | – | 2 | – | – | 6 |
| 123 | *Picea crassifolia* | 3 | 6 | 2 | – | 5 | – | 5 | 2 | – | – | 4 | – | 3 | 2 | 29 |

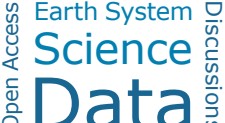

| No. | Species name | Number of studies | The number of biomass equations | | | | | | | | | | | | | |
|---|---|---|---|---|---|---|---|---|---|---|---|---|---|---|---|---|
| | | | SBs | SB | BBs | BB | LBs | LB | FF | CB | AW | AG | BGs | BG | TB | Total |
| 124 | *Picea koraiensis* | 2 | 2 | 1 | – | 2 | – | 2 | – | – | – | 2 | – | 2 | 1 | 12 |
| 125 | *Picea likiangensis* var. *balfouriana* | 1 | 2 | – | – | 1 | – | 1 | – | – | – | – | – | 1 | – | 5 |
| 126 | *Picea purpurea* | 1 | 2 | – | – | 1 | – | 1 | – | – | – | – | – | 1 | – | 5 |
| 127 | *Picea schrenkiana* | 2 | 6 | 1 | – | 3 | – | 3 | – | 1 | – | 3 | – | – | – | 17 |
| 128 | *Pinus armandii* | 8 | 18 | – | – | 9 | 9 | 9 | – | – | – | – | – | 9 | 2 | 56 |
| 129 | *Pinus bungeana* | 1 | 4 | 2 | – | 2 | – | 2 | – | – | – | 2 | 4 | 2 | 2 | 20 |
| 130 | *Pinus densata* | 2 | 2 | 2 | – | 3 | – | 3 | – | – | – | 2 | 2 | – | – | 14 |
| 131 | *Pinus elliottii* | 13 | 14 | 9 | 2 | 15 | – | 16 | – | – | – | 7 | 3 | 7 | 3 | 76 |
| 132 | *Pinus fenzeliana* | 1 | 2 | – | – | 1 | – | 1 | – | – | – | – | – | 1 | – | 5 |
| 133 | *Pinus henryi* | 1 | 2 | – | – | 1 | – | 1 | – | – | – | – | – | 1 | – | 5 |
| 134 | *Pinus kesiya* var. *langbianensis* | 4 | 8 | – | – | 4 | – | 4 | 1 | – | – | – | 8 | – | – | 25 |
| 135 | *Pinus koraiensis* | 32 | 8 | 40 | 8 | 44 | 14 | 41 | – | 3 | – | 6 | 20 | 19 | 15 | 218 |
| 136 | *Pinus massoniana* | 60 | 56 | 46 | 2 | 73 | – | 75 | 1 | 1 | 1 | 19 | 3 | 56 | 32 | 365 |
| 137 | *Pinus sylvestris* var. *mongolica* | 4 | – | 5 | – | 5 | – | 5 | – | – | – | 3 | – | 1 | – | 19 |
| 138 | *Pinus sylvestris* var. *sylvestriformis* | 3 | 2 | 2 | – | 3 | – | 3 | – | – | – | 2 | – | 1 | 1 | 14 |
| 139 | *Pinus tabuliformis* | 46 | 73 | 40 | – | 63 | 4 | 63 | 6 | 2 | – | 32 | 42 | 51 | 19 | 395 |





| No. | Species name | Number of studies | The number of biomass equations | | | | | | | | | | | | | |
|---|---|---|---|---|---|---|---|---|---|---|---|---|---|---|---|---|
| | | | SBs | SB | BBs | BB | LBs | LB | FF | CB | AW | AG | BGs | BG | TB | Total |
| 140 | *Pinus taeda* | 6 | 4 | 8 | – | 8 | – | 9 | – | – | 1 | 7 | – | 6 | 6 | 49 |
| 141 | *Pinus taiwanensis* | 9 | 2 | 11 | – | 12 | – | 12 | – | – | – | 5 | 4 | 13 | 7 | 66 |
| 142 | *Pinus thunbergii* | 2 | 2 | 2 | – | 3 | – | 3 | – | – | – | 1 | – | 3 | 3 | 17 |
| 143 | *Pinus yunnanensis* | 8 | 6 | 5 | – | 8 | – | 8 | – | – | – | 1 | – | 8 | 2 | 38 |
| 144 | *Platycladus orientalis* | 10 | – | 11 | – | 11 | – | 11 | – | – | – | 3 | – | 7 | 1 | 44 |
| 145 | *Podocarpus imbricatus* | 1 | 2 | – | – | 1 | – | 1 | – | – | – | – | – | 1 | 1 | 6 |
| 146 | *Populus alba* | 1 | 2 | – | – | 1 | – | 1 | – | – | – | – | – | 1 | 1 | 6 |
| 147 | *Populus alba* var. *pyramidalis* | 2 | 4 | – | 2 | 1 | – | 2 | – | – | – | – | 2 | 1 | 2 | 14 |
| 148 | *Populus canadensis* cv. 'I-214' | 1 | – | 1 | – | 1 | – | 1 | – | – | – | – | – | 1 | – | 4 |
| 149 | *Populus canadensis* cv. 'I-69' | 4 | 4 | 5 | 4 | 5 | – | 5 | – | – | – | 3 | 4 | 5 | 3 | 38 |
| 150 | *Populus canadensis* cv. 'I-72' | 9 | 10 | 7 | 4 | 10 | – | 12 | – | – | – | 6 | 4 | 7 | 6 | 66 |
| 151 | *Populus canadensis* cv. 'Neva' | 1 | – | 1 | – | 1 | – | 1 | – | – | – | – | – | – | – | 3 |
| 152 | *Populus canadensis* cv. 'Robusta' | 1 | – | 1 | – | 1 | – | 1 | – | – | – | – | – | 1 | – | 4 |
| 153 | *Populus canadensis* cv. | 1 | – | 1 | – | 1 | – | 1 | – | – | – | – | – | 1 | – | 4 |

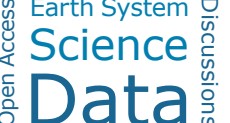



| No. | Species name | Number of studies | The number of biomass equations | | | | | | | | | | | | | |
|---|---|---|---|---|---|---|---|---|---|---|---|---|---|---|---|---|
| | | | SBs | SB | BBs | BB | LBs | LB | FF | CB | AW | AG | BGs | BG | TB | Total |
| | 'Sacrau-79' | | | | | | | | | | | | | | | |
| 154 | *Populus canadensis* cv. 'Zhonglin-46' | 1 | – | 2 | – | 2 | – | 2 | – | – | – | – | – | 2 | 2 | 10 |
| 155 | *Populus dakuanensis* | 1 | – | 2 | – | 2 | – | 2 | – | – | – | – | – | 2 | 2 | 10 |
| 156 | *Populus davidiana* | 8 | 8 | 5 | 2 | 9 | – | 9 | – | – | – | 4 | 2 | 7 | 5 | 51 |
| 157 | *Populus deltoides* | 2 | 4 | – | – | 2 | – | 2 | – | – | – | – | – | 2 | – | 10 |
| 158 | *Populus deltoides* cv. '35' | 1 | 2 | – | – | 1 | – | 1 | – | – | – | – | – | 1 | 1 | 6 |
| 159 | *Populus euphratica* | 4 | – | 4 | – | 4 | – | 4 | – | – | – | 1 | – | 4 | 1 | 18 |
| 160 | *Populus hopeiensis* | 1 | – | 2 | – | 2 | – | 2 | – | – | – | 2 | – | 2 | 2 | 12 |
| 161 | *Populus jrtyschensis* | 1 | 2 | – | – | 1 | – | 1 | – | – | – | – | – | 1 | 1 | 6 |
| 162 | *Populus laurifolia* | 1 | 2 | – | – | 1 | – | 1 | – | – | – | – | – | 1 | 1 | 6 |
| 163 | *Populus szechuanica* var. *tibetica* | 1 | 2 | – | – | 1 | – | 1 | – | – | – | – | – | 1 | 1 | 6 |
| 164 | *Populus tomentosa* | 10 | 26 | 5 | – | 18 | – | 18 | – | – | – | 1 | – | 16 | 16 | 100 |
| 165 | *Populus ussuriensis* | 2 | – | 2 | – | 2 | – | 2 | – | – | – | 1 | – | 1 | 1 | 9 |
| 166 | *Populus wenxianica* | 1 | – | 2 | – | 2 | – | 2 | – | – | – | 2 | – | – | – | 8 |
| 167 | *Populus xiaohei* | 4 | 10 | – | – | 5 | – | 5 | – | – | – | – | – | 5 | 3 | 28 |
| 168 | *Quercus acutissima* | 4 | – | 4 | – | 4 | – | 4 | – | – | – | 2 | – | 2 | 1 | 17 |
| 169 | *Quercus aliena* var. *acutiserrata* | 7 | 14 | 2 | – | 9 | – | 9 | – | – | – | – | – | 8 | 2 | 44 |





| No. | Species name | Number of studies | The number of biomass equations | | | | | | | | | | | | | |
|---|---|---|---|---|---|---|---|---|---|---|---|---|---|---|---|---|
| | | | SBs | SB | BBs | BB | LBs | LB | FF | CB | AW | AG | BGs | BG | TB | Total |
| 170 | *Quercus fabrei* | 1 | 2 | – | – | 1 | – | 1 | – | – | – | – | – | 1 | – | 5 |
| 171 | *Quercus mongolica* | 9 | 4 | 9 | 8 | 8 | – | 11 | – | – | – | 2 | 2 | 7 | 3 | 54 |
| 172 | *Quercus pannosa* | 2 | 2 | 1 | – | 2 | – | 2 | – | – | – | – | 2 | 1 | – | 10 |
| 173 | *Quercus senescens* | 1 | 2 | – | – | 1 | – | 1 | – | – | – | – | 2 | – | – | 6 |
| 174 | *Quercus variabilis* | 5 | 10 | 2 | – | 7 | – | 7 | – | – | – | 2 | 6 | 7 | 4 | 45 |
| 175 | *Quercus wutaishanica* | 2 | 4 | – | – | 2 | – | 2 | – | – | – | – | – | 2 | – | 10 |
| 176 | *Rhus chinensis* | 1 | – | 1 | – | 1 | – | 1 | – | – | – | 1 | – | 1 | 1 | 6 |
| 177 | *Rhus punjabensis* var. *sinica* | 1 | – | 1 | – | 1 | – | 1 | – | – | – | 1 | – | 1 | 1 | 6 |
| 178 | *Robinia pseudoacacia* | 16 | 18 | 11 | – | 20 | – | 20 | 2 | – | – | – | – | 16 | 9 | 96 |
| 179 | *Sabina przewalskii* | 1 | – | 1 | – | 1 | – | 1 | 1 | – | – | – | – | 1 | 1 | 6 |
| 180 | *Salix alba* | 1 | 2 | – | – | 1 | – | 1 | – | – | – | – | – | 1 | 1 | 6 |
| 181 | *Sassafras tzumu* | 2 | – | 1 | – | 1 | – | 1 | – | – | – | 1 | – | 2 | – | 6 |
| 182 | *Schima superba* | 6 | 4 | 4 | 4 | 4 | – | 7 | – | – | 1 | 4 | – | 4 | 2 | 34 |
| 183 | *Schima wallichii* | 1 | – | 1 | – | 1 | – | 1 | – | – | – | – | – | – | – | 3 |
| 184 | *Sumbaviopsis albicans* | 1 | – | 1 | – | 1 | – | 1 | – | – | – | – | – | 1 | 1 | 5 |
| 185 | *Symplocos anomala* | 1 | 2 | – | – | 1 | – | 1 | – | – | – | – | – | 1 | – | 5 |
| 186 | *Symplocos sumuntia* | 1 | 2 | – | – | 1 | – | 1 | – | – | – | – | – | 1 | – | 5 |
| 187 | *Syzygium jambos* | 1 | – | 1 | – | 1 | – | 1 | – | – | – | – | – | 1 | – | 4 |
| 188 | *Ternstroemia* | 1 | – | 1 | – | 1 | – | 1 | – | – | – | – | – | 1 | 1 | 5 |

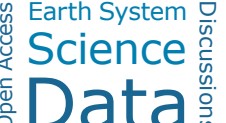



| No. | Species name | Number of studies | The number of biomass equations | | | | | | | | | | | | | |
|---|---|---|---|---|---|---|---|---|---|---|---|---|---|---|---|---|
| | | | SBs | SB | BBs | BB | LBs | LB | FF | CB | AW | AG | BGs | BG | TB | Total |
| | *gymnanthera* | | | | | | | | | | | | | | | |
| 189 | *Tilia amurensis* | 7 | – | 9 | 10 | 5 | – | 9 | – | – | – | 5 | 2 | 5 | 5 | 50 |
| 190 | *Tilia mongolica* | 1 | – | 2 | – | 2 | – | 2 | – | – | – | – | – | 2 | – | 8 |
| 191 | *Trema tomentosa* | 1 | – | 1 | – | 1 | – | 1 | – | – | – | – | – | 1 | – | 4 |
| 192 | *Tsoongiodendron odorum* | 2 | 2 | 2 | – | 3 | – | 3 | – | – | – | 2 | – | 3 | 3 | 18 |
| 193 | *Ulmus davidiana* var. *japonica* | 4 | – | 4 | 6 | 1 | – | 4 | – | – | – | 1 | – | 1 | 1 | 18 |
| 194 | *Ulmus pumila* | 2 | – | 2 | – | 2 | – | 2 | – | – | – | – | 2 | 1 | 1 | 10 |
| 195 | *Vernicia fordii* | 1 | – | 2 | – | 2 | – | 2 | 4 | – | – | – | – | 2 | 2 | 14 |
| 196 | *Vernicia montana* | 1 | – | 1 | – | 1 | – | 1 | – | – | – | – | – | 1 | 1 | 5 |
| 197 | *Zanthoxylum ailanthoides* | 1 | – | 1 | – | 1 | – | 1 | – | – | – | – | – | 1 | – | 4 |
| 198 | Mixed species | 69 | 30 | 75 | – | 88 | – | 88 | 6 | – | – | 21 | 22 | 73 | 33 | 436 |
| | Total | 906 | 910 | 694 | 116 | 1074 | 30 | 1116 | 43 | 26 | 3 | 364 | 246 | 837 | 465 | 5924 |



**Figures**

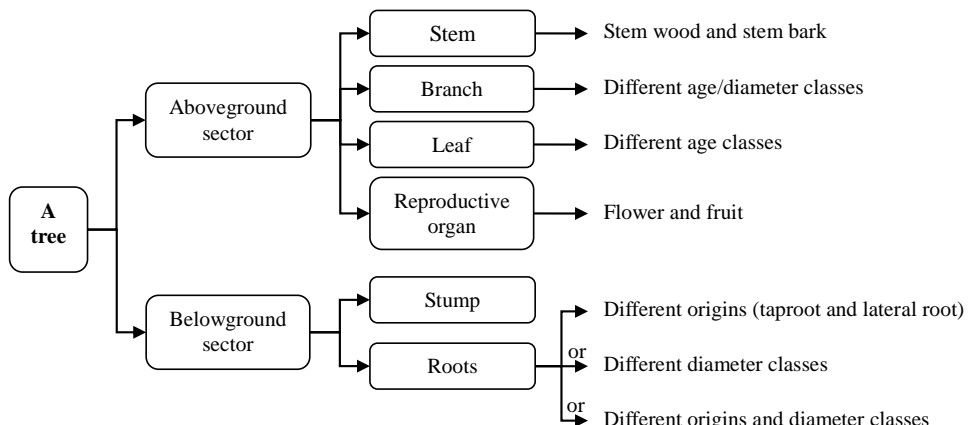

**Figure 1: The division of tree biomass components. A tree can be divided into (1) aboveground sector above the soil surface and**

5 **(2) belowground sector, which are often subdivided into finer components.**

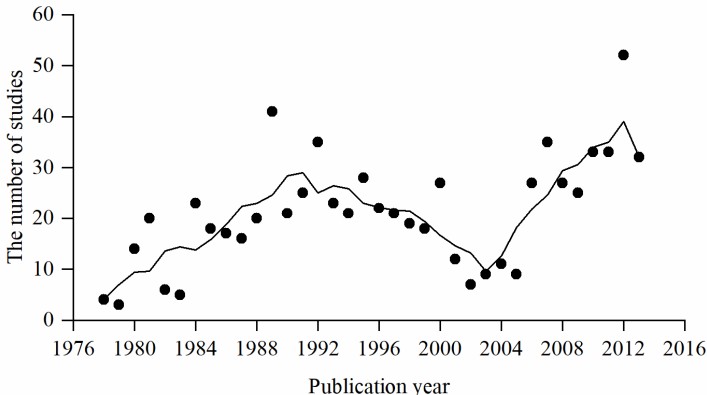

**Figure 2: Temporal change of compiled studies during the period 1978-2013. Trend line is smoothed by using an adjacent**

10 **5-point averaging method.**





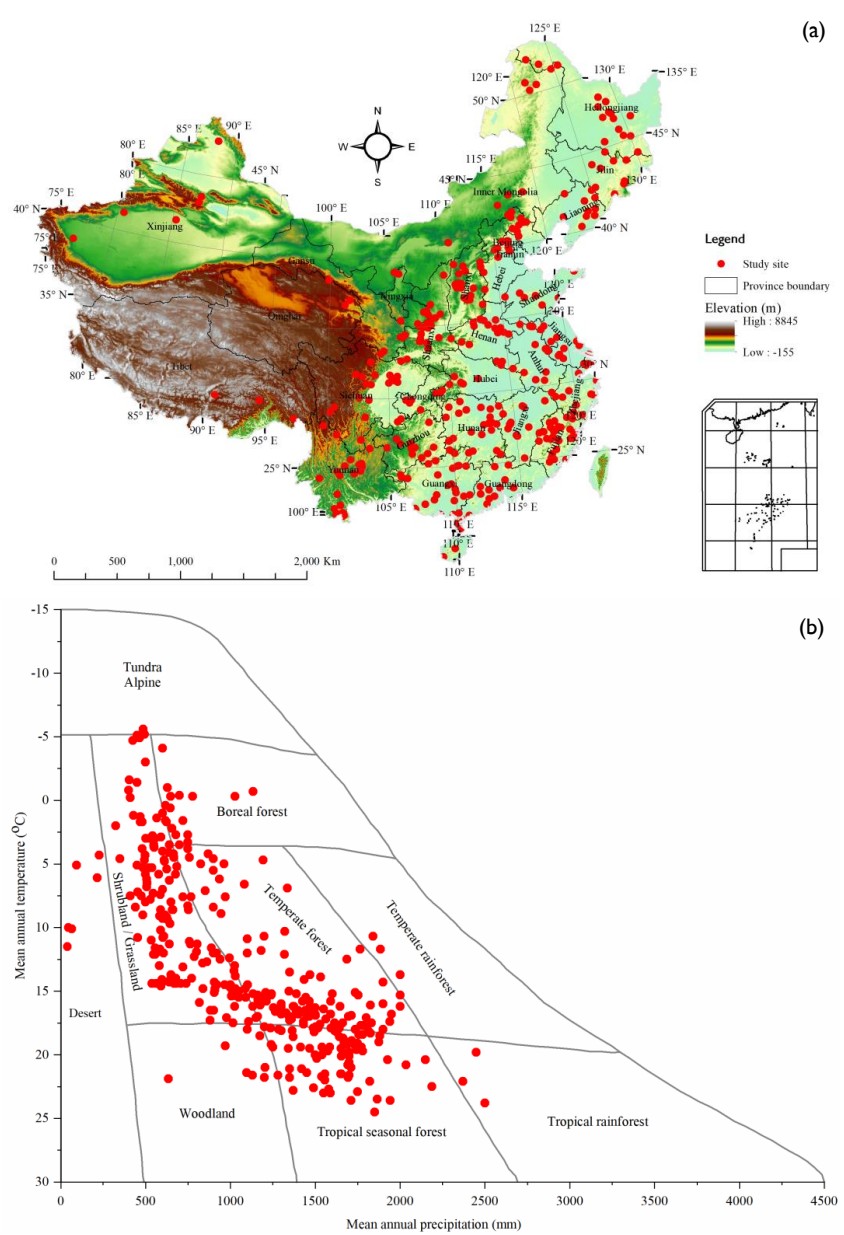

**Figure 3: Spatial distribution of study sites: (a) geographical coverage and (b) climate space. Mean annual temperature and precipitation of sites are superimposed upon Whittaker's climate-biome diagram (Whittaker, 1975).**



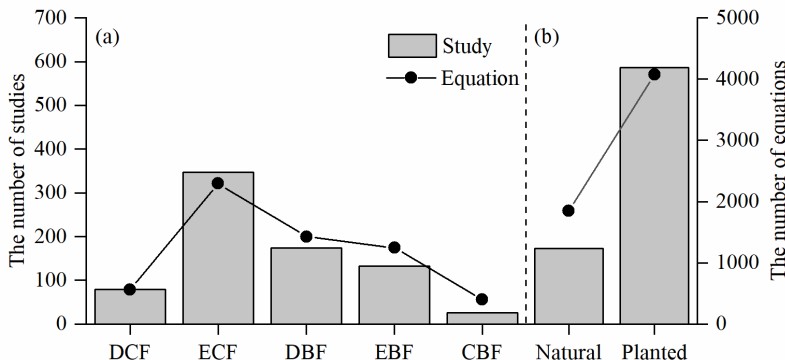

**Figure 4: Distribution of compiled studies and biomass equations by (a) forest type and (b) stand origin. Forests are categorized by forest type into deciduous coniferous forest (DCF), evergreen coniferous forest (ECF), deciduous broadleaved forest (DBF), evergreen broadleaved forest (EBF), and coniferous and broadleaved mixed forest (CBF). and by stand origin into natural forest and planted forest.**

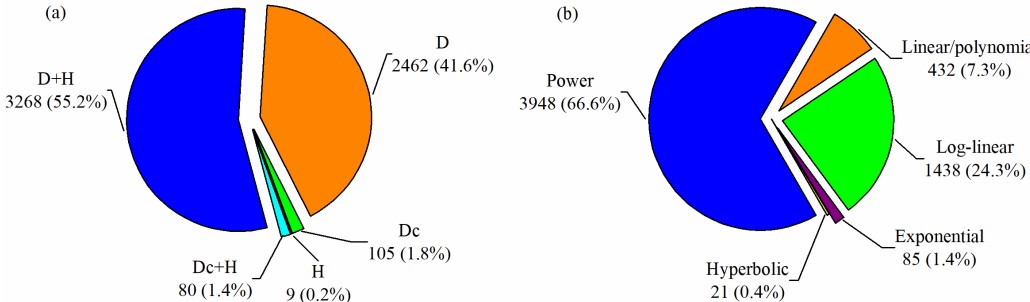

**Figure 5: Distribution of biomass equations by (a) predictor variable and (b) equation form. D and H are diameter at breast height (1.3 m) and height, and Dc is tree diameter at other heights (e.g. 0 m, 0.2 m, and 0.3 m) rather than breast height. Equation forms used in original studies are categorized into power equation, log-linear equation, linear/polynomial equation, exponential equation and hyperbolic equation (Table A2).**

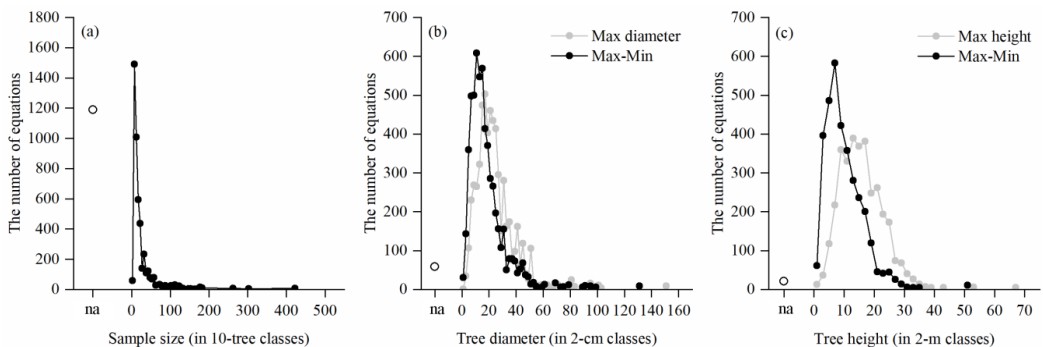

**Figure 6: Distribution of sample size and applicable range of biomass equations: (a) sample size, (b) tree diameter and (c) tree height. The dots represent the number of equations within each class. If sample sizes and applicable ranges are not available, they are indicated by 'na'.**



**Appendix A. Height-diameter curves and biomass equation forms**

**Table A1: Height-diameter curves for main tree species (group) ***

| No. | Tree species (group) † | $a$ (S.E.) | $b$ (S.E.) | $n$ | $R^2$ |
|---|---|---|---|---|---|
| 1 | *Abies, Picea* | 1.1457 (0.1626) | 0.9093 (0.0517) | 30 | 0.917 |
| 2 | *Cunninghamia lanceolata* | 0.7226 (0.0286) | 1.0492 (0.0160) | 236 | 0.948 |
| 3 | *Cupressus* | 0.9808 (0.3725) | 0.8966 (0.1420) | 18 | 0.714 |
| 4 | *Larix* | 1.8234 (0.1739) | 0.7541 (0.0422) | 85 | 0.794 |
| 5 | *Pinus massoniana, P. taiwanensis* | 0.8895 (0.0726) | 0.9910 (0.0325) | 85 | 0.918 |
| 6 | *P. tabuliformis* | 1.0951 (0.1066) | 0.8184 (0.0428) | 106 | 0.778 |
| 7 | Other temperate conifers | 1.2506 (0.1743) | 0.7810 (0.0546) | 75 | 0.737 |
| 8 | Other subtropical conifers | 0.7682 (0.2594) | 0.9740 (0.1307) | 50 | 0.536 |
| 9 | *Populus* | 2.0623 (0.4852) | 0.6679 (0.0881) | 32 | 0.657 |
| 10 | Temperate deciduous broadleaved trees | 1.8784 (0.3111) | 0.7087 (0.0689) | 51 | 0.683 |
| 11 | Subtropical deciduous broadleaved trees | 1.5194 (0.3618) | 0.8057 (0.0978) | 20 | 0.790 |
| 12 | Fast-growing evergreen broadleaved trees | 2.3643 (0.3310) | 0.6932 (0.0555) | 87 | 0.647 |
| 13 | Other evergreen broadleaved trees | 1.8980 (0.2141) | 0.7106 (0.0443) | 87 | 0.751 |

* Data of mean diameter at breast height ($D$, cm) and height ($H$, m) are from Luo et al. (2013). H-D curves are depicted

by using model H=$a$ Đ$^b$, where $a$ and $b$ are equation coefficients. S.E., standard error; $n$, sample size; and $R^2$, coefficient

5    of determination.

† To categorize tree species (group), the following factors are considered in decreasing order of significance: adequate

sample size (generally >20), similar phylogenetic relationship, similar ecophysiological characteristics, and similar

growth conditions.

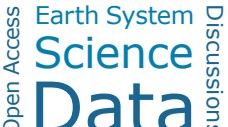


**Table A2: Biomass equation forms used in studies. W is tree biomass (kg) or its components; X is tree diameter (cm), height (m) or their combinations; *a*, *b*, *c*, *d*… are equation coefficients; log(W) is either the natural or the 10-base logarithmic transformation of the tree biomass data.**

| Category | Equation form | Number of equations | Category | Equation form | Number of equations |
|---|---|---|---|---|---|
| Power | $W=a\,X^b$ | 3812 | Exponential | $W=\exp(a+b\,X)$ | 4 |
| | $W=a\,X^b+c$ | 7 | | $W=\exp(a+b/X)$ | 1 |
| | $W=a\,(b+X)^c$ (c=2, 3, 4 or 5) | 43 | | $W=a\,\exp(b\,X^c)$ (c=1 or 2) | 29 |
| | $W=a\,X^b+c\,X^d$ | 1 | | $W=a\,\exp(b\,X)+c$ | 2 |
| | $W=a\,X_1^b\,X_2^c$ | 85 | | $W=a\,\exp(b+c\,X)$ | 1 |
| Linear | $W=a+b\,X$ | 253 | | $W=a\,\exp(b/X)$ | 3 |
| /polynomial | $W=a+b\,X+c\,X^2$ | 90 | | $W=a\,X^b\,\exp(c\,X)$ | 31 |
| | $W=a+b\,X+c\,X^2+d\,X^3$ | 6 | | $W=a\,b^X$ | 10 |
| | $W=a+b\,X^c$ (c=2, 3 or 4) | 82 | | $W=a\,b^X+c$ | 2 |
| | $W=a+b\,X^2+c\,X^4$ | 1 | | $W=a\,\exp[b\,(X_1^c+X_2^d)]$ | 2 |
| Log-linear | $W=a+b\,\log(X)$ | 16 | Hyperbolic | $W=X/(a+b\,X)$ | 17 |
| | $\log(W)=a+b\,X$ | 2 | | $W=a/(b+X)$ | 2 |
| | $\log(W)=a+b\,\log(X)$ | 1378 | | $W=1/[a+b\,\log(X)]$ | 1 |
| | $\log(W)=a+b\,\log(X)+c\,X$ | 26 | | $W=a\,b^{1/X}$ | 1 |
| | $\log(W)=a+b\,\log(X_1)+c\,\log(X_2)$ | 16 | | | |



**Appendix B. Reference list in ChinAllomeTree dataset**

Ai, X.R., and Shen, Z.K.: Growth and biomass of *Larix kaempferi* plantation, Journal of Hubei Institute for

Nationalities (Natural Sciences), 19(2), 20-22, 2001 (in Chinese).

Ai, X.R., and Zhou, G.L.: The biomass of Chinese fir plantation in the north boundary of middle subtropical zone,

Hubei Forestry Science and Technology, (2), 17-20, 1996 (in Chinese).

Ai, X.R., Shen, Z.K., and Yi, Y.M.: Effect of stand density on the biomass of *Pinus massoniana* plantation, Hubei

Forestry Science and Technology, (3), 16-18, 1998 (in Chinese).

Ai, X.R., Yao, L., Yi, Y.M., and Shen, Z.K.: Carbon storage of *Cryptomeria fortunei* plantation in Enshi Autonomous

Prefecture, Journal of Hubei Institute for Nationalities (Natural Sciences), 19(2), 20-22, 2011 (in Chinese).

An, H.P., Jin, X.L., and Yang, C.H.: Growth rhythm and biomass dynamics of major vegetation types in Banqiaohe

Watershed, Guizhou Forestry Science and Technology, 19(4), 20-34, 1991 (in Chinese).

Bai, Y.Q., and Zhan, H.Z.: The biomass of *Larix gmelinii* plantation, Forest Investigation Design, (1), 21-25, 1980 (in

Chinese).

Bao, C.S.: Nutrient cycling in a birch forest, in: Long-term Located Research on Forest Ecosystems, Volume 1, Zhou,

X.F., Wang, Y.H., Zhao, H.X., eds., Northeast Forestry University Press, Harbin, China, 217-227, 1991 (in

Chinese).

Bao, C.S., Bai, Y., Qing, M., Chen, G.W., Zhang, Q.L., and Wang, L.M.: Productivity and carbon storage of *Larix

gmelinii* natural forest, Journal of Inner Mongolia Agricultural University, 31(2), 77-82, 2010 (in Chinese).

Bao, X.C., Chen, L.Z., Chen, Q.L., Ren, J.K., Hu, Y.H., and Li, Y.: The biomass of planted oriental oak (*Quercus

variabilis*) forest, Acta Phytoecologica et Geobotanica Sinica, 8(4), 313-320, 1984 (in Chinese).

Cao, J.X., Wang, X.P., Tian, Y., Wen, Z.Y., and Zha, T.S.: Pattern of carbon allocation across three different stages of

stand development of a Chinese pine (*Pinus tabuliformis*) forest, Ecological Research, 27, 883-892, 2012.

Chai, B.F., Zhang, J.T., Qiu, Y., and Zheng, F.Y.: Aboveground biomass and productivity of *Larix

principis-rupprechtii* artificial forest in the west of Shanxi Province, Henan Science, 17(S), 68-71, 1999 (in

Chinese).

Chen, C.G.: Biomass equations of Korean pine plantation, Forest Investigation Design, (2), 19-23, 1981 (in Chinese).

Chen, C.G.: Biomass and productivity of tree layers in *Pinus armandii* forests, Qinling Mountains, Journal of

Northwestern College of Forestry, (1), 1-18, 1984 (in Chinese).

Chen, H.J.: Biomass and nutrient distribution in a Chinese-fir plantation chronosequence in Southwest Hunan, China,



Forest Ecology and Management, 105, 209-216, 1998.

Chen, Q.C.: Study on the Primary Productivity of an Evergreen Broadleaved Forest Ecosystem, Hangzhou University Press, Hangzhou, China, 1993 (in Chinese).

Chen, W.R.: Study on the dynamics of aboveground net productivity of *Alniphyllum fortunei* plantation, Journal of Fujian Forestry Science and Technology, 27(3), 31-34, 74, 2000 (in Chinese).

Chen, X.G.: The biomass and allometric equation of a 20-years-old *Cunninghamia lanceolata* plantation, Protection Forest Science and Technology, (4), 28-29, 40, 2007 (in Chinese).

Chen, B.H., and Chen, C.Y.: A preliminary study on the biomass and productivity of *Picea koraiensis* forests in the dunes, Scientia Silvae Sinicae, (4), 269-278, 1980 (in Chinese).

Chen, C.G., and Guo, X.F.: The biomass of broadleaved Korean pine forest, Forest Investigation Design, (2), 10-19, 6, 1984 (in Chinese).

Chen, C.G., and Peng, H.: Standing crops and productivity of the major forest types at Huoditang Forest Region of Qinling Mountains, Journal of Northwest Forestry College, 11(S), 92-102, 1996 (in Chinese).

Chen, C.G., and Zhu, J.F.: Manual on Biomass Equations of Major Tree Species in Northeast China, China Forestry Publishing House, Beijing, China, 1989 (in Chinese).

Chen, Z.S., and Fang, Q.: Nutrient content and the biomass of *Populus alba* var. *pyramidalis* plantation, Forest Research, 1(5), 535-540, 1988 (in Chinese).

Chen, B.H., Li, H.Q., and Liu, J.G.: Biomass of *Populus diversifolia* natural forest in middle reach of Tarim River, Xinjiang, Xinjiang Forestry Science and Technology, (3), 8-16, 1984a (in Chinese).

Chen, C.G., Gong, L.Q., Peng, H., and Liu, X.Z.: Biomass and productivity of *Quercus aliena* var. *acuteserrata* forests in Qinling Mountains, Journal of Northwest Forestry College, 11(S), 103-114, 1996 (in Chinese).

Chen, C.X., Yu, K.Y., Yang, Z.Q., Liao, X.L., You, H.C., and Chen, F.H.: The compatible stand biomass estimation model of *Pinus massoniana* stands, Journal of Sanming University, 27(4), 379-382, 2010 (in Chinese).

Chen, D.X., Li, Y.D., Luo, T.S., Lin, M.X., and Sun, Y.X.: Biomass and net primary productivity of *Podocarpus imbricatus* plantation in Jianfengling, Hainan Island, Forest Research, 17(5), 598-604, 2004 (in Chinese).

Chen, H., Ren, C.H., Zheng, L.P., Ruan, C.C., and Liao, Z.H.: Biomass equations of *Phoebe bournei* plantation, Journal of Fujian College of Forestry, 9(4), 411-417, 1989 (in Chinese).

Chen, L.N., Xiao, Y., Gai, Q., and Ji, W.X.: Preliminary study on the biomass of *Larix principis-rupprechtii* forest in Pangquangou Nature Reserve: Community structure, biomass, and net primary productivity, Journal of Shanxi Agricultural University, 11(3), 240-247, 1991 (in Chinese).

Chen, L.Z., Ren, J.K., Bao, X.C., Chen, Q.L., Hu, Y.H., Miao, Y.G., and Li, Y.: Community characteristics and

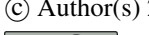



biomass of Chinese pine plantation in Xishan Region, Beijing, Acta Phytoecologica et Geobotanica Sinica, 8(3), 173-181, 1984b (in Chinese).

Chen, L.Z., Chen, Q.L., Bao, X.C., Ren, J.K., Miao, Y.G., and Hu, Y.H.: Study on Chinese arborvitae (*Platycladus orientalis*) forest and its biomass in Beijing, Acta Phytoecologica et Geobotanica Sinica, 10(1), 17-25, 1986a (in Chinese).

Chen, L.Z., Ren, J.K., Chen, Q.L., Hu, Y.H., Bao, X.C., and Miao, Y.G.: The biomass of *Robinia pseudoacacia* plantation in Xishan Region, Beijing, Acta Botanica Sinica, 28(2), 201-208, 1986b (in Chinese).

Chen, T., Wen, Y.G., Sun, Y.P., and Liang, H.W.: Preliminary study on the biomass and productivity of *Eucalyptus urophylla ×E. grandis* plantations with successive rotations, Guangxi Forestry Science, 34(1), 8-12, 2005 (in Chinese).

Chen, X.Y., Peng, Y.Y., and Kang, W.X.: Study on biomass and productivity of a subtropical evergreen broadleaved forest dominated by *Engelhardtia roxburghiana* and *Schima superba*, in: Long-term Located Research on Forest Ecosystem, Liu, X.Z., Kang, W.X., Chen, X.Y., Wen, S.Z., eds., China Forestry Publishing House, Beijing, China, 68-72, 1993a (in Chinese).

Chen, Z.H., Zhang, H.D., Wang, B.S., and Zhang, Z.Q.: The biomass and its allocation of an evergreen broadleaved forest in Heishiding, Guangdong, Acta Phytoecologica et Geobotanica Sinica, 17(4), 289-298, 1993b (in Chinese).

Chen, Z.J., Zhang, J.Z., Chen, J., Xia, Y.F., and Huang, S.Q.: Preliminary study on the aboveground biomass of *Pinus massoniana* air-seeding forest, East China Forest Management, 12(4), 51-53, 1998 (in Chinese).

Chen, Z.X., He, Y.J., Bai, F.M., Zhang, J.H., and Li, Z.H.: Effects of stand density on the biomass and productivity of *Pinus massoniana* air-seeding stands, Journal of Central South Forestry University, 21(1), 44-47, 2001 (in Chinese).

Cheng, Y.X., and Li, Z.X.: Preliminary study on the biomass of three *Larix gmelinii* forests, Inner Mongolia Forestry Investigation and Design, (4), 29-39, 1989 (in Chinese).

Cheng, T.R., Ma, Q.Y., Feng, Z.K., and Luo, X.: Study on forest biomass in Xiaolong Mountains, Gansu Province, Journal of Beijing Forestry University, 29(1), 31-36, 2007 (in Chinese).

Cheng, X.Q., Han, H.R., and Kang, F.F.: Biomass, carbon accumulation and their partitioning of a *Pinus tabuliformis* plantation ecosystem in Shanxi Province, China, Chinese Journal of Ecology, 31(10), 2455-2460, 2012 (in Chinese).

Cheng, Y., Hong, W., Wu, C.Z., and Qi, X.H.: Aboveground biomass and its productivity of *Schima superba* population, Chinese Journal of Applied and Environmental Biology, 15(3), 318-322, 2009 (in Chinese).

Cui, W., Mu, C.C., Lu, H.C., Bao, X., and Wang, B.: Effects of draining for forestation on carbon storage of wetland




ecosystem in Daxing'an Mountains of northeastern China, Journal of Beijing Forestry University, 35(5), 28-36, 2013 (in Chinese).

Dai, H.J., He, H.J., Zhao, X.H., Zhang, C.Y., Wang, J.S., and Yang, S.: Biomass allocation patterns and allometric models of two dominant tree species in broadleaved and Korean pine mixed forest, Chinese Journal of Applied and Environmental Biology, 19(4), 718-722, 2013 (in Chinese).

Dang, C.L., and Wu, Z.L.: Study on the biomass of *Pinus yunnanensis* forest. Acta Botanica Yunnanica, 13(1), 59-64, 1991 (in Chinese).

Dang, C.L., and Wu, Z.L.: Study on the biomass of a monsoon evergreen broadleaved forest dominated by *Castanopsis echidnocarpa*, Journal of Yunnan University (Natural Sciences), 14(2), 95-107, 1992 (in Chinese).

Dang, C.L., and Wu, Z.L.: Study on the biomass of *Castanopsis orthacantha* community, Journal of Yunnan University (Natural Sciences), 16(3), 195-199, 1994 (in Chinese).

Dang, C.L., Yu, H., and Li, Y.: Study on the relationship between ecological factors and overstory biomass in *Pinus yunnanensis* forest, Journal of Yunnan University (Natural Sciences), 14(2), 146-151, 1992 (in Chinese).

Dang, C.L., Wu, Z.L., and Zhang, Z.: Study on the biomass of *Cyclobalanopsis delavayi* community, Journal of Yunnan University (Natural Sciences), 16(3), 205-209, 1994a (in Chinese).

Dang, C.L., Wu, Z.L., Wang, C.Y., He, Z.R., and Gu, Z.F.: Study on the biomass and net primary productivity of *Abies georgei* community, Journal of Yunnan University (Natural Sciences), 16(3), 214-219, 1994b (in Chinese).

Dang, C.L., Wu, Z.L., and Zhang, Q.: Study on the biomass of the ravine tropical rain forest in Xishuangbanna, Acta Botanica Yunnanica, (S8), 123-128, 1997 (in Chinese).

Deng, S.J., Wang, K.P., and Gao, H.: Biological productivity and nutrient distribution in overmature *Cunninghamia lanceolata* plantation, Chinese Journal of Ecology, 7(1), 13-18, 1988 (in Chinese).

Deng, S.J., Liao, L.P., Wang, S.L., Gao, H., and Lin, B.: Bioproductivity of *Castanopsis hysrix - Cyclobalanopsis glauca - Machilus pauhoi* community in Huitong, Hunan, Chinese Journal of Applied Ecology, 11(5), 651-654, 2000 (in Chinese).

Di, D.S., Liao, H.Z., Zhang, C.N., and Chen, Z.Z.: Study on ecosystem productivity and tree growth of *Ormosia xylocarpa* plantations, Journal of Nanjing Forestry University, 15(3), 60-65, 1991 (in Chinese).

Ding, Z.F.: Study on Community Characteristics, Biomass and Productivity of Forests in Xiaokeng Watershed, Anhui Province, Thesis for Master's Degree, Anhui Agricultural University, Hefei, China, 62pp., 2005 (in Chinese).

Ding, B.Y., and Sun, J.H.: Study on biological productivity and nutrient cycling of artificial Korean pine forest ecosystem, Journal of Northeast Forestry University, 17(S), 1-98, 1989 (in Chinese).

Ding, G.J., Wang, P.C., and Yan, R.F.: Study on the biomass dynamics and modeling of Masson pine pulpwood stands,



Scientia Silvae Sinicae, 34(1), 33-41, 1998 (in Chinese).

Ding, Z.F., Jiang, C.W., Wang, X.J., and Wu, Z.M.: Forest biomass and productivity of main evergreen communities in Xiaokeng Watershed, Anhui Province, Journal of Nanjing Forestry University (Natural Science Edition), 33(2), 129-133, 2009 (in Chinese).

Dong, L.H., Li, F.R., and Jia, W.W.: Development of tree biomass model for *Pinus koraiensis* plantation, Journal of Beijing Forestry University, 34(6), 16-22, 2012 (in Chinese).

Dong, L.H., Li, F.R., and Jia, W.W.: Effects of tree competition on the biomass and biomass models of *Pinus koraiensis* plantation, Journal of Beijing Forestry University, 35(6), 15-22, 2013 (in Chinese).

Du, W.Z.: The individual biomass of *Pinus tabuliformis* plantation in Xiaolong Mountains, Gansu Province, Gansu Science and Technology, 28(23), 153-154, 70, 2012 (in Chinese).

Du, G.J., Hong, L.X., and Yao, G.X.: Study on the aboveground biomass of major evergreen broadleaved secondary forests, northwestern Zhejiang Province, Journal of Zhejiang Forestry Science and Technology, 7(5), 5-12, 1987 (in Chinese).

Du, H., Song, T.Q., Zeng, F.P., Wen, Y.G., and Peng, W.X.: Biomass and its allocation in *Pinus massoniana* plantation at different stand ages in eastern Guangxi Province, Acta Botanica Boreali-Occidentalia Sinica, 33(2), 394-400, 2013 (in Chinese).

Fan, J.J.: Biomass model of *Larix olgensis* plantation in Dongzhelenghe Nature Reserve, Shanxi Forestry Science and Technology, 41(4), 26-28, 2012 (in Chinese).

Fang, W., and Wang, G.Q.: Biomass and productivity of farmland shelterbelt network, Forest Science and Technology, (6), 11-13, 1989.

Fan, S.H., Yu, X.T., Chen, Z.T., Liu, P.S., He, Z.Y., Shen, G.F., and Sheng, W.T.: Biomass of Chinese fir plantation under different site conditions and stand ages: (1) Stand biomass accumulation, Forest Research, 9(S), 78-85, 1996 (in Chinese).

Fan, S.H., Liu, G.L., Zhang, Q., Feng, H.X., Zong, Y.C., and Ren, H.Q.: Biomass and productivity of *Populus xiaohei* plantation on sandy land in north China, Forest Research, 23(1), 71-76, 2010 (in Chinese).

Fan, Z.F., An, Y.T., and Zhao, X.H.: Biomass and productivity of *Pinus tabuliformis* plantation, Journal of Beijing Forestry University, 19(S2), 93-98, 1997 (in Chinese).

Fang, J.P., and Xiang, W.H.: Biomass and its distribution of a primeval *Abies georgei* var. *smithii* forest in Sejila Mountain in Tibet Plateau, Scientia Silvae Sinicae, 44(5), 17-23, 2008 (in Chinese).

Fang, X., and Tian, D.L.: Dynamics of carbon stock and sequestration in Chinese fir plantation, Guihaia, 26(5), 516-522, 2006 (in Chinese).



Fang, C.L., Zhu, X.W., and Zhang, H.C.: Preliminary study on the biomass and productivity of *Picea crassifolia* natural secondary forests, Journal of Qinghai University, 9(1), 71-77, 1991 (in Chinese).

Fang, H.B., Tian, D.L., and Kang, W.X.: Biomass dynamics of a thinned Chinese fir plantation ecosystem, Journal of Central South Forestry University, 19(1), 16-19, 1999 (in Chinese).

Fang, S.Z., Cai, S.Z., Chen, J.H., Yang, B., Ding, X.F., and Song, Z.Y.: Biomass production and pulp-making performance of *Metasequoia glyptostroboides* plantation, Journal of Nanjing Forestry University, 19(4), 51-56, 1995 (in Chinese).

Feng, Z.W.: Study on the biomass of *Cunninghamia lanceolata* plantations, in: Comprehensive Investigation Reports of Taoyuan County, Taoyuan Agricultural Experiment Station of Chinese Academy of Sciences, eds., Hunan Science and Technology Press, Changsha, China, 322-333, 1980 (in Chinese).

Feng, Y.Z.: Man-made Community, Yunnan Science and Technology Press, Kunming, China, 48-55, 2007 (in Chinese).

Feng, L., and Yang, Y.G.: The biomass of natural secondary forests of *Pinus tabuliformis*, *Betula platyphylla* and *Populus davidiana* in Inner Mongolia Region, Journal of Inner Mongolia Forestry College, (3), 1-17, 1981 (in Chinese).

Feng, L., and Yang, Y.G.: Biomass and productivity of three *Larix gmelinii* virgin forests, Scientia Silvae Sinicae, 21(1), 86-92, 1985 (in Chinese).

Feng, Z.L., Zheng, Z., Zhang, J.H., Cao, M., Sha, L.Q., and Deng, J.W.: Biomass and its distribution of a tropical wet seasonal rainforest in Xishuangbanna, Acta Phytoecologica Sinica, 22(6), 481-488, 1998 (in Chinese).

Feng, Z.L., Tang, J.W., Zheng, Z., Song, Q.S., Cao, M., Zhang, J.H., and Xie, J.W.: Biomass dynamics of the pioneer *Trema orientalis* community in the early stages of secondary succession of tropical forest in Xishuangbanna, Chinese Journal of Ecology, 18(5), 1-6, 1999 (in Chinese).

Feng, Z.L., Zheng, Z., Tang, J.W., Song, Q.S., and Zhang, J.H.: Biomass of tropical secondary *Mallotus paniculatus* forest in Xishuangbanna, Chinese Journal of Ecology, 24(3), 238-242, 2005 (in Chinese).

Feng, Z.W., Chen, C.Y., Zhang, J.W., Wang, K.P., and Zhao, J.L.: Biological productivity of two forest communities in Huitong County of Hunan Province, Acta Phytoecologica et Geobotanica Sinica, 6(4), 257-267, 1982 (in Chinese).

Feng, Z.W., Zhang, J.W., Chen, C.Y., Wang, K.P., Zhao, J.L., Zeng, S.Y., and Ma, J.X.: Biological productivity and nutrient distribution in artificial *Michelia macclurei* stand, Journal of Northeastern Forestry Institute, 11(2), 13-20, 1983 (in Chinese).

Feng, Z.W., Chen, C.Y., Zhang, J.W., Zhao, J.L., Wang, K.P., and Zeng, S.Y.: The biological productivity of Chinese fir stands at different zones, Acta Phytoecologica et Geobotanica Sinica, 8(2), 93-100, 1984 (in Chinese).



Fu, Z.J.: Study on synecological features and biomass of *Larix chinensis* forest in Taibai Mountain, Journal of Hanzhong Teachers College (Natural Sciences), (2), 69-72, 1994 (in Chinese).

Gao, Z.H.: Preliminary study on the biomass of *Cunninghamia lanceolata* plantations in different management regimes, Journal of Zhejiang Forestry Science and Technology, (2), 25-30, 1986 (in Chinese).

Gao, Z.H., and Li, G.L.: Preliminary study on the growth and biomass of *Metasequoia glyptostroboides* shelter forest, Forest Science and Technology, (3), 10-12, 1987 (in Chinese).

Gao, C.J., Tang, G.Y., Sun, Y.Y., Zhang, C.H., Xie, Q.H., and Li, K.: Biomass and allocation of young *Azadirachta indica* and *Acacia auriculiformis* for different restoration patterns in dry-hot valley, Journal of Zhejiang A&F University, 29(4), 482-490, 2012 (in Chinese).

Gao, H.Y., Zhou, G.Y., Zhou, Z.P., Zhao, H.B., and Qiu, Z.J., Aboveground biomass of 27-year-old *Cunninghamia lanceolata* plantation in Tianjingshan Forest Farm, Guangdong Province, Guangdong Forestry Science and Technology, 29(4), 1-6, 2013 (in Chinese).

Gao, H.Z., You, L.Q., and Wang, C.: Individual biomass and productivity of *Pinus tabuliformis* plantation in Yanshan Mountain, Journal of Hebei Forestry Science and Technology, (4), 7-9, 2009 (in Chinese).

Gao, S.C., Tian, D.L., Yan, W.D., Zhu, F., Fang, X., and Liang, X.C., Pattern characteristics of stand biomass of urban forest in Changsha City, Journal of Central South University of Forestry and Technology, 30(12), 56-65, 2010 (in Chinese).

Gao, X.R., Zhao, H., Yang, H.Q., Ling, X.M., and Fan, W.: Biomass and carbon storage of *Rhus typhina* in hilly area of Taihang Mountain, Journal of Central South University of Forestry and Technology, 32(12), 172-175, 2012b (in Chinese).

Gao, Y.P., Pan, M.L., Ding, F.J., Zhou, F.J., and Wu, P.: Biomass and net productivity of natural secondary forests of *Betula luminifera* in west Guizhou, Journal of Central South University of Forestry and Technology, 32(4), 55-60, 2012c (in Chinese).

Gao, Z.H., Jiang, G.H., Xing, A.J., and Yu, M.R.: The biomass of *Metasequoia glyptostroboides* plantation in Zhebei Plain, Acta Phytoecologica et Geobotanica Sinica, 16(1), 64-71, 1992 (in Chinese).

Gu, Y.K., Chen, B.G., and Feng, Y.H.: An investigation on the aboveground biomass and its productivity of artificial *Cunninghamia lanceolata* stands in Xijiang Region, Guangdong Province, Journal of South China Agricultural University, 8(1), 41-50, 1987 (in Chinese).

Guan, D.S.: The biomass and productivity of four stands in the forest area of Liuxihe Reservoir, Ecological Science, (2), 45-52, 1986 (in Chinese).

Guan, D.Y., and Huang, G.Q.: The biomass and its predictive models of *Castanopsis fissa* natural forest, Journal of




Fujian Forestry Science and Technology, 27(2), 34-36, 2000.

Guan, H.S., and Liu, Y.L.: Study on the biomass of poplar plantations in the middle region of "One River and Two Streams" Watershed, Tibet, Forest Science and Technology, (9), 20-22, 32, 1993 (in Chinese).

Guo, L.Q., and Xiao, Y.: The biomass table of *Larix principis-rupprechtii* natural forest, Forest Resources Management, (5), 36-39, 1989 (in Chinese).

Guo, X.Y., Cai, T., Duan, X.W., Han, Y.J., Huang, D., and Da, L.J.: Carbon storage and distribution pattern in main economic fruit forest ecosystems in Shanghai, East China, Chinese Journal of Ecology, 32(11), 2881-2885, 2013 (in Chinese).

Han, M.Z.: The biomass and production of *Larix gmelinii* plantation, Acta Agriculturae Boreali-Sinica, 2(4), 134-138, 1987 (in Chinese).

Han, M.Z.: The biomass and net primary productivity of a Dahurian larch-birch forest ecosystem, in: Long-term Research on China's Forest Ecosystems, Department of Science and Technology of Ministry of Forestry, eds., Northeast Forestry University Press, Harbin, China, 451-458, 1994 (in Chinese).

Han, Y.Z., and Liang, S.F.: Study on the tree root system and its biomass of *Larix principis-rupprechtii* plantation, Shanxi Forestry Science and Technology, (3), 36-40, 1997 (in Chinese).

Han, F.Y., Zhou, Q.Y., Chen, S.X., Chen, W.P., Li, T.H., Wu, Z.H., and Jian, M.: The biomass and energy of two different aged Eucalyptus stands, Forest Research, 23(5), 690-696, 2010 (in Chinese).

Han, Y.Z., Li, Y.E., Liang, S.F., and Li, H.Y.: Study on the tree biomass of *Larix principis-rupprechtii* plantation, Journal of Shanxi Agricultural University, 17(3), 278-283, 1997 (in Chinese).

He, R.B.: The biomass model of *Metasequoia glyptostroboides* plantation, Forestry Prospect and Design, (2), 46-49, 1997 (in Chinese).

He, F., Wang, Y.Q., Tan, X.F., and Wang, C.N.: The biomass and nutrient cycle in *Vernicia fordii* plantations, Non-wood Forest Research, 8(2), 6-20, 1990 (in Chinese).

He, G.P., Chen, Y.T., Hu, B.T., Feng, J.W., Liu, H.T., and Cai, H.M.: Study on the biomass and soil fertility of pure and mixed stands of *Cunninghamia lanceolata*, *Liriodendron chinense* and *Sassafras tzumu*, Forest Research, 14(5), 540-547, 2001 (in Chinese).

He, H.Z., Huang, L.H., Duan, X., and He, R.K.: Study on the biomass in main afforestation tree species of the second-ring forest belt of Guiyang City, Guizhou Science, 25(3), 33-39, 2007 (in Chinese).

He, H.Z., Song, J.X., Liu, Y.Y., Zhang, Y.W., and Huang, L.H.: The biomass and allocation of Chinese fir forest in southeast Guizhou, Guangdong Agricultural Sciences, (21), 58-60, 2013 (in Chinese).

He, Y.J., Qin. L., Li. Z.Y., Shao. M.X., Liang. X.Y., and Tan, L.: Carbon storage capacity of a *Betula alnoides* stand



and a mixed *Betula alnoides* × *Castanopsis hystrix* stand in southern subtropical China: A comparison study, Acta Ecologica Sinica, 32(23), 7586-7594, 2012 (in Chinese).

He, Z.M., Fan, S.H., Lu, J.M., Yang, X.J., and Weng, X.Q.: Effects of site management treatments on growth of 6-year-old, second rotation Chinese fir plantations, Scientia Silvae Sinicae, 42(11), 47-51, 2006 (in Chinese).

Hong, T., Wu, C.Z., Lin, Y.M., Chen, C., Li, J., and Lin, H.: Biomass characteristics in the arbor layer of *Vernicia montana* plantation, Journal of Mountain Science, 30(6), 648-654, 2012 (in Chinese).

Hong, Y.C., Xu, W.Q., Ye, G.F., and Zhang, L.H.: Model for estimating biomass of *Casuarina equisetifolia* planation in coastal region of the southeastern China, Journal of Zhejiang Forestry Science and Technology, 30(4), 66-69, 2010 (in Chinese).

Hu, Y.H., and Pang, Q.L.: The biomass and distribution pattern of *Cunninghamia lanceolata* plantation in western Hubei, Hubei Forestry Science and Technology, (3), 6-9, 2012 (in Chinese).

Hu, D.L., Li, Z.H., and Xie, X.D.: The biomass and productivity of *Pinus taiwanensis* plantation, Journal of Central South Forestry University, 18(1), 60-64, 1998 (in Chinese).

Hu, J.R., Huang, R.K., Zeng, H.D., Li, P.Y., and Wang, Z.X.: Individual biomass model of mature *Pinus massoniana* plantation, Forestry Prospect and Design, (1), 9-12, 2011 (in Chinese).

Hu, S.S., Zhang, Y.T., Li, J.M., Lu, J.J., Li, X., Wang, Q.J., and Wang, X.K.: Biomass distribution of *Populus alba* var. *pyramidalis* plantation, Xinjiang Agricultural Sciences, 49(6), 1059-1065, 2012 (in Chinese).

Huang, D.C.: The biomass of *Cryptomeria fortunei* plantation, Journal of Southwest Forestry College, (1), 23-28, 1986 (in Chinese).

Huang, C.B., and Liang, H.W.: Growth rhythm and biomass of young *Pinus massoniana* plantation in the southeast Guangxi, Journal of Guangxi Academy of Sciences, 14(1), 22-27, 1998 (in Chinese).

Huang, Z.Z., and Bi, J.: Study on the biomass of *Robinia pseudoacacia* stands in Taihang Mountains, Journal of Hebei Forestry Science and Technology, (2), 48-52, 1992 (in Chinese).

Huang, C.H., Zhou, G.Y., Zhao, H.B., Zhou, Z.P., and Qiu, Z.J.: Root system biomass of mature *Cunninghamia lanceolata* plantation in Tianjingshan Forest Farm, Guangdong Province, Journal of Central South University of Forestry and Technology, 33(9), 80-86, 2013 (in Chinese).

Huang, L.M., Xue, L., Wang, X.E., Xie, T.F., Ren, X.R., and Cao, H.: Growth and biomass allocation of young *Acacia auriculiformis* stands under different densities, Journal of South China Agricultural University, 29(3), 52-55, 2008 (in Chinese).

Huang, Q., Li, Y.D., Lai, J.Z., and Peng, G.J.: The biomass of tropical mountain rain forest in Limushan Forest Park, Hainan Island, Acta Phytoecologica et Geobotanica Sinica, 15(3), 197-206, 1991 (in Chinese).





Huang, S.S., Han, H.R., and Ma, Q.Y.: Study on the biomass of broadleaved tree species in Taiyue Forest Region, in: Forum of Forest Ecology, Volume 1, Zhu, Z.H., Luo, J.C., eds., China Agricultural Science and Technology Press, Beijing, China, 96-102, 1999 (in Chinese).

Huang, T., Zhong, Q.P., and Peng, X.Y.: Study on the biomass and productivity of *Liriodendron chinense* plantation, Jiangxi Forestry Science and Technology, (5), 4-9, 2000 (in Chinese).

Huang, X.S., Wu, C.Z., Hong, W., Li, Z.K., and Cheng, Z.P.: The relationship between stand density and biomass of two rotation Chinese fir plantations, Journal of Fujian College of Forestry, 31(2), 102-105, 2011 (in Chinese).

Huang, Y.Q., Chen, S.Y., and Wu, X.F.: Models for estimating biomass of *Eucalyptus urophylla* plantation, Journal of Anhui Agricultural University, 28(1), 44-48, 2001 (in Chinese).

Hui, G.Y., Tong, S.Z., Liu, J.F., and Luo, Y.W.: Effects of afforestation density on the biomass of young *Cunninghamia lanceolata* plantation, Forest Research, 1(4), 413-417, 1988 (in Chinese).

Hui, G.Y., Luo, Y.W., and Zhang, X.L.: The productivity of Chinese fir (*Cunninghamia lanceolata*) plantation at hilly area in Dagang Mountain, Jiangxi Province, Scientia Silvae Sinicae, 25(6), 564-569, 1989 (in Chinese).

Huo, C.F., You, W.Z., Zhang, H.D., Yan, T.W., Wei, W.J., Zhao, G., Guo, J.S., and Xing, Z.K.: Biomass and net primary productivity of *Quercus mongolica* natural secondary forest in Bingla Mountain, Liaoning Province, Journal of Liaoning Forestry Science and Technology, (4), 4-6, 11, 2011 (in Chinese).

Ji, Y.H., Zhang, J.L., and Kang, L.X.: A study on biomass equations for *Metasequoia glyptostroboides* shelterbelt in the coastal agroforestry, Journal of Jiangsu Forestry Science and Technology, 24(2), 1-5, 1997 (in Chinese).

Jia, Y., and Qi, L.X.: The relationship between *Lophodermium maximum* and the biomass of *Pinus koraiensis* plantation, Journal of Northeast Forestry University, 16(5), 7-14, 1988 (in Chinese).

Jia, Y., and Zhang, F.: The biomass of *Pinus koraiensis* plantation in Caohekou Forest Region, Liaoning Province, Journal of Liaoning Forestry Science and Technology, (5), 18-23, 1985 (in Chinese).

Jia, K.X., Zheng, Z., and Zhang, Y.P.: Changes in the aboveground biomass of rubber plantations along an elevation gradient in Xishuangbanna, Chinese Journal of Ecology, 25(9), 1028-1032, 2006 (in Chinese).

Jia, W.W., Jiang, S.W., and Li, F.R.: Individual biomass of *Pinus sylvestris* var. *mongolica* plantation in the eastern Heilongjiang, Journal of Liaoning Forestry Science and Technology, (3), 5-10, 2008 (in Chinese).

Jia, Y.L., Xu, Z.Q., Ji, X.L., Xu, X.H., and Huang, X.R.: Biological carbon storage of a plantation and natural secondary forest in the north region of Yanshan Mountain, Journal of Natural Resources, 27(7), 1241-1251, 2012 (in Chinese).

Jiang, H.: A study on the biomass and productivity of *Picea purpurea* natural forest, Acta Phytoecologica et Geobotanica Sinica, 10(2), 146-152, 1986 (in Chinese).




Jiang, H.: Study on the biomass of *Quercus liaotungensis* and *Betula dahurica* forest in Dongling Mountain, in: Studies on Structures and Functions of Warm-temperate Forest Ecosystems, Chen, L.Z., Huang, J.H., eds., Science Press, Beijing, China, 104-115, 1997 (in Chinese).

Jiang, Z.L., and Zhao, S.: Study on the biomass of Loblolly pine plantation, in: Proceedings of forest ecosystems on Xiashu Ecological Station, Jiang, Z.L., eds., China Forestry Publishing House, Beijing, China, 10-15, 1992 (in Chinese).

Jiang, H., and Zhu, J.J.: Study on the biomass and productivity of *Picea asperata* natural forest, Journal of Sichuan Forestry Science and Technology, 7(2), 5-13, 1986 (in Chinese).

Jiang, B., Yuan, W.G., Zhu, G.Q., Shi, J.T., Zhao, P.F., and Xu, Z.W.: A preliminary study on the biomass and productivity of *Pinus massoniana*, *P. elliottii* and *P. taeda* plantations, Journal of Zhejiang Forestry Science and Technology, 12(5), 1-8, 22, 1992 (in Chinese).

Jiang, J.P., Yang, X., and Li, R.X.: Net productivity and organic matter return of *Paulownia* plantation ecosystem, Acta Agriculturae Universitatis Henanensis, 23(4), 327-337, 1989 (in Chinese).

Jiang, P., Dong, S.G., Sui, Y.L., Wang, J.Y., and Wang, G.Z.: Study on biomass model of *Larix principis-rupprechtii* stands in Beigou Forest Farm, Journal of Central South University of Forestry and Technology, 33(7), 131-135, 2013 (in Chinese).

Jiang, T., Zhao, M., Zhang, S.Z., Yuan, M.L., and Huang, X.R.: Individual biomass and its allocation pattern of *Pinus tabuliformis* in Hebei, Hebei Journal of Forestry and Orchard Research, 27(3), 239-244, 2012 (in Chinese).

Jiang, Y.X., Xu, D.Y., and Nie, D.P.: Study on the Productivity and Nutrient Cycling of Chinese Fir Plantation, China Forestry Publishing House, Beijing, China, 78-79, 1995 (in Chinese).

Jiang, Z.H., Fan, S.H., Feng, H.X., Zhang, Q., Liu, G.L., and Zong, Y.C.: Biomass and distribution pattern of *Populus xiaohei* plantation in sandy land of north China, Scientia Silvae Sinicae, 43(11), 15-20, 2007 (in Chinese).

Jiao, S.R.: A preliminary study of the biomass and nutrient element distribution in *Pinus sylvestris* var. *mongolica* plantations in Zhanggutai Region, Liaoning Province, Acta Phytoecologica et Geobotanica Sinica, 9(4), 257-265, 1985 (in Chinese).

Jie, J.L., Zhan, Y.S., Huang, W.C., Long, W., Luo, Y.C., Hu, H.Y., and Xie, Z.R.: Study on the biomass of *Pinus elliottii* stand near Jinggangshan Line of Jing-Jiu Railway, Jiangxi Forestry Science and Technology, (2), 17-20, 2002 (in Chinese).

Jin, A.L.: Study on the biomass of main tree species in Bayingzhuang Forest Farm, Hebei Province, Thesis for Master's Degree, Beijing Forestry University, Beijing, China, 53pp., 2012 (in Chinese).

Jin, A.L., Rao, L.Y., Li, J., and Zhang, T.: The biomass of *Larix principis-rupprechtii* plantation, Guangdong




Agricultural Sciences, (12), 165-168, 2012 (in Chinese).

Jin, Z.Y., Jia, W.W., and Liu, W.: Biomass model for *Larix olgensis* plantation, Bulletin of Botanical Research, 30(6), 747-752, 2010 (in Chinese).

Jing, Y., Lu, J.M., and Xiao, H.S.: Effect of treatment regimens on the growth of *Pinus massoniana* regenerated stands, China Forestry Science and Technology, 18(1), 24-26, 2004 (in Chinese).

Kong, F.B., and Fang, H.: Comparative study on the biomass of *Pinus taeda* plantations with different densities and ages, Forestry Science and Technology, 28(3), 6-9, 2003 (in Chinese).

Kong, X.R., and Liu, Z.G.: The biomass and productivity of tree layer of *Cunninghamia lanceolata* plantation, Journal of Guangxi Agricultural College, (2), 29-40, 1983 (in Chinese).

Kang, B., Liu, S.R., Zhang, G.J., Chang, J.G., Wen, Y.G., Ma, J.M., and Hao, W.F.: Carbon accumulation and distribution in *Pinus massoniana* and *Cunninghamia lanceolata* mixed forest ecosystem in Daqingshan, Guangxi of China, Acta Ecologica Sinica, 26(5), 1320-1329, 2006 (in Chinese).

Li, B.T.: Preliminary study on biomass investigation method of Chinese fir plantation, Forest Resources Management, (6), 57-60, 1988 (in Chinese).

Li, H.L.: Carbon storage and carbon budget of poplar-crop intercropping ecosystem in the agricultural region of northern Jiangsu Plain, PhD Dissertation, Nanjing Forestry University, Nanjing, China, 125pp., 2010a (in Chinese).

Li, Z.: Study on the biomass of *Platyclatdus orientalis* plantation in Xuzhou City, Thesis for Master's Degree, Nanjing Forestry University, Nanjing, China, 36pp., 2010b (in Chinese).

Li, Z.H.: Effect of thinning on *Cunninghamia lanceolata* plantation in the eastern Hunan Province, Scientia Silvae Sinicae, 36(S1), 131-136, 2000 (in Chinese).

Li, S.L., and Hou, J.Z.: The biomass of *Populus tomentosa* clonal plantation, Acta Agriculturae Universitatis Henanensis, 29(2), 134-140, 1995 (in Chinese).

Li, B.B., Jian, W.H., Qin, Y., Zhang, Y.Z., and Wang, Z.B.: Relationship between stand density and the biomass of young *Larix principis-rupprechtii* plantation, Hebei Journal of Forestry and Orchard Research, 24(3), 244-247, 2009a (in Chinese).

Li, C.Y., Zha, T.S., Liu, J.L., and Jia, X.: Carbon and nitrogen distribution across a chronosequence of secondary lacebark pine in China, The Forestry Chronicle, 89(2), 192-198, 2013a.

Li, D.L., Jiang, P., and Wang, Y.F.: Individual biomass and productivity of *Larix principis-rupprechtii* plantation in Yanshan Mountain, Hebei Journal of Forestry and Orchard Research, 26(4), 334-339, 2011a (in Chinese).

Li, G., Li, Y.G., Liu, M.Z., and Jiang, G.M.: Vegetation biomass and net primary production of sparse forest grassland

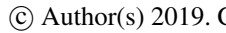



in Hunshandake Sandland, Science and Technology Review, 29(25), 30-37, 2011b (in Chinese).

Li, G.X., Meng, G.T., Fang, X.J., Lang, N.J., Yuan, C.M., and Wen, S.L.: Community characteristics and the biomass of *Alnus cremastogyne* plantation in central Yunnan Plateau, Journal of Zhejiang Forestry College, 23(4), 362-366, 2006a (in Chinese).

Li, H.L., Chen, L.B., Fang, S.Z., and Sun, Q.X.: Comparison of carbon storage and distribution in different poplar-crop intercropping patterns, Scientia Silvae Sinicae, 45(11), 9-14, 2009b (in Chinese).

Li, J., Li, H.M., Chen, G.X., and Shi, J.X.: A study on the biomass of young *Rhus chinensis* and *R. punjabensis* plantation, Journal of Jishou University (Natural Science Edition), 21(1), 1-5, 2000 (in Chinese).

Li, J.H., Li, C.J., and Peng, S.K.: Study on the biomass expansion factor of poplar plantation, Journal of Nanjing Forestry University (Natural Sciences Edition), 31(4), 37-40, 2007a (in Chinese).

Li, J.L., Liang, S.C., and Cheng, S.Z.: A preliminary study on the community characteristics of *Keteleeria davidiana* var. *chien-peii* forest in Guizhou, Guizhou Science, 11(4), 7-11, 1993 (in Chinese).

Li, J.L., Liang, S.C., and Cheng, S.Z.: Preliminary study on the biomass of *Keteleeria davidiana* var. *chien-peii* population in Qingyan Town, Guizhou Province, Journal of Guizhou Normal University (Natural Sciences), 15(1), 7-12, 1997 (in Chinese).

Li, J.Z., Qin, W.M., Qin, Y., Qin, J., Tang, K., and Duan, W.W.: The biomass and productivity of *Manglietia glauca* plantation, Journal of Fujian Forestry Science and Technology, 38(1), 1-5, 2011c (in Chinese).

Li, N., Xu, W.B., Lai, J.S., Yang, B., Lin, D.M., and Ma, K.P.: The coarse root biomass of eight common tree species in subtropical evergreen forest, Chinese Science Bulletin, 58(4), 329-335, 2013b (in Chinese).

Li, S.C., Liu, X.L., Liu, X.Q., and Shi, Y.H.: The biomass and hydrological effects of *Pinus massoniana* plantation, Central South Forest Inventory and Planning, (3), 26-29, 1995 (in Chinese).

Li, S.H., Shi, J.N., and Lei, P.: The biomass and its vertical distribution of the second-generation Chinese fir plantation, Journal of Sichuan Forestry Science and Technology, 28(1), 64-67, 2007b (in Chinese).

Li, W.B., Bao, W.K., He, B.H., Wu, W.Y., and Li, F.L.: Biomass distribution and its influencing factors of *Pinus tabuliformis* plantations in the Dagou Valley of the upper Minjiang River, Journal of Mountain Science, 25(2), 236-244, 2007c (in Chinese).

Li, X.C., Huang, L.B., Gui, G.R., Li, Z.J., Fan, K.S., Lu, H.L., Du, X.X., Ma, W.M., Yang, J.M., and Li, Y.L.: Effects of initially planted density on young Chinese fir plantation for construction timber, Journal of Jiangsu Forestry Science and Technology, 23(3), 1-6, 1996a (in Chinese).

Li, X.G., Xu, J.Y., and Zhai, C.Y.: Study on the biomass of *Gordonia acumenata* community in Jinyun Mountains, in: Ecological Research on Evergreen Broadleaved Forests, Zhong, Z.C., eds., Southwest Normal University Press,




Chongqing, China, 234-250, 1988 (in Chinese).

Li, X.R., Liu, Q.J., Chen, Y.R., Hu, L.L., and Yang, F.T.: Aboveground biomass of three coniferous plantations in Qianyanzhou Research Station, Chinese Journal of Applied Ecology, 17(8), 1382-1388, 2006b (in Chinese).

Li, Y., Zhang, J.G., Duan, A.G., and Xiang, C.W.: Selection of biomass estimation models for Chinese fir plantation, Chinese Journal of Applied Ecology, 21(12), 3036-3046, 2010a (in Chinese).

Li, Y.S., Shi, M.H., and Kang, G.L.: The biomass and water-holding capacity of young *Pinus tabuliformis* plantation, Shanxi Forestry Science and Technology, (4), 35-38, 1994 (in Chinese).

Li, Z., Wang, Z.Y., Wang, Y., Guan, Q.W., Wei, W., Dong, P., and Zhang, H.N.: Effect of thinning on the biomass of *Platyclatdus orientalis* plantation, China Forestry Science and Technology, 24(1), 68-71, 2010b (in Chinese).

Li, Z., Zhou, W., Guan, Q.W., Wei, W., Dong, P., and Zhang, H.N.: Biomass and its influencing factors of *Platyclatdus orientalis* plantation in the limestone mountains of Xuzhou City, Journal of Anhui Agricultural University, 37(4), 669-674, 2010c (in Chinese).

Li, Z.H., He, L.X., Zhou, Y.P., and Zhou, Q.K.: The biomass and productivity of *Metasequoia glyphostrobodes* plantation, Journal of Central South Forestry University, 16(2), 47-51, 1996b (in Chinese).

Li, Z.L., Zhang, Y.L., Zhang, K., Ma, H.Z., Jiang, Z.H., Tong, G.S., Zhu, T.F., Li, Y.K., Wang, J.M., Li, X.R., and Zhao, C.G.: Dry matter production and its distribution of black locust (*Robinia pseudoacacia*) plantations in eastern Henan Province, Acta Agriculturae Universitatis Henanensis, 19(4), 382-392, 1985 (in Chinese).

Liang, H.W., Wen, Y.G., Wu, G.X., Huang, X.Z., Zhou, G.F., and Chen, D.L.: Effects of successive rotations on growth and productivity of *Eucalyptus urophylla × E. grandis* short-rotation plantation, Journal of Fujian Forestry Science and Technology, 35(3): 14-18, 2008 (in Chinese).

Liang, J.P., Zhang, B.X., Yang, H.B., Gu, Y.B., and Zhang, Y.L.: Study on tree biomass of *Pinus tabuliformis* plantation, Journal of Shanxi Agricultural University, 20(4), 339-341, 2000 (in Chinese).

Liang, K.N., Zhou, W.L., and Li, Y.Q.: Study on 6-year-old fertilizer trial of *Eucalyptus urophylla* plantation, Forest Research, 15(6), 644-653, 2002 (in Chinese).

Liang, K.N., Zhou, W.L., and Li, Y.Q.: Effects of fertilization on biomass and nutrient contents of young *Eucalyptus urophylla* cl. MLA plantation, Forest Research, 17(3), 327-333, 2004 (in Chinese).

Liang, N., Wang, W.B., and Tian, K.: Biomass distribution characteristics of 4- and 13-year-old *Betula alnoides* plantations, Journal of West China Forestry Science, 35(4), 188-192, 2006 (in Chinese).

Liang, Y.X., Wei, Z.M., Yu, G.C., Qin. J., and Chen, J.C.: The biomass and productivity of *Michelia macclurei* plantation in southeast Guangxi, China Forestry Science and Technology, 24(5), 45-49, 2010 (in Chinese).

Liao, H.Z., Zheng, Y.M., Zhang, C.N., and Chen, D.Y.: Study on the biomass of *Cinnamomum camphora* plantation,



Forest Science and Technology, (9), 15-18, 1986 (in Chinese).

Liao, H.Z., Zhang, C.N., and Chen, D.Y.: Study on the biomass of *Phoebe bournei* plantations, Journal of Fujian College of Forestry, 8(3), 252-257, 1988 (in Chinese).

Liao, H.Z., Di, D.S., and Zhang, C.N.: Study on the productivity of *Ormosia hosiei* planation, Forest Science and Technology, (10), 5-9, 1992 (in Chinese).

Lin, W.J.: Study on the productivity and nutrient elements cycling of *Cunninghamia lanceolata* plantation in mid-mountain regions of Laoshan Forest Farm, Tianlin County, Journal of Guangxi Agricultural College, 10(4), 27-39, 1991 (in Chinese).

Lin, W.F., and Lin, D.X.: Biomass and productivity of *Pinus massoniana* plantation, Journal of Heilongjiang Vocational Institute of Ecological Engineering, 23(5), 19-21, 2010 (in Chinese).

Lin, K.M., Zheng, Y.S., Huang, Z.Q., and Wu, Z.X.: Study on the models for biomass of young Chinese fir and Masson pine plantation, Journal of Fujian College of Forestry, 13(4), 351-356, 1993 (in Chinese).

Lin, S.M., Xu, T.G., and Zhou, G.M.: Study on the biomass of Chinese fir plantation, Journal of Zhejiang Forestry College, 8(3), 288-294, 1991 (in Chinese).

Lin, W.H., Chen, K.M., and Liu, Z.G.: Biomass and nutrient element content of *Eucalyptus camaldulensis* plantations of dry-hot valley in southwest Sichuan, Mountain Research, 12(4), 251-255, 1994 (in Chinese).

Ling, L.: Growth rules of individual biomass of *Populus wenxianica* forest, Protection Forest Science and Technology, (3), 9-11, 2011 (in Chinese).

Liu, B.: Study on individual biomass and allocation pattern of *Pinus tabuliformis* natural forest in Helan Mountains, China, Thesis for Master's Degree, Northwest Agricultural and Forestry University, Yangling, China, 46pp., 2010 (in Chinese).

Liu, F.M.: Biomass and nutrient storage of *Pinus massoniana* plantation on different slope positions, Journal of Fujian Forestry Science and Technology, 22(S1), 7-11, 1995a (in Chinese).

Liu, Q.: Biomass and productivity of different age-group *Pinus massoniana* plantations, Journal of Central South Forestry University, 16(4), 47-51, 1996 (in Chinese).

Liu, T.T.: Calculating biomass and carbon storage of poplar plantation based on tree structure, Thesis for Master's Degree, Beijing Forestry University, Beijing, China, 52pp., 2009.

Liu, W.Y.: Study on the biomass and productivity of *Acacia dealbata* plantation in the protected district of water sources in north Kunming, Guihaia, 15(4), 327-334, 1995b (in Chinese).

Liu, X.Y.: Study on the biomass of *Pinus elliottii* plantation in red soil hills, Forest Science and Technology, (9), 10-13, 1984 (in Chinese).





Liu, Y.C.: Tree biomass of *Pinus taiwanensis* plantation, Journal of Henan Agricultural College, (2), 21-31, 1980 (in Chinese).

Liu, E., and Liu, S.R.: Carbon storage and distribution pattern of *Mytilaria laosensis* plantation in south subtropical area, Acta Ecologica Sinica, 32(16), 5103-5109, 2012 (in Chinese).

Liu, K., and Chen, Y.E.: Biological productivity of *Robinia pseudoacacia* plantation in the Loess Plateau Area of the north Weihe River, Acta Botanica Boreali-Occidentalia Sinica, 9(3), 197-201, 1989 (in Chinese).

Liu, X.Z., and Cai, B.Y.: Comparative study between the biomass of *Pinus massoniana* plantation and *P. elliottii* plantation, Forest Resources Management, (5), 28-31, 1993 (in Chinese).

Liu, X.Z., and Kang, W.X.: The biomass of *Eucommia ulmoides* plantation, *Phellodendron chinense* plantation, and *Magnolia officinalis* var. *biloba* plantation, in: Long-term Located Research on Forest Ecosystems, Liu, X.Z., Kang, W.X., Chen, X.Y., Wen, S.Z., eds., China Forestry Publishing House, Beijing, China, 49-52, 1993 (in Chinese).

Liu, Z.G., and Ma, Q.Y.: An approach to methods for estimating biomass of *Larix principis-rupprechtii* artificial forests, Journal of Beijing Forestry University, 14(S), 105-113, 1992 (in Chinese).

Liu, B., Liu, J.J., Ren, J.H., and Du, C.X.: Individual biomass regression models for *Pinus tabuliformis* natural forest in Helan Mountains, Journal of Northwest Forestry University, 25(6), 69-74, 2010 (in Chinese).

Liu, E., Wang, H., and Liu, S.R.: Characteristics of carbon storage and sequestration in different aged beech (*Castanopsis hystrix*) plantations in south subtropical area of China, Chinese Journal of Applied Ecology, 23(2), 35-340, 2012 (in Chinese).

Liu, G.Y., Zhao, G.H., Wang, G.H., Ma, C.M., and Li, J.: Biomass allocation pattern of *Larix principis-rupprechtii* plantation, Hebei Journal of Forestry and Orchard Research, 26(3), 222-226, 2011a (in Chinese).

Liu, S.R., Chai, Y.X., Cai, T.J., and Peng, C.H.: Study on the biomass and net primary productivity of Dahurian larch plantation, Journal of Northeast Forestry University, 18(2), 40-45, 1990 (in Chinese).

Liu, T.T., Chen, J., Zhuang, J.L., and Guan, W.B.: Biomass and carbon storage of poplar plantation, China Science and Technology Fortune Magazine, (8), 192-194, 2009 (in Chinese).

Liu, X.Z., Xiang, W.H., and Kang, W.X.: Biomass dynamical pattern in a Chinese fir plantation ecosystem, in: Long-term Located Research on Forest Ecosystems, Liu, X.Z., Kang, W.X., Chen, X.Y., Wen, S.Z., eds., China Forestry Publishing House, Beijing, China, 8-22, 1993 (in Chinese).

Liu, X.Z., Wen, S.Z., and Xiang, W.H.: Effect of thinning on the biomass of *Pinus massoniana* plantation, Forest Resources Management, (5), 43-48, 1995a (in Chinese).

Liu, X.Z., Tian, D.L., Kang, W.X., and Fang, H.B.: The biomass of a young second-generation Chinese fir plantation,





Scientia Silvae Sinicae, 33(2), 61-66, 1997 (in Chinese).

Liu, Y.B., Zhang, Y.L., Zhao, T.S., Li, Z.L., Shi, Z.X., Li, Z.A., Zhang, Y.L., Meng, X.T., Hao, Z.Y., Mao, X.Q., Zhang, Y.S., and Dong, C.Y.: Biomass and productivity of the agroforestry ecosystems, Journal of Henan Agricultural College, (1), 13-20, 1984 (in Chinese).

Liu, Y.C., Wu, M.Z., Guo, Z.M., Jiang, Y.X., Liu, S.R., Wang, Z.Y., Liu, B.D., and Zhu, X.L.: Biomass and net productivity of *Quercus variabilis* forest in Baotianman Nature Reserve, Chinese Journal of Applied Ecology, 9(6), 569-574, 1998 (in Chinese).

Liu, Y.C., Wu, M.Z., Guo, Z.M., Jiang, Y.X., Liu, S.R., Wang, Z.Y., Liu, B.D., and Zhu, X.L.: Biomass and net productivity of *Quercus acutidentata* forest in Baotianman Nature Reserve, Acta Ecologica Sinica, 21(9), 1450-1456, 2001 (in Chinese).

Liu, Y.C., Jiang, Y.B., Chen, H.W., and Li, J.: Regression equations for individual tree of *Betula alnoides* plantation, Journal of Fujian Forestry Science and Technology, 35(2), 42-46, 2008 (in Chinese).

Liu, Y.H., Wang, Y.H., Yu, P.T., Xiong, W., Mo, F., and Wang, Z.Y.: Biomass and its allocation of the main vegetation types in Liupan Mountains, Forest Research, 24(4), 443-452, 2011b (in Chinese).

Liu, Z.Q., Chen, G.H., Meng, Y.Q., Li, J.G., and Liu, M.R.: Biomass and nutrient storage of *Larix principis-rupprechtii* plantation, Forest Research, 8(1), 88-93, 1995b (in Chinese).

Long, S.M., and Wang, Y.: Biomass and economic benefits of *Casuarina equisetifolia* plantation, Central South Forest Inventory and Planning, (2), 25-34, 1985 (in Chinese).

Lu, Q., Li, Z.J., and Li, X.D.: The models of biological productivity of *Castanopsis fargesii* forest, Journal of Guangxi Agricultural College, 9(3), 55-64, 1990 (in Chinese).

Lu, S.W., Cao, Y.S., Li, F.S., Chen, F.J., Liu, T., and Yang, Z.: The volume and biomass of *Larix principis-rupprechtii* plantation in mountainous areas of northwest Hebei Province, Forest Resources Management, (1), 33-36, 2012 (in Chinese).

Lu, X.Y., Chen, S.X., Li, M.Q., and Chang, X.M.: The biomass of *Paulownia elongata* in Paulownia-crop intercropping ecosystem, Paulownia and Agroforestry, (1), 12-23, 1990 (in Chinese).

Lu, Y.S., Liang, Z.H., Wu, Z.X., Cai, Y.P., Zhou, K.M., Yang, G.F., and Yin, Z.X.: Biomass and productivity of main afforestation tree species on the seawall of northern Jiangsu, Journal of Jiangsu Forestry Science and Technology, 27(2), 12-15, 2000 (in Chinese).

Luan, K.Z., and Liu, Z.G.: Biomass estimation models for *Pinus sylvestris* var. *mongolica* plantation, Forestry Science and Technology, 37(3), 35-38, 2012 (in Chinese).

Luo, R.: Effect of intermediate cutting on biological productivity of *Pinus massoniana* plantation, Journal of Sichuan





Forestry Science and Technology, 13(2), 29-34, 1992 (in Chinese).

Luo, W.X., Liu, G.Q., Tang, D.R., and Ma, S.T.: Growth increment and biomass of *Eucommia ulmoides* plantation in Weibei Loess Plateau, Journal of Northwest Forestry University, 9(4), 22-26, 1985 (in Chinese).

Luo, Y.J., Zhang, X.Q., Wang, X.K., Zhu, J.H., Zhang, Z.J., Sun, G.S., and Gao, F.: Biomass and its distribution patterns of *Larix principis-rupprechtii* plantations in northern China, Journal of Beijing Forestry University, 31(1), 13-18, 2009 (in Chinese).

Lyu, X.T., Tang, J.W., He, Y.C., Duan, W.G., Song, J.P., Xu, H.L., and Zhu, S.Z.: Biomass and its allocation in tropical seasonal rain forest in Xishuangbanna, southwest China, Chinese Journal of Plant Ecology, 31(1), 11-22, 2007.

Ma, Q.Y.: A study on the biomass and primary productivity of Chinese pine (*Pinus tabuliformis* Carr.) forests, PhD dissertation, Beijing Forestry University, Beijing, China, 178pp., 1988 (in Chinese).

Ma, Q.Y.: A study on the biomass of Chinese pine forests, Journal of Beijing Forestry University, 11(4), 1-10, 1989 (in Chinese).

Ma, W.: Measurement and estimation of ecosystem carbon density for *Larix olgensis* plantation based on FIM and FFE-FVS, PhD Dissertation, Beijing Forestry University, Beijing, China, 206pp., 2012 (in Chinese).

Ma, X.X., and Li, W.J.: Biomass table of Korean pine natural forest in northern Changbai Mountains, Forest Investigation Design, (3), 74-75, 2008 (in Chinese).

Ma, M.D., Jiang, H., and Yang, J.Y.: The biomass of *Phoebe bournei* plantation in the western boundary of Sichuan Basin, Journal of Sichuan Forestry Science and Technology, 10(3), 6-14, 1989 (in Chinese).

Ma, W., Sun, Y.J., Guo, X.Y., Ju, W.Z., and Mu, J.S.: Carbon storage of *Larix olgensis* plantation at different stand ages, *Acta Ecologica Sinica*, 30(17), 4659-4667, 2010 (in Chinese).

Ma, Z.W., Bi, J., Meng, X.S., and Li, Z.C.: Individual biomass of *Platycladus orientalis* plantation, Journal of Hebei Forestry Science and Technology, (3), 1-3, 2006 (in Chinese).

Mei, L.: Fine root turnover and carbon allocation in Manchurian ash and Dahurian larch plantations, PhD Dissertation, Northeast Forestry University, Harbin, China, 102pp., 2006 (in Chinese).

Mei, L., Zhang, Z.W., Gu, J.C., Quan, X.K., Yang, L.J., and Huang, D.: Carbon and nitrogen storages and allocation in tree layers of *Fraxinus mandshurica* and *Larix gmelinii* plantations, Chinese Journal of Applied Ecology, 20(8), 1791-1796, 2009 (in Chinese).

Meng, L., Cheng, J.M., Yang, X.M., Han, J.J., Fan, W.J., and Hu, X.J.: Carbon storage and density of *Pinus tabuliformis* plantation in Ziwuling Forest Region of the Loess Plateau, Bulletin of Soil and Water Conservation, 30(2), 133-137, 2010 (in Chinese).

Ming, A.G., Tang, J.X., Yu, H.L., Shi, Z.M., Lu, L.H., Jia, H.Y., and Cai, D.X.: Individual biomass regression model of





*Mytilaria laosensis* in southwest Guangxi, Forest Resources Management, (6), 83-87, 93, 2011 (in Chinese).

Ming, A.G., Zhang, Z.J., Chen, H.H., Zhang, X.Q., Tao, Y., and Su, Y.: Effect of thinning on the biomass and carbon storage in *Pinus massoniana* plantation, Scientia Silvae Sinicae, 49(10), 1-6, 2013 (in Chinese).

Mo, D.X.: Study on carbon and nitrogen pattern and ecosystem biomass of *Cryptomeria fortunei* plantation of different density in southeastern Guangxi Province, Thesis for Master's Degree, Guangxi University, Nanjing, China, 52pp., 2013 (in Chinese).

Mu, T.M.: Study on the biomass of *Picea crassifolia* forest in Helan Mountains, Inner Mongolia Forestry Science and Technology, (1), 34-45, 1981 (in Chinese).

Mu, C.C., Wan, S.C., Su, P., Song, H.W., and Zun, Z.H.: Biomass distribution patterns of *Alnus hirsuta* and *Betula platyphylla*-swamp ecotone communities in Changbai Mountains, Chinese Journal of Applied Ecology, 15(12), 2211-2216, 2004 (in Chinese).

Mu, C.C., Wang, B., Lu, H.C., Bao, X., and Cui, W.: Carbon storage of natural wetland ecosystem in Daxing'anling of China, Acta Ecologica Sinica, 33(16), 4956-4965, 2013 (in Chinese).

Mu, L.Q., Liu, X.J., Xu, H., Li, G.X., and Guo, B.Q.: Analysis on key factors affecting the biomass of *Picea koraiensis* artificial forests, Journal of Northeast Forestry University, 23(6), 95-102, 1995a (in Chinese).

Mu, L.Q., Zhang, J., Liu, X.J., Wu, S.N., and Yang, G.C.: Study on the tree biomass of *Picea koraiensis* artificial forests, Bulletin of Botanical Research, 15(4), 551-557, 1995b (in Chinese).

Ning, X.B., and Liu, Q.: Biological productivity of young *Pinus massoniana* plantation in different regions, Forest Resources Management, (1), 48-51, 1996 (in Chinese).

Ouyang, S.L., Dai, C.D., Hou, Y.N., Xu, Y.X., and Luo, J.: Biomass models for main constructive tree species of the protection forest system around Dongting Lake, Hunan Forestry Science and Technology, 37(5), 22-24, 2010 (in Chinese).

Pan, K.W., and Liu, Z.G.: The biomass of 10-year-old *Cercidiphyllum japonicum* plantation, Chinese Journal of Applied and Environmental Biology, 5(2), 121-130, 1999 (in Chinese).

Pan, R.D., and Wu, H.L.: Community structure of *Acacia mearnsii* plantation, Journal of Zhejiang Forestry College, 5(1), 28-35, 1988 (in Chinese).

Pan, F.J., Zhang, Z.F., Huang, Y.Q., and Mo, L.: Aboveground biomass of *Cyclobalanopsis glauca* analyzed by tree-ring method in karst region, Guihaia, 32(4), 464-467, 2012 (in Chinese).

Pan, P., Li, R.W., Qin, Z.G., Quan, M.J., and Cao, J.: The biomass and productivity of *Eucommia ulmoides* plantation, Resources and Environment in the Yangtze Basin, 9(1), 71-77, 2000 (in Chinese).

Pan, P., Li, R.W., Xiang, C.H., Zhu, Z.F., and Yin, X.M.: The biomass and productivity of *Cupressus lusitanica*





plantation, Resources and Environment in the Yangtze Basin, 11(2), 133-136, 2002 (in Chinese).

Pan, W.C., Li, L.C., Gao, Z.H., Zhang, X.Q., and Tang, D.Y.: The biomass and productivity of Chinese fir plantations, Hunan Forestry Science and Technology, (5), 1-12, 1978 (in Chinese).

Pan, W.C., Tian, D.L., Li, L.C., and Gao, Z.H.: Study on the nutrient cycling in Chinese fir plantation: (1) Biomass allocation and nutrient dynamics of Chinese fir plantations with different stand ages, Journal of Central South Forestry Institute, (1), 1-21, 1981 (in Chinese).

Pan, W.C., Tian, D.L., Lei, Z.X., and Kang, W.X.: Study on the nutrient cycling in Chinese fir plantation: (2) Content, accumulation rate and biological cycling of nutrient elements in the fast-growing Chinese fir forest in the hilly regions, Journal of Central South Forestry Institute, 3(1), 1-17, 1983 (in Chinese).

Pan, Y.J., Wang, B., Chen, B.F., and Peng, Q.Z.: Carbon sink of Chinese fir plantation ecosystem in Dagang Mountain, Jiangxi Province, Journal of Central South University of Forestry and Technology, 33(10), 120-125, 2013 (in Chinese).

Pang, J.P.: Carbon storage and its allocation of rubber plantation in Xishuangbanna, southwest China, Thesis for Master's Degree, Xishuangbanna Tropical Botanical Garden, Chinese Academy of Sciences, Xishuangbanna, China, 54pp., 2009 (in Chinese).

Peng, H.C.: Biomass measurement of *Eucommia ulmoides* plantation, Nonwood Forest Research, 10(2), 28-33, 1992 (in Chinese)

Peng, S.L., and Zhang, Z.P.: Biomass and primary productivity of two dominant species, *Cryptocarya concinna* and *C. chinensis*, of forest vegetation in Dinghu Mountain, Acta Phytoecologica et Geobotanica Sinica, 14(1), 23-32, 1990 (in Chinese).

Peng, S.L., and Zhang, Z.P.: Biomass and primary productivity of dominant species *Aporosa yunnanensis* and *Blastus cochinchinensis* of forest vegetation in Dinghu Mountain, Chinese Journal of Applied Ecology, 3(3), 202-206, 1992 (in Chinese).

Peng, P.H., Peng, J.S., Wang, C.S., and Wang, J.X.: The biomass and productivity of *Populus schneideri* var. *tibetica* plantation, Forestry Science and Technology, 28(4), 14-18, 2003 (in Chinese).

Peng, S.L., Li, M.G., and Lu, Y.: A preliminary study on the biomass and productivity of *Pinus massoniana* population in Dinghushan Biosphere Reserve, in: Tropical and Subtropical Forest Ecosystem, Volume 5, Dinghushan Forest Ecosystem Station, eds., Science Press, Beijing, China, 75-82, 1989 (in Chinese).

Peng, S.L., Yu, Z.Y., Zhang, W.Q., and Zeng, X.P.: Coenological analysis of five man-made forests on the downland of Heshan, Guangdong Province, Acta Phytoecologica et Geobotanica Sinica, 18(1), 1-10, 1992 (in Chinese).

Peng, S.L., Fang, W., Cao, H.L., Yu, Z.Y., and Ren, H.: Influence of human disturbance on tropical artificial



*Eucalyptus* forest ecosystem, Acta Ecologica Sinica, 15(S1), 31-37, 1995 (in Chinese).

Qi, L.H.: Study on the comprehensive managing technologies of aerially seeded *Pinus massoniana* stands in Hunan
Province, Thesis for Master's Degree, Central South Forestry University, Zhuzhou, China, 59pp., 2003 (in
Chinese).

Qi, J.F., and Tang, J.W.: Biomass and its allocation pattern of limestone monsoon rain forest in Xishuangbanna,
Chinese Journal of Ecology, 27(2), 167-177, 2008 (in Chinese).

Qi, X.S., and Zhou, C.R.: Community characteristics and the biomass of *Platycladus orientalis* and *Pinus tabuliformis*
mixed plantation, Journal of Shandong Forestry Science and Technology, (2), 1-5, 1991 (in Chinese).

Qi, G., Wang, Q.L., Wang, X.C., Qi, L., Wang, Q.W., Ye, Y.J., and Dai, L.M.: Vegetation carbon storage in *Larix
gmelinii* plantations in Daxing'an Mountains, Chinese Journal of Applied Ecology, 22(2), 273-279, 2011.

Qi, L.H., Zhang, X.D., Zhou, J.X., Li, Z.H., Huang, L.L., and Yang, M.H.: Changing regularity and structural
characteristics of the biomass and productivity of aerially seeded *Pinus massoniana* plantation, Forest Research,
20(3), 344-349, 2007 (in Chinese).

Qi, Z.Y., Ma, J.X., and Li, S.M.: The biomass and productivity of artificial *Michelia macclurei* forest, Chinese Journal
of Ecology, (2), 17-19, 30, 1985 (in Chinese).

Qian, G.Q.: Dynamics of the net primary productivity of *Liquidambar formosana* plantation, Acta Agriculturae
Universitatis Jiangxiensis, 22(3), 399-404, 2000 (in Chinese).

Qin, Z.G.: Community features and the biomass of *Alnus cremastogyne* fuelwood forest, Journal of Sichuan Forestry
Science and Technology, 13(1), 24-28, 33, 1992 (in Chinese).

Qin, J., Meng, H.S., Qin, W.M., Yu, J.M., and Qin, D.W.: The biomass and productivity of *Tsoongiodenron odorum*
plantation, China Forestry Science and Technology, 25(6), 65-68, 2011a (in Chinese).

Qin, L., He, Y.J., Li, Z.Y., Shao, M.X., Liang, X.Y., and Tan, L.: Allocation pattern of biomass and productivity for
three plantations of *Castanopsis hystrix*, *Pinus massoniana* and their mixture in south subtropical area of Guangxi,
China, Scientia Silvae Sinicae, 47(12), 17-21, 2011b (in Chinese).

Qin, S.J., Li, K., Mo, D.X., and Wu, Q.B.: Biomass regression model of *Cryptomeria fortunei* plantation in southeast
Guangxi, Journal of Southern Agriculture, 44(2), 261-265, 2013 (in Chinese).

Qin, W.M., Qiu, B.F., Qin, J., Xu, T.B., and Qin, D.W.: The biomass and growth rhythm of *Paramichelia baillonii*
plantation, Journal of Fujian College of Forestry, 31(2), 110-114, 2011c (in Chinese).

Qiu, X.Z., Xie, S.C.: Studies on The Forest Ecosystem in Ailao Mountains, Yunnan, China, Yunnan Science and
Technology Press, Kunming, China, 1998 (in Chinese)

Qiu, X.Z., Xie, S.C., and Jing, G.F.: A preliminary study on the biomass of *Lithocarpus xylocarpus* forest in Xujiaba



Region, Ailao Mountain, Yunnan, Acta Botanica Yunnanica, 6(1), 85-92, 1984 (in Chinese).

Qiu, Y., Zhang, J.T., Chai, B.F., and Zheng, F.Y.: The aboveground biomass and productivity of *Pinus tabuliformis* planted forest in the west of Shanxi Province, Henan Science, 17(S), 72-76, 79, 1999 (in Chinese).

Shen, Y.: Biomass and nutrient cycling of natural secondary mixed forest in subtropical area, Thesis for Master's Degree, Central South University of Forestry and Technology, Changsha, China, 116pp., 2011 (in Chinese).

Shen, J.P., Zhang, W.H., Li, Y.H., You, J.J., Yu, B.Y., Yang, X.Z., and He, J.F.: Characteristics of carbon storage and sequestration of *Pinus tabuliformis* forest land converted from farmland in the loess hilly area, Acta Botanica Boreali-Occidentalia Sinica, 33(11), 2309-2316, 2013 (in Chinese).

Shen, Y., Tian, D.L., Yan, W.D., and Xiao, Y.: Biomass and its distribution of natural secondary *Quercus fabri* + *Sassafras tsumu* + *Cunninghamia lanceolata* community in Yuanling County, Hunan Province, Journal of Central South University of Forestry and Technology, 31(5), 44-51, 2011a (in Chinese).

Shen, Y.Z., Sun, X.M., Zhang, J.T., Du, Y.C., and Ma, J.W.: Individual tree biomass of *Larix kaempferi* plantation in Xiaolong Mountain, Gansu Province, Forest Research, 24(4), 517-522, 2011b (in Chinese).

Shen, Z.K., Lu, S.P., and Ai, X.R.: The biomass and productivity of *Larix kaempferi* plantation, Journal of Hubei Institute for Nationalities (Natural Science Edition), 23(3), 289-292, 2005 (in Chinese).

Shi, Y.L.: Study on the ecosystem biomass of the artificial *Cunninghamia lanceolata* forest in Changling (Wanli District), Nanchang City, Acta Agriculturae Universitatis Jiangxiensis, 11(4), 32-45, 1989 (in Chinese).

Shi, P.L., Zhong, Z.C., and Li, X.G.: A study on the biomass of alder-cypress artificial mixed forest in Sichuan, Acta Phytoecologica Sinica, 20(6), 524-533, 1996 (in Chinese).

Shi, P.L., Yang, X., and Zhong, Z.C.: Dynamics of population biomass and its density-dependent regulation in alder and cypress mixed forest, Chinese Journal of Applied Ecology, 8(4), 341-346, 1997 (in Chinese).

Shi, Z., Liu, J., Lan, X., Liang, D., Lu, M.B., and Deng, M.Y.: Individual biomass regression models of *Pinus massoniana* plantation in Tianlin County, Guangxi Province, Guangxi Forestry Science, 38(3), 167-170, 2009 (in Chinese).

Si, J.P., Yao, R.M., Chen, D.B., and Wu, C.H.: The biomass of *Magnolia officinalis* plantation, Journal of Zhejiang Forestry College, 10(2), 162-168, 1993 (in Chinese).

Song, D.Q., Jiang, Z.L., Zheng, Z.M., and He, W.P.: Study on the biomass of *Betula luminifera* population, Journal of Nanjing Forestry University (Natural Science Edition), 26(3), 40-42, 2002 (in Chinese).

Su, Y.M.: Study on the biomass and productivity of *Larix kaempferi* plantation, Journal of Sichuan Forestry Science and Technology, 16(3), 36-42, 1995 (in Chinese).

Su, Y., Qin, W.M., Huang, S.D., Duan, W.W., Wei, Z.M.: The biomass and productivity of *Parashorea chinensis*



plantation in the southwest Guangxi, Practical Forestry Technology, (9), 16-19, 2011 (in Chinese).

Su, Y.M., Liu, X.L., Xiang, C.H.: Study on the biomass and net primary productivity of *Abies fabri* plantation, Journal of Sichuan Forestry Science and Technology, 21(2), 31-35, 2000 (in Chinese).

Sun, B.G., Chen, F., Wang, J.M., Chen, X.M., Yang, Z.X., Cai, X.Y., and Li, B.: Biomass distribution pattern of *Pinus yunnanensis* with different diameter classes, Forest Research, 25(1), 71-76, 2012a (in Chinese).

Sun, D., Ruan, H.H., and Ye, J.Z.: The biomass structure of the secondary natural oakery in Kongqing Hill, in: Proceedings of forest ecosystems on Xiashu Ecological Station, Jiang, Z.L., eds., China Forestry Publishing House, Beijing, China, 16-22, 1992 (in Chinese).

Sun, N., Li, Y.Z., and Zhang, Y.C.: Influence of afforestation density on the biomass of hybrid larch, Forestry Science and Technology, 37(4), 14-16, 2012b (in Chinese).

Sun, Q.X., Yu, F.A., and Peng, Z.H.: The biomass of poplar plantation in the beach land of Yangtze River, Forest Science and Technology, (3), 4-6, 1998 (in Chinese).

Tang, L.Z., and Sun, Y.L.: Effect of water table on the growth of I-69 poplar in Lixiahe wetland, Jiangsu Province, China, Wetland Science, 5(2), 140-145, 2007 (in Chinese).

Tang, W., and Xu, R.Q.: Study on the biomass of *Abies fabri* plantation, Sichuan Forestry Exploration and Design, (2), 27-32, 1993 (in Chinese).

Tang, J.W., Pang, J.P., Chen, M.Y., Guo, X.M., and Zeng, R.: Biomass and its estimation model of rubber plantations in Xishuangbanna, southwest China, Chinese Journal of Ecology, 28(10), 1942-1948, 2009 (in Chinese).

Tang, J.W., Zhang, J.H., Song, Q.S., Cao, M., Feng, Z.L., Dang, C.L., and Wu, Z.L.: A preliminary study on the biomass of secondary tropical forest in Xishuangbanna, Acta Phytoecologica Sinica, 22(6), 489-498, 1998 (in Chinese).

Tang, L.Z., Haibara, K., Huang, B.L., Toda, H., Yang, W.Z., and Arai, T.: Storage and dynamics of carbon in a poplar plantation in Lixiahe Region, Jiangsu Province, Journal of Nanjing Forestry University (Natural Science Edition), 28(2), 1-6, 2004a (in Chinese).

Tang, L.Z., Yu, M.K., Yan, C.F., Liu, Z.L., and Fang, S.Z.: Effects of site condition and cultivation on the growth of sawtooth oak plantations, Journal of Fujian College of Forestry, 28(2), 130-135, 2008 (in Chinese).

Tang, W.P., Wang, Y.R., and Zheng, L.Y.: Study on the biomass and productivity of southern type poplar plantation, Hubei Forestry Science and Technology, (S), 43-47, 2004b (in Chinese).

Tian, D.L., Pan, W.C., Lei, Z.X., Long, E.H., and Cai, B.Y.: A preliminary study on the biomass and density effects of *Pinus massoniana* plantation, Journal of Central South Forestry Institute, 2(1), 41-50, 1982 (in Chinese).

Tian, D.L., Zhang, C.J., Luo, Z.P., and Yuan, W.X.: Biomass and distribution of nutrient elements in natural *Sassafras*



*tzumu* mixed forest: (1) The biomass and productivity, Journal of Central South Forestry College, 10(2), 121-128, 1990 (in Chinese).

Tian, D.L., Pan, H.H., Kang, W.X., and Fang, H.B.: The biomass of a second-generation Chinese fir plantation, Journal of Central South Forestry University, 18(3), 11-16, 1998 (in Chinese).

Tian, D.L., Xiang, W.H., Yan, W.D., and Kang, W.X.: Effect of successive rotation on the biomass and productivity of Chinese fir plantation at fast-growing stage, Scientia Silvae Sinicae, 38(4), 14-18, 2002 (in Chinese).

Tian, D.L., Yin, G.Q., Fang, X., Xiang, W.H., and Yan, W.D.: Carbon density, storage and spatial distribution under different 'Grain for Green' patterns in Huitong, Hunan Province, Acta Ecologica Sinica, 30(22), 6297-6308, 2010 (in Chinese).

Tian, Q.F., Du, L.H., and Li, X.J.: Study on the biomass of *Robinia pseudoacacia* plantation in Xishan National Forest Park, Beijing, Journal of Beijing Forestry University, 19(S2), 104-107, 1997a (in Chinese).

Tian, Q.F., Yan, H.P., and Wang, P.X.: Comparison of the productivity between three deciduous broadleaved plantations, Journal of Beijing Forestry University, 19(S2), 118-122, 1997b (in Chinese).

Tian, Q.F., Zhou, R.W., and Zhang, J.S.: Study on the biomass of *Quercus variabilis* plantation in Xishan National Forest Park, Beijing, Journal of Beijing Forestry University, 19(S2), 113-117, 1997c (in Chinese).

Tong, J.Q.: Study on the productivity and growth of Chinese fir plantations with different sites and densities, Journal of Fujian Agriculture and Forestry University (Natural Science Edition), 37(4), 369-373, 2008 (in Chinese).

Wan, S.C.: The biomass allocation patterns of constructive tree species in forest-swamp ecotone, Journal of Mudanjiang University, 22(8), 124-127, 2013 (in Chinese).

Wan, M., Tian, D.L., Fan, W., and Li, Q.Y.: Biomass production and carbon sequestration in poplar-crop agroforestry ecosystems in eastern Henan Plain, Scientia Silvae Sinicae, 45(8), 27-33, 2009 (in Chinese).

Wang, B.: Carbon storage of natural wetland ecosystem in southern Daxing'an Mountains of China, Thesis for Master's Degree, Northeast Forestry University, Harbin, China, 39pp., 2013 (in Chinese).

Wang, C.K.: Biomass allometric equations for 10 co-occurring tree species in Chinese temperate forests, Forest Ecology and Management, 222, 9-16, 2006.

Wang, J.: Community characteristics and biomass of young mixed plantation of *Alnus cremastogyne* and *Cupressus funebris*, Sichuan Forestry Science and Technology, 14(1), 66-69, 1993.

Wang, L.M.: Biomass determination of stem, branch and needle of *Pinus sylvestris* var. *mongolica* natural forest, Journal of Inner Mongolia Forestry College, (2), 63-68, 1986 (in Chinese).

Wang, M.B.: The biomass and afforestation prospect of *Pinus tabuliformis* plantation in the northwest Shanxi Province, Journal of Shanxi University (Natural Science Edition), (4), 98-102, 1988 (in Chinese).



Wang, M.B.: The biomass of *Populus hopeiensis* forest, Journal of Shanxi University (Natural Science Edition), 14(1), 103-107, 1991 (in Chinese).

Wang, W.C.: The aboveground biomass of *Pinus tabuliformis* plantation, Shanxi Forestry Science and Technology, (2), 10-15, 1985 (in Chinese).

Wang, X.Y.: Carbon storage distribution of *Larix olgensis* plantation at different stand ages, PhD Dissertation, Beijing Forestry University, Beijing, China, 96pp., 2011 (in Chinese).

Wang, J., and Wen, Z.W.: Aboveground biomass distribution and predictive model of secondary *Betula luminifera* forest in the northwest Guizhou Province, Guizhou Forestry Science and Technology, 39(2), 18-21, 2011 (in Chinese).

Wang, Q.M., and Shi, Y.G.: A preliminary study on the biomass and production of slash pine plantation in Jiangsu Province, Acta Phytoecologica et Geobotanica Sinica, 14(1), 1-12, 1990 (in Chinese).

Wang, X.H., and Song, Y.C.: Dynamics of population and biomass at early natural recovery stage of vegetation on abandoned quarry of Tiantong National Forest Park, Chinese Journal of Applied Ecology, 10(5), 545-548, 1999 (in Chinese).

Wang, Y.M., and Liu, B.Z.: Ecological Characteristics of Protection Forests in the Loess Plateau, China Forestry Publishing House, Beijing, China, 42-48, 1994 (in Chinese).

Wang, Z., and Zhang, S.Y.: Larch Forests in China, China Forestry Publishing House, Beijing, China, 50-57, 1992 (in Chinese).

Wang, Z.F., and Feng, Z.K.: The parameter estimation of tree biomass using the nonlinear least square method, Journal of Jilin Agricultural University, 28(3), 261-264, 2006 (in Chinese).

Wang, B., Nie, D.P., Guo, Q.S., and Xia, L.F.: Studies on Forest Ecosystems in Dagang Mountains of Jiangxi Province, China Science and Technology Press, Beijing, China, 183-185, 2003a (in Chinese).

Wang, C., Gao, H.Z., and Zang, Y.Q.: Study on the individual biomass of *Betula platyphylla* natural secondary forest, Forestry Science and Technology, 35(1), 7-9, 13, 2010 (in Chinese).

Wang, D.Y., Li, D.Y., and Feng, X.Q.: Forest Ecosystems on Warm Temperate Zone, China Forestry Publishing House, Beijing, China, 190-205, 2003b (in Chinese).

Wang, H.M., Chen, Z.J., and Li, X.W.: Biomass and nutrient content of different forest-grass ecosystems converted from farmlands, Research of Soil and Water Conservation, 12(2), 125-128, 2005 (in Chinese).

Wang, H.Y., Wang, W.J., Qiu, L., Su, D.X., An, J., Zheng, G.Y., and Zu, Y.G.: Differences in biomass, litter layer mass and SOC storage changing with tree growth in *Larix gmelinii* plantations in northeast China, Acta Ecologica Sinica, 32(3), 833-843, 2012a (in Chinese).



Wang, J.S., Zhang, C.Y., Fan, X.H., and Zhao, Y.Z.: Biomass allocation patterns and allometric models of *Abies nephrolepis*, Acta Ecologica Sinica, 31(14), 3918-3927, 2011a (in Chinese).

Wang, J.S., Fan, X.H., Fan, J., Zhang, C.Y., and Xia, F.C.: Effect of aboveground competition on biomass partitioning of understory Korean pine (*Pinus koraiensis*), Acta Ecologica Sinica, 32(8), 2447-2457, 2012b (in Chinese).

Wang, J.Y., Che, K.J., Fu, H.E., Chang, X.X., Song, C.F., and He, H.Y.: Study on the biomass of water conservation forest on the north slope of Qilian Mountains, Journal of Fujian College of Forestry, 18(4), 319-323, 1998 (in Chinese).

Wang, N., Wang, B.T., Wang, R.J., Cao, X.Y., Wang, W.J., and Chi, L.: Biomass allocation patterns and allometric models of *Populus davidiana* and *Pinus tabuliformis* in the west Shanxi Province, Bulletin of Soil and Water Conservation, 33(2), 151-155, 159, 2013 (in Chinese).

Wang, Q.C., Wang, Q.L., Li, D., Wu, N.S., and Cheng, Z.C.: Suitable management density of multi-benefit management pattern in *Pinus elliottii* plantation on hilly region of Jiangxi Province, Chinese Journal of Applied Ecology, 12(5), 663-666, 2001 (in Chinese).

Wang, S.J., Chen, B.H., and Li, H.Q.: *Populus euphratica* Forests, China Environmental Science Press, Beijing, China, 1995.

Wang, T., Yuan, Z.L., Ye, Y.Z., and Zhang, J.: Preliminary study on the biomass of *Platycladus orientalis* plantation at different stand ages in the Songshan Mountain National Forest Park, Henan Science, 27(7), 817-820, 2009 (in Chinese).

Wang, X.W., Li, X.Y., Guan, X.D., and Bai, G.X.: Preliminary study on the biomass of *Larix olgensis* plantation, Journal of Liaoning Forestry Science and Technology, (6), 30-34, 1993 (in Chinese).

Wang, X.Y., Hu, D., and He, J.S.: Study on the biomass of *Fagus engleriana* forest and *Quercus aliena* var. *acuteserrata* forest in Shennongjia Region, Journal of Capital Normal University (Natural Science Edition), 28(2), 62-67, 2007a (in Chinese).

Wang, X.Y., Sun, Y.J., and Ma, W.: Biomass and carbon storage distribution of *Larix olgensis* plantations with different density, Journal of Fujian College of Forestry, 31(3), 221-226, 2011b (in Chinese).

Wang, Y., Li, Q., Zhang, P.Y., Wu, X.X., Jiang, Y.Z., and Jiang, J.T.: The biomass and nutritive element content of the fast-growing poplar plantations, Journal of Shandong Forestry Science and Technology, (2), 1-7, 1990 (in Chinese).

Wang, Y.T., Ma, Q.Y., Hou, G.W., Kan, Z.G., and Chen, Y.: Dynamics of the biomass and productivity of the naturally-regenerated *Pinus densata* forest in the burned areas of the western Sichuan Province, Forestry Science and Technology, 32(1), 37-40, 2007b (in Chinese).




Wang, Z., Han, Y.J., Kang, H.Z., Huang, D., Xue, C.Y., Yin, S., and Liu, C.J.: Carbon storage of main water conservation plantations in the upper reaches of Huangpu River, Shanghai, Chinese Journal of Ecology, 31(8), 1930-1035, 2012c (in Chinese).

Wei, Y.J.: Study on the biomass of *Fokienia hodginsii* plantations in south Fujian, Journal of Fujian Forestry Science and Technology, 28(3), 21-23, 38, 2001 (in Chinese).

Wei, F., Li, X.X., Wen, Y.G., Liang, H.W., and Tang, C.Y.: An ecological study on Chinese fir plantation in Laoshan Mountain, Tianlin County, the northwest of Guangxi, Journal of Guangxi Agricultural College, 10(4), 1-26, 1991 (in Chinese).

Wen, S.Z.: The biomass and productivity of *Castanopsis fissa* stand, in: Long-term Located Research on Forest Ecosystems, Liu, X.Z., Kang, W.X., Chen, X.Y., Wen, S.Z., eds., China Forestry Publishing House, Beijing, China, 42-48, 1993 (in Chinese).

Wen, D.C., Deng, Z.R., and Xiang, H.H.: Study on the productivity of Ussuri poplar plantation, Journal of Northeast Forestry University, 15(2), 42-49, 1987 (in Chinese).

Wen, D.Z., Wei, P., Kong, G.H., Zhang, Q.M., and Huang, Z.L.: Biomass study of the *Castanopsis chinensis + Cryptocarya concinna + Schima superba* community in Dinghushan Biosphere Reserve, Acta Ecologica Sinica, 17(5), 497-504, 1997 (in Chinese).

Wen, D.Z., Wei, P., Zhang, Q.M., and Kong, G.H.: Study on the biomass of three subtropical evergreen broadleaved forests in Dinghushan Biosphere Reserve, Acta Phytoecologica Sinica, 23(S), 11-21, 1999 (in Chinese).

Wen, S.Z., Pan, W.C., Tian, D.L., and Kang, W.X.: Hydroecological benefits from Chinese fir plantation ecosystem: Preliminary report of comprehensive trial of small forested watershed, Journal of Central South Forestry College, 9(S), 11-22, 1989 (in Chinese).

Wen, Y.G., Liang, L.R., Li, J.J., Wei, B.E., and Xiong, C.D.: Biological productivity of *Cunninghamia lanceolata* plantations in different ecogeographical regions, Guangxi, Journal of Guangxi Agricultural College, 7(2), 55-66, 1988 (in Chinese).

Wen, Y.G., Liang, H.W., and Jiang, H.P.: The biomass and its distribution of Chinese fir plantations in Guangxi, Journal of Guangxi Agricultural University, 14(1), 55-64, 1995 (in Chinese).

Wen, Y.G., He, T.P., Li, X.X., Liang, H.W., Zhou, J.Y., and Su, Y.J.: The biomass and productivity of *Eucalyptus exserta* coastal protection plantation in Hepu County, Guangxi, Journal of Guangxi Agricultural and Biological Science, 19(1), 1-5, 2000a (in Chinese).

Wen, Y.G., Liang, H.W., Zhao, L.J., Zhou, M.Y., He, B., Wang, L.H., Wei, S.H., Zheng, B., Liu, D.J., and Tang, Z.S.: Biomass and productivity of *Eucalyptus urophylla* plantation, Journal of Tropical and Subtropical Botany, 8(2),





123-127, 2000b (in Chinese).

Wu, Q.Z.: Study on the biomass of *Mytilaria laosensis* plantation, Journal of Fujian Forestry Science and Technology, 32(3), 125-129, 2005 (in Chinese).

Wu, X.C.: Study on the productivity and carbon density of poplar-willow natural forest in Ergis River, Xinjiang, PhD Dissertation, Inner Mongolia Agricultural University, Huhhot, China, 109pp., 2009 (in Chinese).

Wu, Z.L., and Dang, C.L.: Study on the biomass and net primary productivity of *Pinus kesiya* var. *langbianensis* stands in Changning District, Yunnan Province, Journal of Yunnan University (Natural Science Edition), 14(2), 137-145, 1992a (in Chinese).

Wu, Z.L., and Dang, C.L.: Study on the biomass of *Pinus kesiya* var. *langbianensis* stands in Pu'er District, Yunnan Province, Journal of Yunnan University (Natural Science Edition), 14(2), 119-127, 1992b (in Chinese).

Wu, Z.L., and Dang, C.L.: Preliminary study on the biomass and net primary productivity of *Quercus senescens* forest near Kunming, Journal of Yunnan University (Natural Science Edition), 16(3), 235-239, 244, 1994 (in Chinese).

Wu, G., Feng, Z.W., Kong, H.M., and Qin, Y.Z.: The biomass and productivity of agroforestry ecosystems in northern Henan Province, Research of Agricultural Modernization, 14(2), 99-102, 1993 (in Chinese).

Wu, H.X., Shi, Y.G., Zhang, H.Z., and Zhang, S.C.: Study on the biomass of *Pinus tabuliformis* plantation in Badaling Forest Farm, Hebei Journal of Forestry and Orchard Research, 21(3), 240-242, 2006 (in Chinese).

Wu, M., Li, S.T., Qin, W.M., and Qin, X.L.: The biomass and productivity of 11-year-old *Michelia hedyosperma* plantation, Journal of West China Forestry Science, 42(4), 46-51, 2013 (in Chinese).

Wu, P., Ding, F.J., Cui, Y.C., Zhu, J., and Li, C.R.: The biomass and productivity of young *Pinus massoniana* forests of Pearl River Shelterbelt Construction Project in Qiannan City, Guizhou Agricultural Sciences, 40(6), 169-172, 2012a (in Chinese).

Wu, S.D., Ye, G.F., Pan, H.Z., Xu, J.S., Long, X.W., Zheng, R., and Huang, C.Y.: Biomass and its distribution of *Casuarina equisetifolia* plantation, Protection Forest Science and Technology, (S1), 21-24, 1996 (in Chinese).

Wu, S.R., Yang, H.Q., Hong, R., Zhu, W., and Chen, X.Q.: The biomass and its distribution of *Pinus massoniana* plantation, Journal of Fujian Forestry Science and Technology, 26(1), 18-21, 1999 (in Chinese).

Wu, T., Wang, Y., Yu, C.J., Chiarawipa, R., Zhang, X.Z., Han, Z.H., and Wu, L.H.: Carbon sequestration by fruit trees: Chinese apple orchards as an example, PLoS ONE, 7(6), e38883, 2012b.

Wu, W.Y., Huang, Q.F., and Huang, C.L.: The biomass and growth increment of poplar plantations in the Nangeng marshlands of Yangtze River, Anhui Forestry Science and Technology, 38(2), 53-57, 72, 2012c (in Chinese).

Wu, X.C., Zhang, Q.L., Lei, Q.Z., and Bai, Z.Q.: Biomass distribution characteristics of natural forests in Ergis River, Xinjiang, Forest Resources Management, (4), 61-67, 2009 (in Chinese).

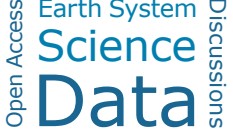

Wu, Z.L., Dang, C.L., He, Z.R., and Wang, C.Y.: Preliminary study on the biomass and net primary productivity of *Quercus pannosa* forest in the northwest Yunnan Province, Journal of Yunnan University (Natural Science Edition), 16(3), 245-249, 1994a (in Chinese).

Wu, Z.L., Dang, C.L., He, Z.R., and Wang, C.Y.: Preliminary study on the biomass of *Picea brachytla* var. *complanata* forests in the northwest Yunnan Province, Journal of Yunnan University (Natural Science Edition), 16(3), 230-234, 1994b (in Chinese).

Wu, Z.L., Dang, C.L., Wang, C.Y., and He, Z.R.: Preliminary study on the biomass of *Pinus densata* forests in the northwest Yunnan Province, Journal of Yunnan University (Natural Science Edition), 16(3), 220-224, 1994c (in Chinese).

Wu, Z.M., Sun, Q.X., and Duan, W.X.: Relationship between flooded situation and poplar growth on beach land of Yangtze river in Anhui, Chinese Journal of Applied Ecology, 11(1), 25-29, 2000 (in Chinese).

Wu, Z.M., Sun, Q.X., and Chen, M.G.: Biomass and nutrient accumulation of poplar plantation in the beach land of Yangtze River in Anhui Province, Chinese Journal of Applied Ecology, 12(6), 806-810, 2001 (in Chinese).

Xia, L.Y., Li, S.C., Liao, X.L., Wang, W.D., and Li, D.F.: A study on the biomass of slash pine (*Pinus elliottii*) plantation, Subtropical Forest Science and Technology, 15(1), 20-25, 1987 (in Chinese).

Xiang, W.H., Chen, X.Y., and Cai, B.Y.: The time-lapse characteristics of the biomass in slash pine plantation, in: Long-term Located Research on Forest Ecosystem, Liu, X.Z., Kang, W.X., Chen, X.Y., Wen, S.Z., eds., China Forestry Publishing House, Beijing, China, 60-64, 1993 (in Chinese).

Xiao, Y.: Study on the productivity of *Betula platyphylla* natural secondary forest, Acta Biologica Plateau Sinica, (8), 147-157, 1988 (in Chinese).

Xiao, Y.: Comparative study on the biomass and productivity of *Pinus tabuliformis* plantations in different climatic zones, Shaanxi Province, Acta Phytoecologica et Geobotanica Sinica, 14(3), 237-246, 1990 (in Chinese).

Xiao, Y.: Biomass and productivity of natural *Pinus henryi* forest, Acta Phytoecologica et Geobotanica Sinica, 16(3), 227-233, 1992 (in Chinese).

Xiao, X.C., Li, Z.H., Tang, Z.J., Sheng, J.M., Gong, X.J., and Zhu, N.: Effects of stand density on the biomass and productivity of *Pinus elliottii* plantation, Journal of Central South University of Forestry and Technology, 31(3), 123-129, 2011 (in Chinese).

Xiao, Y., Wu, B.S., Chen, B.Q., Zhang, J.C., Wang, M.H., and Wang, L.G.: A preliminary study on the aboveground biomass of *Pinus tabuliformis* plantation, Shanxi Forestry Science and Technology, (2), 5-14, 1983 (in Chinese).

Xiao, Y.T., He, A.H., and Liang, J.Z.: Community characteristics and the biomass of *Pinus taiwanensis* natural forest in Mufu Mountain of Hubei-Hunan border, Hunan Forestry Science and Technology, (4), 1-4, 1989 (in Chinese).



Xiao, Z.W., Wang, L.J., Mao, J.M., Zhu, X.Z., Wang, X.L., Zheng, L., and Tang, J.W.: Carbon storage of different tree-tea agroforestry systems in Xishuangbanna, Yunnan Province of southwest China, Chinese Journal of Ecology, 31(7), 1617-1625, 2012 (in Chinese).

Xie, W.D., Wen, Y.G., Zhou, M.Y., Liang, H.W., Liu, S.R., and Chen, F.: Biomass and productivity of *Eucalyptus urophylla* plantation in new cultivated areas, Journal of Central South University of Forestry and Technology, 27(5), 13-18, 2007 (in Chinese).

Xie, W.D., Wen, Y.G., Liang, H.W., Li, X.X., He, B., Zhong, J.B., Liao, G.S., and Liu, J.: The biomass and productivity of *Casuarina equisetifolia* shelterbelt in the coastal sandy land of Guangxi, Protection Forest Science and Technology, (5), 6-8, 2008 (in Chinese).

Xie, W.D., Ye, S.M., Yang, M., and Zhao, L.J.: Biomass and distribution pattern of *Pinus massoniana* plantation in the southeast Guangxi, Journal of Beihua University (Natural Science Edition), 10(1), 68-71, 2009 (in Chinese).

Xie, Z.S., Chen, B.G., Han, J.G., and Deng, Y.S.: The biomass estimative model of two *Eucalypts* in Leizhou Peninsula, in: Studies on the *Eucalypt* Ecosystem of Short Rotation in Leizhou, Zeng, T.X., eds., China Forestry Publishing House, Beijing, China, 66-75, 1995 (in Chinese).

Xu, X.Q., and Chen, Z.D.: A preliminary study of the biomass of cultivated Chinese white poplar stands, Journal of Nanjing Forestry University, (1), 130-136, 1987 (in Chinese).

Xu, Z.L., and Luo, T.Y.: A study on biomass structure of the natural secondary forest of *Pinus massoniana*, Acta Agriculturae Universitatis Jiangxiensis, 15(2), 168-173, 1993 (in Chinese).

Xu, D.P., Yang, M.Q., Zeng, Y.T., and Li, W.X.: Aboveground primary productivity and nutrient cycling of *Acacia mangium* plantation, Guangdong Forestry Science and Technology, (4), 11-16, 1994a (in Chinese).

Xu, D.P., Zeng, Y.T., and Li, W.X.: Aboveground biomass and nutrient cycling of young *Eucalyptus urophylla* plantation, Forest Research, 7(6), 600-605, 1994b (in Chinese).

Xu, D.P., He, Q.X., Yang, Z.J., Long, Y.S., and Jian, X.H.: Aboveground primary productivity and nutrient cycling of *Eucalyptus grandis × E. urophylla* plantation, Forest Research, 10(4), 365-372, 1997 (in Chinese).

Xu, D.P., Bernie, D., Gong, M.Q., Nick, M., and Wang, Z.H.: Effects of phosphorus fertilization and ectomycorrhizal fungal inoculation on the productivity and nutrient accumulation of *Eucalyptus globulus* plantation, Forest Research, 17(1), 26-35, 2004 (in Chinese).

Xu, F.W., Gao, Y.P., He, K.Q., Ding, F.J., and Dai, Q.H.: Biomass and net productivity of *Pinus massoniana* stands at different stand ages, Hubei Agricultural Sciences, 52(8), 1853-1858, 2013 (in Chinese).

Xu, H.Y., Zheng, S.K., and Lu, Y.N.: The biomass of I-72/58 poplar plantation, Scientia Silvae Sinicae, 26(1), 22-29, 1990 (in Chinese).



Xu, J.H., Wang, M.L., Huang, Q.F., and Gong, S.F.: Aboveground biomass model of *Quercus acutissima* natural forest, Anhui Forestry Science and Technology, 37(4), 3-6, 2011a (in Chinese).

Xu, J.W., Li, C.R., Wang, W.D., Qiao, Y.J., Cheng, H.Y., and Wang, Y.H.: The biomass and productivity of *Pinus thunbergii* protective forests in sandy coastal area, Journal of Northeast Forestry University, 33(6), 29-32, 2005 (in Chinese).

Xu, J.W., Ai, S.J., Han, F., Fan, Y.X., Zhao, F., Lu, S.S., and Fu, Y.: The biomass of young *Pinus taeda* plantations in sandy coastal areas, Journal of Shandong Agricultural University (Natural Science Edition), 38(2), 252-256, 2007 (in Chinese).

Xu, W., Hu, H.B., and Zhou, C.H.: Biomass structure and distribution characteristics of *Pinus massoniana* plantation in eastern Anhui Province, Journal of Central South University of Forestry and Technology, 31(6), 111-115, 2011b (in Chinese).

Xu, X.N., and Li, H.K.: Growth characteristics of a mixed plantation of Chinese fir and *Pinus taiwanensis*, Forest Research, 9(3), 278-283, 1996 (in Chinese).

Xu, Z.B., Li, X., Dai, H.C., Tan, Z.X., Zhang, Y.P., Guo, X.F., Peng, Y.S., and Dai, L.M.: Study on the biomass of the broadleaved Korean pine forest in Changbai Mountain, Research of Forest Ecosystem, 5, 33-47, 1985 (in Chinese).

Xu, Z.Q., Li, W.H., Liu, W.Z., and Wu, X.B.: Study on the biomass and productivity of Mongolian oak forests in northeast region of China, Chinese Journal of Eco-Agriculture, 14(3), 21-24, 2006 (in Chinese).

Xu, Z.Z., Liu, G.Y., Wang, G.H., Zhao, G.H., and Ma, C.M.: The biodiversity and biomass in *Larix principis-rupprechtii* plantation community in Yanshan Mountains, Forest Resources Management, (2), 43-49, 2010 (in Chinese).

Xuan, Z.L., Zhang, Q.C., Ge, L.L., He, H.J., Xu, M.M., and Xu, W.S.: Biomass structure and distribution of Korean larch plantations, Forest Resources Management, (1), 53-57, 2013 (in Chinese).

Xue, P.: Growth and biomass of 6-year-old *Eucalyptus urophylla* plantation in Leizhou Forestry Bureau, Eucalypt Science and Technology, 26(1), 18-21, 2009 (in Chinese).

Xue, X.K., and Sheng, W.T.: Study on the biomass of *Fokienia hodginsii* plantation in Zhuting Town, Hunan Province, Forest Science and Technology, (4), 16-19, 1993 (in Chinese).

Xue, L., Jacobs, D.F., Zeng, S.C., Yang, Z.Y., Guo, S.H., and Liu, B.: Relationship between aboveground biomass and stand density index in *Populus × euramericana* stands, Forestry, 85, 611-619, 2012.

Yan, R.F., and Zhu, S.Q.: Comparison of biomass estimates of *Pinus massoniana* plantation, Collection of Guizhou Agricultural College, (4), 101-105, 1984 (in Chinese).

Yan, H.P., Li, H., and Zhang, J.S.: Study on the biomass of *Acer truncatum* forest, Journal of Beijing Forestry



University, 19(S2), 108-112, 1997 (in Chinese).

Yan, J.F., Guan, Q.W., Deng, S.Q., Yu, S.Q., and Shan, X.D.: The biomass and productivity of *Platycladus orientalis* plantation in Yunlong Mountain of Xuzhou, China Forestry Science and Technology, 23(2), 48-50, 2009 (in Chinese).

Yan, W.X., Su, Y.M., Liu, X.L., and Liang, H.C.: The biomass and productivity of *Picea likiangensis* var. *balfouriana* plantation, Journal of Sichuan Forestry Science and Technology, 12(4), 17-22, 1991 (in Chinese).

Yang, L.: The biomass model of *Populus tomentosa* plantation in Daxing District, Beijing, Hebei Journal of Forestry and Orchard Research, 26(4), 345-348, 2011 (in Chinese).

Yang, T.: The biomass and root distribution of *Pinus massoniana* and *Quercus acutissima* natural secondary mixed forest, Journal of Xinyang Agricultural College, 14(4), 4-6, 9, 2004 (in Chinese).

Yang, X.: The biomass and productivity of the intercropping ecosystem with paulownia (*Paulownia elongata*) and crops, Acta Agriculturae Universitatis Henanensis, 20(4), 485-509, 1986 (in Chinese).

Yang, Z.Q.: Biomass distribution and productivity characteristics of forests in Karst desertification area, Guangdong Agricultural Sciences, (7), 16-163, 2013 (in Chinese).

Yang, D., and Yang, X.Q.: Study on the biomass and productivity of *Pinus tabuliformis* plantation in Wufeng Mountain, Wudu County, Gansu Province, Journal of Northwest Normal University (Natural Science Edition), 40(1), 70-75, 2004 (in Chinese).

Yang, H.K., and Cheng, S.Z.: Study on the biomass of a karst forest community in Jizi Mountain, Maolan County, Guizhou Province, Acta Ecologica Sinica, 11(4), 307-312, 1991 (in Chinese).

Yang, Q.P., Li, M.G., Wang, B.S., Li, R.W., and Wang, C.W.: Dynamics of the biomass and net primary productivity in succession of south subtropical forests in Southwest Guangdong, Chinese Journal of Applied Ecology, 14(12), 2136-2140, 2003a (in Chinese).

Yang, Q.P., Wang, B., Guo, Q.R., Zhao, G.D., Fang, K., and Liu, Y.Q.: Effects of *Phyllostachys edulis* expansion on carbon storage of evergreen broadleaved forest in Dagang Mountain, Jiangxi, Acta Agriculturae Universitatis Jiangxiensis, 33(3), 529-536, 2011 (in Chinese).

Yang, R., Deng, Z.J., Qin, M.C., and Dai, P.Y.: Study on the biomass of *Cupressus funebris* plantation in the hilly region, the middle of Sichuan Province, Sichuan Forestry Science and Technology, 8(1), 21-24, 1987 (in Chinese).

Yang, T.H., Song, K., Da, L.J., Li, X.P., and Wu, J.P.: The biomass and aboveground net primary productivity of *Schima superba-Castanopsis carlesii* forests in east China, *Science China Life Science*, 40(7), 610-619, 2010a (in Chinese).

Yang, X., Wu, G., Huang, D.M., and Yang, C.Q.: Quantitative study on biomass accumulation of *Paulownia elongate*,



Chinese Journal of Applied Ecology, 10(2), 143-146, 1999 (in Chinese).

Yang, X.M., Cheng, J.M., and Meng, L.: Carbon storage and density features of *Pinus tabuliformis f. shekannesis* natural forest in the Loess Plateau, Science of Soil and Water Conservation, 8(2), 41-45, 58, 2010b (in Chinese).

Yang, Y., Ran, F., Wang, G.X., Zhu, W.Z., Yang, Y., and Zhou, P.: Biomass model and carbon storage of *Pinus yunnanensis* on Tibet Plateau of China, Chinese Journal of Ecology, 32(7), 1674-1682, 2013 (in Chinese).

Yang, Y.L., Gao, J.B., Cao, F., Lu, D.B., Zhao, Q.X., Wu, Y.X., and Lu, Z.M.: Effect of thinning on the growth of *Larix gmelinii*, Journal of Jilin Forestry Science and Technology, 32(5), 21-24, 2003b (in Chinese).

Yang, Z.W., Tan, F.L., Xiao, X.X., Chen, L.S., and Zhuo, K.F.: Study on the biomass of *Fokienia hodginsii* plantation, Scientia Silvae Sinicae, 36(S1), 120-124, 2000 (in Chinese).

Yao, D.H., and He, Y.J.: Study on the biomass and productivity of *Pinus massoniana* air-seeding forests, Journal of Hunan Environmental-Biological Polytechnic, 7(1), 23-27, 2001 (in Chinese).

Yao, D.H., and Li, Z.H.: Study on the biomass dynamics of *Cryptomeria japonica* plantation, Scientia Silvae Sinicae, 33(S2), 203-207, 1997 (in Chinese).

Yao, D.H., Yang, M.S., and Li, Z.H.: Effect of stand density on the biomass and productivity of *Eucalyptus grandis × E. urophylla* plantation, Journal of Central South Forestry University, 20(3), 20-23, 2000 (in Chinese).

Yao, G.Q., Chi, G.Q., Dong, Z.Q., and Guo, D.W.: The productivity of three artificial Korean pine stands in the mountainous areas of Liaoning Province, Journal of Northeast Forestry University, 14(4), 42-47, 1986 (in Chinese).

Yao, Y.J., Kang, W.X., and Tian, D.L.: Study on the biomass and productivity of *Cinnamomum camphora* plantation, Journal of Central South Forestry University, 23(1), 1-5, 2003 (in Chinese).

Ye, W.H.: Study on Tree Architecture of Three Hardwoods. Heilongjiang Science and Technology Press, Harbin, China, 84-88, 1995 (in Chinese).

Ye, W.P.: Structure dynamics and vegetation productivity in the processes of ecological restoration in red earth of Lijiang Valley, Thesis for Master's Degree, Guangxi Normal University, Guilin, China, 41pp., 2005 (in Chinese).

Ye, J.Z., and Jiang, Z.L.: Study on the biomass and its distribution of *Cunninghamia lanceolata* plantations in the hilly regions of the south Jiangsu Province, Acta Ecologica Sinica, 3(1), 7-14, 1983 (in Chinese).

Ye, J.Z., Jiang, Z.L., Zhou, B.L., Han, F.Q., and Chen, S.B.: Annual dynamics of the biomass of Chinese fir forests in Yangkou Forest Farm, Fujian Province, Journal of Nanjing Institute of Forestry, (4), 1-9, 1984 (in Chinese).

Ye, S.M., Zheng, X.X., Xie, W.D., and Zhao, L.J.: Sprout regeneration and plant regeneration influences on the yield of *Eucalyptus urophylla × E. grandis* plantation, Journal of Nanjing Forestry University (Natural Science Edition), 31(3), 43-46, 2007 (in Chinese).



Ye, S.M., Zheng, X.X., Yang, M., Xie, W.D., Zhao, L.J., and Liang, H.W.: Biomass and productivity of stratified mixed stands of *Eucalyptus urophylla* and *Acacia mangium*, Journal of Beijing Forestry University, 30(3), 37-43, 2008 (in Chinese).

Ye, W.P., Li, X.K., Lyu, S.H., Ou, Z.L., Pan, Z., Su, Z.M., and Xie, X.: Dynamics of vegetation structure and biomass in the processes of ecological restoration in the red soil region of Lijiang Valley, Journal of Ecology and Rural Environment, 22(1), 5-10, 2006 (in Chinese).

Yi, A.Y.: Biomass and productivity of mixed plantation of *Cryptomeria japonica* and *Cunninghamia lanceolata*, Sichuan Forestry Exploration and Design, (3), 50-52, 59, 1998 (in Chinese).

Yi, W.M., Zhang, Z.P., Ding, M.M., and Wang, B.S.: Biomass and efficiency of radiation utilization in *Erythrophleum fordii* community, *Acta Ecologica Sinica*, 20(2), 397-403, 2000 (in Chinese).

You, W.Z., Huo, C.F., Xing, Z.K., Zhao, G., Zhang, H.D., Wei, W.J., and Yan, T.W.: Biomass and net primary productivity of *Larix olgensis* plantation in Bingla Mountains, northeast China, Journal of Shenyang Agricultural University, 42(5), 565-569, 2011 (in Chinese).

Yu, M.J.: Dynamics of an evergreen broadleaved forest dominated by *Cyclobalanopsis glauca* in southeast China, Scientia Silvae Sinicae, 35(6), 42-51, 1999 (in Chinese).

Yu, B., Zhang, Q.L., Wang, L.M., and Wu, J: Characteristics of biomass and productivity in *Larix gmelinii* natural forests with different stand structures, Journal of Zhejiang Agriculture and Forestry University, 28(1), 52-58, 2011 (in Chinese).

Yu, X.T., Chen, C.J., and Lin, S.Z.: Preliminary study on the biomass of Chinese fir plantation in Fujian Province, Fujian College of Forestry, Nanping, China, 46-68, 1979 (in Chinese).

Yu, Y.F., Song, T.Q., Zeng, F.P., Peng, W.X., Wen, Y.G., Huang, C.B., Wu, Q.B., Zeng, S.X., and Yu, Y.: Dynamic changes of biomass and its allocation in *Cunninghamia lanceolata* plantations of different stand ages, Chinese Journal of Ecology, 32(7):1660-1666, 2013 (in Chinese).

Yuan, C.M., Lang, N.J., Meng, G.T., Fang, X.J., Li, G.X., and Wen, S.L.: Community structure characteristics and biological productivity of *Acacia dealbata* protective plantation in Toutang mountainous area, Yunnan Forestry Science and Technology, (2), 24-27, 1998 (in Chinese).

Yuan, C.M., Lang, N.J., Meng, G.T., Fang, X.J., Li, G.X., and Wen, S.L.: The structural feature and biomass of soil-water conservative plantation of Armand pine in the upper reach of the Yangtze River, Journal of Northeast Forestry University, 30(3), 5-7, 2002 (in Chinese).

Zeng, X.P., Peng, S.L., and Zhao, P.: Measurement of respiration amount in artificial *Acacia mangium* froest in a low subtropical hill forest region of Guangdong, Acta Phytoecologica Sinica, 24(4), 420-424, 2000 (in Chinese).



Zeng, X.P., Cai, X.A., Zhao, P., Rao, X.Q., Zou, B., Zhou, L.X., Lin, Y.B., and Fu, S.L.: Biomass and net primary productivity of three plantation communities in hilly land of lower subtropical China, Journal of Beijing Forestry University, 30(6), 148-152, 2008 (in Chinese).

Zhai, B.G., Song, C.H., Zhang, H.D., and Wang, W.X.: Study on the biomass and productivity of *Pinus tabuliformis* plantation at a permanent plot in Taiyue Forest Region, Shanxi Province, Journal of Beijing Forestry University, 14(S1), 156-163, 1992 (in Chinese).

Zhang, B.L.: Study on the biomass and productivity of *Quercus liaotungensis* stands in Ziwuling Forest Region, Shaanxi Province, Journal of Northwestern College of Forestry, 5(1), 1-7, 1990 (in Chinese).

Zhang, H.T.: Preliminary study on the biomass of *Pinus sylvestris* var. *sylvestriformis* forest, Jilin Forestry Science and Technology, (3), 5-7, 1992 (in Chinese).

Zhang, J.D.: Relationship between initial stand density and the biomass of *Pinus taiwanensis* plantations, Forest Science and Technology, (1), 7-10, 2000 (in Chinese).

Zhang, J.F.: Determination of the biomass of Chinese white poplar plantation in Daxing District, Beijing, Contemporary Eco-Agriculture, (3-4), 54-57, 2010a (in Chinese).

Zhang, S.L.: Study on forest biomass and carbon storage estimation of *Cunninghamia lanceolata* in Minjiang Watershed based on RS and GIS, Thesis for Master's Degree, Fujian Agriculture and Forestry University, Fuzhou, China, 84pp., 2008 (in Chinese).

Zhang, W.Z.: The biomass and productivity of *Pinus elliottii* plantation in Pingnan County, Forest Investigation Design, (2), 11-16, 2010b (in Chinese).

Zhang, Z.J.: Study on spatial characteristics of *Pinus massoniana* biomass and root distribution in an acid rain area, Chongqing, Thesis for Master's Degree, Agricultural University of Hebei, Baoding, China, 50pp., 2006 (in Chinese).

Zhang, Z.Q.: The standing crop of *Pinus koraiensis* plantations in the eastern Heilongjiang Province, Journal of Northeastern Forestry Institute, (4): 85-98, 1981 (in Chinese).

Zhang, B.L., and Chen, C.G.: Biomass and productivity of *Robinia pseudoacacia* plantation in Hongxing Forest Farm of Changwu County, Shaanxi Province, Shaanxi Forest Science and Technology, (3), 13-17, 1992 (in Chinese).

Zhang, C.L., and Zhou, X.F.: Biomass of a natural secondary birch stand, in: Long-term Located Research on Forest Ecosystems, Volume 1, Zhou, X.F., Wang, Y.H., Zhao, H.X., eds., Northeast Forestry University Press, Harbin, China, 428-435, 1991 (in Chinese).

Zhang, F., and Shangguan, T.L.: Synecological features and biomass of *Larix principis-rupprechtii* forest in Guandi Mountain, Shanxi Province, Journal of Shanxi University (Natural Science Edition), 15(1), 72-77, 1992 (in



Chinese).

Zhang, J.X., and Yuan, Y.Z.: A study on the biomass and productivity of *Pinus fenzeliana* forest in Hainan, Acta Phytoecologica et Geobotanica Sinica, 12(1), 63-69, 1988 (in Chinese).

Zhang, S.Y., and Pan, C.D.: The compatible biomass model of *Picea schrenkiana* young plantation, Journal of Fujian College of Forestry, 22(3), 201-204, 2002 (in Chinese).

Zhang, G.B., Li, X.Q., She, X.S., Hu, C.Q., and Hu, G.H.: Biomass characteristics of dominant tree species (group) in Lingnan Forest Farm, Anhui Province, Scientia Silvae Sinicae, 48(5), 136-140, 2012 (in Chinese).

Zhang, J.G., Li, Y.Q., Ji, J.S., and Li, R.C.: Effect of fertilization on the biomass of young Chinese fir plantation, Forest Research, 9(S), 41-47, 1996a (in Chinese).

Zhang, J.W., Chen, C.Y., Deng, S.J., and Feng, Z.W.: Comparison of the mathematical patterns for estimating the standing crop of *Cunninghamia lanceolata* plantation, Journal of Northeastern Forestry Institute, 12(4), 1-6, 1984 (in Chinese).

Zhang, J.X., Zhou, W., Li, Y., Luo, W., and Xie, C.M.: The biomass of Chinese corktree plantations, Non-wood Forest Research, 8(1), 35-40, 1990 (in Chinese).

Zhang, J.Z., Zhou, L.F., Huang, S.Q., and Chen, J.: The growth and productivity of *Acacia dealbata* plantation, East China Forest Management, 8(3), 35-38, 1994a (in Chinese).

Zhang, L., Luo, T.X., Deng, K.M., Dai, Q., Huang, Y., Jiang, Z.F., Tao, M.Y., and Zeng, K.Y.: Biomass and net primary productivity of secondary evergreen broadleaved forest in Huangmian Forest Farm, Guangxi, Chinese Journal of Applied Ecology, 15(11), 2029-2033, 2004a (in Chinese).

Zhang, L., Huang, Y., Luo, T.X., Dai, Q., and Deng, K.M.: Age effects on stand biomass allocations to different components: A case study of *Cunninghamia lanceolata* plantations and *Pinus massoniana* plantations, Journal of the Graduate School of the Chinese Academy of Sciences, 22(2), 170-178, 2005 (in Chinese).

Zhang, Q., Fan, S.H., Liu, G.L., Feng, H.X., Zong, Y.C., and Fei, B.H.: A study on biomass and productivity of *Populus × euramericana* cv. 'San Martino' (I-72/58) plantation on beach land of Yangtze River, Forest Research, 21(4), 542-547, 2008a (in Chinese).

Zhang, S.G., Yao, Y.T., Qi, L.W., and Xue, J.J.: The biomass and nutrient element (Cu and Mo) distribution of *Larix principis-rupprechtii* plantation, Forest Science and Technology, (4), 19-21, 1999a (in Chinese).

Zhang, S.G., Liu, J., Huang, K.Y., Liang, R.L., and Lan, X.: Biomass and distribution patterns of *Pinus massoniana* plantation in northwest Guangxi, Guangxi Forestry Science, 39(4), 189-192, 219, 2010 (in Chinese).

Zhang, S.L., Liu, J., and Yu, K.Y.: Study on compatible nonlinear stand biomass model based on SPSS, Journal of Fujian Agriculture and Forestry University (Natural Science Edition), 37(5), 496-500, 2008b (in Chinese).



Zhang, W.H., Wang, Y.P., Kang, Y.X., and Liu, X.J.: Study on the relationship between *Larix chinensis* population's structure and environment factors, Acta Ecologica Sinica, 24(2), 41-47, 2004b (in Chinese).

Zhang, W.Q., Peng, S.L., Ren, H., and Peng, Z.W.: The allocation of biomass and energy in *Acacia mangium* forest, Acta Ecologica Sinica, 15(S1): 44-48, 1995 (in Chinese).

Zhang, W.Y., Zheng, Y.S., and You, X.Z.: A study on the biomass models for organs of *Tsoongiodenron odorum*, Acta Agriculturae Universitatis Jiangxiensis, 21(3), 410-413, 1999b (in Chinese).

Zhang, X.D., Xue, M.H., Xu, J., Wang, T., Chen, Y.F., Zhao, G.P., and Song, C.: Biomass structure of *Pinus massoniana* plantations, Journal of Anhui Agricultural College, 19(4), 268-273, 1992 (in Chinese).

Zhang, X.D., Wu, Z.M., and Peng, Z.H.: Mathematical models of the biomass structure of *Pinus thunbergii* plantation, Journal of Biomathematics, 9(5), 60-65, 1994b (in Chinese).

Zhang, Y.S., Wang, X.L., and Zhou, L.S.: Preliminary study on the biomass of *Pinus schrenkiana* forest, Journal of August 1st Agricultural College, (3), 19-25, 1980 (in Chinese).

Zhang, Z.H., Li, Y., and Xie, R.G.: A preliminary study on the growth and biomass of the mixed plantation of *Magnolia officinalis* and *Cunninghamia lanceolata*, Journal of Fujian Forestry Science and Technology, 23(3), 28-31, 1996b (in Chinese).

Zhang, Z.H., Wang, L.C., Luo, J.X., and Zheng, D.R.: Study on tree biomass models of *Pinus yunnanensis* in northwest Yunnan Province, Shandong Forestry Science and Technology, (4), 4-6, 2011 (in Chinese).

Zhang, Z.J., Wang, Y.H., Yuan, Y.X., Li, Z.Y., Cao, L., Zhang, G.Z., Yu, P.T., and Wang, Y.: Study on the biomass and distribution of *Pinus massoniana* natural secondary forest, Journal of Agricultural University of Hebei, 29(5), 37-43, 2006 (in Chinese).

Zhang, Z.P., Ding, M.M., Zai, G.L., Yi, W.M., and Huang, Y.J.: Biomass and net primary productivity of *Cryptocarya concinna* community in Dinghushan Nature Reserve, Ecological Science, (1), 8-11, 1991 (in Chinese).

Zhao, L.: Study on the biomass and its allocation of *Schima superba* planted forests, Jiangxi Forestry Science and Technology, (4), 5-7, 59, 2006 (in Chinese).

Zhao, K., and Tian, D.L.: Study on the biomass and productivity of mature Chinese fir stand in Huitong County, Journal of Central South Forestry University, 20(1), 7-13, 2000 (in Chinese).

Zhao, P., and He, Y.J.: Stand growth and biomass of *Zanthoxylum ailanthoides*, Journal of Central South University of Forestry and Technology, 29(5), 67-71, 2009 (in Chinese).

Zhao, T.S., and Zhang, P.C.: Comprehensive effects of tending and felling on *Pinus taiwanensis* plantation, Acta Agriculturae Universitatis Henanensis, 23(4), 409-421, 1989 (in Chinese).

Zhao, G.L., Wang, J.X., Wang, X.Z., Shen, Y.B., and Zhou, J.C.: Nutrient element cycling and density effects of *Pinus*



*tabuliformis* plantations, Journal of Beijing Forestry University, 28(4), 39-44, 2006 (in Chinese).

Zhao, J.M., Wu, Z.W., and Xie, S.X.: Individual growth and biomass characteristics of wild *Idesia polycarpa* in Guizhou, Guizhou Forestry Science and Technology, 40(4), 7-13, 23, 2012 (in Chinese).

Zhao, T.S., Guang, Z.Y., Zhao, Y.M., and Liu, G.W.: Study on the biomass and productivity of *Larix kaempferi* plantation, Acta Agriculturae Universitatis Henanensis, 33(4), 350-353, 1999 (in Chinese).

Zheng, D.H.: Comparison of biomass estimation methods of *Pinus taiwanensis* plantation, Journal of Fujian Forestry Science and Technology, 26(S), 36-39, 1999 (in Chinese).

Zheng, H., Ouyang, Z.Y., Xu, W.H., Wang, X.K., Miao, H., Li, X.Q., and Tian, Y.X.: Variation of carbon storage by different reforestation types in the hilly red soil region of southern China, Forest Ecology and Management, 255, 1113-1121, 2008.

Zheng, Z., Feng, Z.L., Cao, M., Liu, H.M., and Liu, L.H.: Biomass and net primary productivity of primary tropical wet seasonal rainforest in Xishuangbanna, *Acta Phytoecologica Sinica*, 24(2), 197-203, 2000 (in Chinese).

Zheng, Z., Liu, H.M., and Feng, Z.L.: Biomass of tropical montane rain forest in Xishuangbanna, Southwest China, Chinese Journal of Ecology, 25(4), 347-353, 2006 (in Chinese).

Zhong, Q.L., Zhang, Z.Y., Zhang, C.H., Zhou, H.L., and Huang, Z.Q.: The dynamic analysis of the biomass and its structure of *Machilus pauhoi*, Acta Agriculturae Universitatis Jiangxiensis, 23(4), 533-536, 2001 (in Chinese).

Zhou, S.Q., and Huang, J.Y.: Biomass estimation models of *Larix mastersiana* plantation, Journal of Sichuan Forestry Science and Technology, 12(2), 67-69, 1991a (in Chinese).

Zhou, S.Q., and Huang, J.Y.: A study on biomass and productivity of *Larix mastersiana* plantation in Sichuan, Acta Phytoecologica et Geobotanica Sinica, 15(1): 9-16, 1991b (in Chinese).

Zhou, Z.X., and Xie, S.X.: The biomass and productivity of *Eucommia ulmoides* plantation, Forest Research, 7(6), 646-651, 1994.

Zhou, G.M., Yao, J.X., Qiao, W.Y., Yang, Q.H., Zhu, G.J., and Xu, W.Y.: Biomass of Chinese fir planted forest in Qingyuan County of Zhejiang Province, Journal of Zhejiang Forestry College, 13(3), 235-242, 1996 (in Chinese).

Zhou, G.Y., Zeng, Q.B., Lin, M.X., Chen, B.F., Li, Y.D., and Wu, Z.M.: Study on the biomass and nutrient allocation in *Manglietia hainanensis* plantation ecosystem at Jianfengling, Hainan Province, Forest Research, 10(5), 453-457, 1997 (in Chinese).

Zhou, W.C., Mu, C.C., Liu, X., and Gu, W.: Carbon sink in natural swamp forest ecosystems in Xiaoxing'an Mountains, Journal of Northeast Forestry University, 40(7), 71-75, 127, 2012 (in Chinese).

Zhou, Z.Z., Zheng, H.S., Yin, G.T., Yang, Z.J., and Chen, K.T.: Biomass equations for rubber tree in southern China, Forest Research, 8(6), 624-629, 1995 (in Chinese).





Zhu, S.Q., and Yang, S.Y.: Preliminary study on the biomass and its distribution of *Cunninghamia lanceolata* plantation, Guizhou Agricultural College, Guiyang, 14pp., 1979 (in Chinese).

Zhu, B.L., Li, Z.H., and Chen, S.X.: Effect of stand density on the biomass and productivity of *Eucalyptus urophylla* × *E. grandis* plantation, Journal of Hunan Environmental-Biological Polytechnic, 13(4), 11-14, 2007 (in Chinese).

Zhu, S.Q., Wei, L.M., Chen, Z.R., and Zhang, C.G.: A preliminary study on biomass components of karst forest in Maolan of Guizhou Province, China, Acta Phytoecologica Sinica, 19(4), 358-367, 1995 (in Chinese).

Zhu, X.W., Xiao, Y., and Cai, W.C.: A preliminary study on the biomass of *Populus davidiana* natural forest, Science and Technology of Qinghai Agriculture and Forestry, (1), 30-34, 1988 (in Chinese).

Zhuang, H.L., Becuwe, X., Xiao, C.B., Wang, Y.H., Wang, H., Yin, B., and Liu, C.J.: Allometric equation-based estimation of biomass carbon sequestration in *Metasequoia glyptostroboides* plantations in Chongming Island, Shanghai, Journal of Shanghai Jiaotong University (Agricultural Science Edition), 29(2), 48-55, 2012 (in Chinese).

Zou, C.J., Bu, J., and Xu, W.D.: Biomass and productivity of *Pinus sylvestriformis* plantation, Chinese Journal of Applied Ecology, 6(2), 123-127, 1995 (in Chinese).