# Peer review of "A review of biomass equations for China's tree species"

_Earth System Science Data, 2019_

## Referee Comment (RC1) · Anonymous Referee #1 · 5 Apr 2019

General comments

The present study describes a dataset with tree allometric equations for China, which is - as the authors state - up-to-date missing. The dataset comprises an extensive amount of allometric equations, gathered from literature between 1978 and 2013. The authors describe in detail, how they dealt with missing data and how they rate the quality of the applicable range. The dataset itself is available in Pangaea as Excel-file with two data sheets. However, I have some concerns on the manuscript and especially the dataset, which I will explain in the following.

MANUSCRIPT

Specific comments

[Figure]

- The authors state, that allometric equations for China are missing (e.g. page 3, line1-2). However, the cited reference (Henry et al. 2015) uses data from 2014. In a quick internet search, I found the http://www.globallometree.org database containing more than 1000 allometric equations for China. Also the name of the new dataset (ChinAllomeTree) is quite near to that of the global database (GlobAllomeTree). Please explain in the manuscript and work out the differences. Will this new collection be introduced into GlobAllomeTree?

- Page 3, line 3-4. Please provide a citation.

- Page 9, line 3-4. "As it is often the case" – I would just skip that part of the sentence, as it is rather discussion than result

- Table 1. (i) Stand density - Unclear to me, if it refers to the whole number of trees/ha or only those given in "Tree species (group)". (ii) Biomass component - How can a biomass component (after all it is said to be a string) be given in units of kilogram? I would say it's a unitless name.

- Table 3. (i) It might be sufficient to put it in the Appendix? (ii) Please make two sentences out of the first one in the caption. I further do not understand the second part of the sentence ("..., and mixed species in Column 'Species name' does two or more tree species that equations are developed for. "). Please rephrase (iii) I would appreciate to have the authority after the each species name.

- Figure 1a. (i) The figure does not work when printed in greyscale (dots and height classes). (ii) In the coloured version, red dots on green background are a potential problem for colour-blind people. (iii) I do not understand how the map and the small 'overview' in the lower right belong together. What is the small rectangle in the lower right within that 'overview'? It might be helpful to provide Longitude and Latitude also for the 'overview' and to describe its function in the figure caption.

Technical corrections
- Write rather "Tree-biomass equations" or "Tree-allometric equations" than "The tree biomass equation" or "The tree allometric equation", as there is not only one equation, but several different ones.

- Replace $\sim$ with –, as – is the usual from-to sign, while $\sim$ is rather used for approximations

- Page 1, line 12 and page 2, line 4. "spatio-temporal scales" or "spatial and temporal scales"

- Page 7, line 13. Shift "(Fig. 2)" to the end of the sentence.

- Page 9, line 5. ranges

- Is Figure 2 really necessary? I think it is sufficient, that the values are presented in the text

- Table 2, l. "For former two forms. . .." – change to "For texts and tables. . ."

- Table 3, header: "Number of biomass equations" instead of "The number. . ."

- Table 3, caption: "column" instead of "Column"

DATASET

Specific comments

Please provide explanations of the dataset (like abbreviations) on Pangaea or within the Excel-file. The dataset should be understandable without having to read the paper, which is – at least at the moment – not linked on the Pangaea page.

In Table 1 of the manuscript, it is described how parameters like e.g. Latitude, Altitude or MAT are retrieved (from original studies or other sources). It would be helpful to add the information on data-origin within the "General" sheet as a new column to give the user the ability to rate its quality.

Sheet "General"

- What is the difference between dominant species and tree species? What are MAT, MAP? I know it is described in the manuscript, but it should be clear from the dataset as well.

- I am not sure about how equations were pooled or separated and stumbled over this example: ID 268, Li et al. 2013a, has a stand age from 16 to 68 years, while in the original publication, values are given separately per age class. ID 286-289, Li et al. 2010a, give different equations for different age classes. Unfortunately, the original publication is given in Chinese and I thus cannot have a look to see, if I understand the different splitting of age classes in the dataset.

- A further question concerning stand age: ID 268, Li et al. 2013a, has a stand age from 16 to 68 years, the publication gives values for stands of 16, 35, 50 and 68 years. Your dataset states "16∼68" as stand age. In other cases, e.g. ID 508, Wang and Shi 1990, stand age is given as "6, 12, 22, 40". What is the difference? Unfortunately, the original publication of Wang and Shi is in Chinese and I thus cannot have a look to see, if I would understand the difference.

Sheet "Equation"

- What do the variables and coefficients (W, D, H, a, b, c, d) stand for? What are Methods and applicable ranges? Again, I know it is described in the manuscript, but it should be clear from the dataset as well.

- Applicable ranges Height and Diameter: Why are they sometimes "/", sometimes "na"? What is the difference?

- It is to some part impractical to search for equations belonging to a specific ID as they are given e.g. as 5911∼5918. It would be helpful, to have the general ID as additional column in the Equation-sheet.

Technical corrections

- Avoid merging cells as these might be unreadable for other programs.

- Would it be possible to provide the dataset as .txt or .csv file in general?

Sheet "General"

- Replace "∼" with "to"

- Provide complete citations (Appendix B) within or together with the dataset

- Please avoid formulas within the cells, as these can easily and unwittingly be changed by clicking in the cell (e.g. clicking in the first cell of the column "Equations included" gives '=Equation!B3&"∼"&Equation!B14'). This should be changed into plain text.

Sheet "Equation"

- Formulas: saving the dataset as .csv or .txt to import it into other programs results in e.g. W=aÂůDb. It might be better to write W=aÂůDˆb, which is accepted as power-function in a number of programming languages

---

## Short Comment (SC1) · 23 Apr 2019

Dear Authors, General speaking, this manuscript is well organized and concise, with very strong logic. The data information are huge and useful, especially for studying China, and can be of use for the scientific community. Although the flaws of this paper still need further polish and perfect, but the value and significance of the article is still very prominent. Thus, I propose that the review article can be moved ahead for further process. Kind regards,

---

## Short Comment (SC2) · 28 Apr 2019

Dear Yang XIAO, Many thanks to your high appreciation! Our article will be completely polished in the final version after peer review. We would like to communicate in tree growth and biomass with you if possible in future. Best wishes from China. Sincerely, Yunjian Luo
* * *

---

## Author Comment (AC1) · 28 Apr 2019

Dear Editor, Thank you very much for your letter dated 5th April, and the reviewer's reports. We are very grateful to the reviewer's thoughtful and constructive comments on our manuscript and dataset, which are very valuable in improving the quality of the manuscript and dataset. According to the reviewer's comments, we have revised our manuscript and dataset, and rechecked the grammar, spelling, punctuation and clarity carefully in the revised version. All changes made to the original version are marked with RED in revised version, and are listed in the response to the reviewer's comments. Here we attached our responses and revised manuscript. Furthermore, I have contacted PANGAEA assignee (Stefanie Schumacher) of our dataset (https://doi.pangaea.de/10.1594/PANGAEA.895244), and then the revised dataset will

be accessible soon. We sincerely hope that our revised manuscript will be appropriate for your journal. Thank you very much for your kind consideration.

Yunjian Luo, Xiaoke Wang

Please also note the supplement to this comment:
https://www.earth-syst-sci-data-discuss.net/essd-2019-1/essd-2019-1-AC1-supplement.pdf

**Supplement:**

**Response to Anonymous Referee #1 on "ChinAllomeTree 1.0: China's normalized tree biomass equation dataset" by Y. Luo et al.**

Thanks a lot for the reviewer's thoughtful and constructive comments on our manuscript and dataset. According to the reviewer's comments, we have revised our manuscript and dataset, and also rechecked the grammar, spelling, punctuation and clarity carefully in the revised version. All changes made to the original manuscript are marked with RED in the revised manuscript, and are listed in the response to the reviewer's comments. Here we attached our responses and revised manuscript. Furthermore, I have contacted PANGAEA assignee (Stefanie Schumacher) of our dataset (https://doi.pangaea.de/10.1594/PANGAEA.895244), and then the revised dataset will be accessible soon.

**1. MANUSCRIPT**

**1.1 The authors state, that allometric equations for China are missing (e.g. page 3, line1-2). However, the cited reference (Henry et al. 2015) uses data from 2014. In a quick internet search, I found the http://www.globallometree.org database containing more than 1000 allometric equations for China. Also the name of the new dataset (ChinAllomeTree) is quite near to that of the global database (GlobAllomeTree). Please explain in the manuscript and work out the differences. Will this new collection be introduced into GlobAllomeTree?**

>> The state that allometric equation for China are missing has been deleted, and the GlobAllomeTree (http://www.globallometree.org) database and China's equations within the database have been briefed in Introduction (P2, L18-19; P3, L9-11). In the database GlobAllomeTree, only 1173 tree biomass equations were listed from China, and very limited in scope such as data sources (24 scientific articles), spatial coverage (39 sites) and tree species (*ca*. 50 species) (accessed on April 10, 2019). Importance of our article has been highlighted in the revised version (P3, L11-14). Our dataset showed in this manuscript is more complete, containing 5924 equations of near 200 tree species that were derived from 518 references (journals, reports and books). Our article will be reviewed by rigorous peer reviewer and published in ESSD journal, it will benefit more readers and stakeholders worldwide.

>> In order to distinguish our dataset from GlobAllomeTree, the abbreviation (ChinAllomeTree) of our dataset has been deleted in this version, and thus the title of our article has been revised as "A review of biomass equations for China's tree species".

**1.2. Page 3, line 3-4. Please provide a citation.**

>> A citation (Zhang, 2007) has been added in this version of our manuscript (P3, L2-3).

Zhang, X.S.: Vegetation Map of China and Its Geographic Patterns, Geological Publishing House, Beijing, China, 91-124pp., 2007 (In Chinese).

**1.3. Page 9, line 3-4. "As it is often the case" – I would just skip that part of the sentence, as it is rather discussion than result.**

>> Revised as suggested. "As it is often the case" has been deleted (P9, L12-13).

**1.4. Table 1. (i) Stand density - Unclear to me, if it refers to the whole number of trees/ha or only those given in "Tree species (group)". (ii) Biomass component - How can a biomass component (after all it is said to be a string) be given in units of kilogram? I would say it's a unitless name.**

>> In revised Table 1, (i) "Stand density" has been revised as "tree spacing" with unit of trees/ha; "biomass component" has been revised as "tree component", and its unit has also revised as unitless.

**1.5. Table 3. (i) It might be sufficient to put it in the Appendix? (ii) Please make two sentences out of the first one in the caption. I further do not understand the second part of the sentence (". . ., and mixed species in Column 'Species name' does two or more tree species that equations are developed for. "). Please rephrase (iii) I would appreciate to have the authority after each species name.**

>> (i) Table 3 has been put as Appendix D in the revised version. (ii) Column "species name" has been changed as "tree species (group)". Mixed species in column "Tree species (group)" refer to tree species groups (e.g., deciduous broadleaved trees, a certain diameter-class mixed species, even generalized) that biomass equations are developed for. (iii) The authority for each tree species are added.

**1.6. Figure 1a. (i) The figure does not work when printed in greyscale (dots and height classes). (ii) In the colored version, red dots on green background are a potential problem for color-blind people. (iii) I do not understand how the map and the small 'overview' in the lower right belong together. What is the small rectangle in the lower right within that 'overview'? It might be helpful to provide Longitude and Latitude also for the 'overview' and to describe its function in the figure caption.**

>> From the reviewer's comment, we think that the reviewer referred to Figure 3a rather than Figure 1a in the early version. (i) and (ii): To highlight the spatial distribution of study sites, the base map of elevation has been removed, and study sites have been marked with black dots in this version (Figure 2 in the revised version). (iii) The map is redesigned and geographical coordinates (longitude and latitude) have been added in the revised version.

**1.7 Write rather "Tree-biomass equations" or "Tree-allometric equations" than "The tree biomass equation" or "The tree allometric equation", as there is not only one equation, but several different ones.**

>> Revised as suggested (P1, L10; P2, L7 and L10).

**1.8 Replace ~ with –, as – is the usual from-to sign, while ~ is rather used for approximations**

>> Revised as suggested.

**1.9 Page 1, line 12 and page 2, line 4. "spatio-temporal scales" or "spatial and temporal scales"**

>> Revised as suggested. "spatial-temporal scales" has been revised to "spatial and temporal scales" in this version of manuscript (P1, L11; P2, L4).

**1.10 Page 7, line 13. Shift "(Fig. 2)" to the end of the sentence.**

>> Revised as suggested. Considering the Question 1.12, Figure 2 has been put as Appendix C in the revised version. Thus, "(Appendix C)" has been shifted to the end of the sentence (P7, L22). The citation of Appendix C has shown at the end of the sentence.

**1.11 Page 9, line 5. ranges**

>> Revised as suggested (P9, L14).

**1.12 Is Figure 2 really necessary? I think it is sufficient, that the values are presented in the text**

>> Figure 2 showed temporal change of compiled studies during the period 1978-2013. We think that this figure can support the text. However, Figure 2 has been put as Appendix C. The rest figures and appendices have been rearranged.

**1.13 Table 2, I. "For former two forms. . .." – change to "For texts and tables. . ."**

>> Revised as suggested (Table 2).

**1.14 Table 3, header: "Number of biomass equations" instead of "The number. . ."**

>> Revised as suggested (Appendix D in the revised version).

**1.15 Table 3, caption: "column" instead of "Column"**

>> Revised as suggested (Appendix D in the revised version).

**2. DATASET**

**Please provide explanations of the dataset (like abbreviations) on Pangaea or within the Excel-file. The dataset should be understandable without having to read the paper, which is – at least at the moment – not linked on the Pangaea page.**

>> Good suggestion. The sheet "Description" has been added within the Excel file, including the explanations of all variables used in the dataset.

**2.1 Sheet "General"**

**2.1.1 In Table 1 of the manuscript, it is described how parameters like e.g. Latitude, Altitude or MAT are retrieved (from original studies or other sources). It would be helpful to add the information on data-origin within the "General" sheet as a new column to give the user the ability to rate its quality.**

>> Column "Data origin" has been added within both Table 1 and sheet "Description", which describes data origins (e.g., original studies, Google earth, author defined or estimated).

**2.1.2 What is the difference between dominant species and tree species? What are MAT, MAP? I know it is described in the manuscript, but it should be clear from the dataset as well.**

>> Sheet "Description" has been added within the Excel file, including the explanations of all variables used in the dataset. Dominant species indicates that the tree species play the most important roles in the investigated forest (or community, stand). Tree species is the tree species whose biomass equations were shown in original studies.

**2.1.3 I am not sure about how equations were pooled or separated and stumbled over this example: ID 268, Li et al. 2013a, has a stand age from 16 to 68 years, while in the original publication, values are given separately per age class. ID 286-289, Li et al. 2010a, give different equations for different age classes. Unfortunately, the original publication is given in Chinese and I thus cannot have a look to see, if I understand the different splitting of age classes in the dataset.**

>> The equations were not pooled or separated and shown the same as the original reports. In Li et al (2013a), the values of biomass for age class were given. These values were estimated based on the allometric equations that was reported in another reference (Li et al. 2014, in English). In the reference (Li et al., 2014), the allometric equations were reported for the whole age range from 16 to 68 yrs but not for each age class. So in record ID 268, the equations were shown with the stand age from 16 to 68 years. In Li et al. (2010a), different equations for different age class were shown because in the original reference equations for each age class were reported. In revised version, we add a standardized categorization of age classes used by Chinese scientists (Appendix A) for readers to best select equations from our dataset.

Li, H., Li, C.Y., Zha, T.S., Liu, J.L., Jia, X., Wang, X.P., Chen, W.J., and He, G.M.: Patterns of biomass allocation in an age-sequence of secondary *Pinus bungeana* forests in China, The Forestry Chronicle, 90(2), 169-176, 2014. https://doi.org/10.5558/tfc2014-034

**2.1.4 A further question concerning stand age: ID 268, Li et al. 2013a, has a stand age from 16 to 68 years, the publication gives values for stands of 16, 35, 50 and 68 years. Your dataset states "16∼68" as stand age. In other cases, e.g. ID 508, Wang and Shi 1990, stand age is given as "6, 12, 22, 40". What is the difference? Unfortunately, the original publication of Wang and Shi is in Chinese and I thus cannot have a look to see, if I would understand the difference.**

>> For ID 268, see the explanation for Question 2.1.3. For ID 508, stand age was given as several separate ages but not an age range because the original report was give such ones. We always consistently follow the original report.

**2.1.5 Replace "∼" with "to"**

>> Revised as suggested.

**2.1.6 Provide complete citations (Appendix B) within or together with the dataset**

>> Good suggestion. Sheet "Source" has added within the dataset.

**2.1.7 Please avoid formulas within the cells, as these can easily and unwittingly be changed by clicking in the cell (e.g. clicking in the first cell of the column "Equations included" gives'=Equation!B3&"∼"&Equation!B14'). This should be changed into plain text.**

>> Column "Equations included" has been deleted in the revised version, largely due to the weak practicability of equation number in sheet "Equation". Please refer to section 2.2.3.

**2.2 Sheet "Equation"**

**2.2.1 What do the variables and coefficients (W, D, H, a, b, c, d) stand for? What are Methods and applicable ranges? Again, I know it is described in the manuscript, but it should be clear from the dataset as well.**

>> Sheet "Description" has been added within the Excel file, including the explanations of all variables used in the dataset.

**2.2.2 Applicable ranges Height and Diameter: Why are they sometimes "/", sometimes "na"? What is the difference?**

>> In our early version, differences between "/" and "na" were not explained clearly. They have been normalized in the revised version of manuscript and dataset, and also explained in the newly added sheet "Description". "NA" refers to not available; "/" does not necessary or applicable.

**2.2.3 It is to some part impractical to search for equations belonging to a specific ID as they are given e.g. as 5911∼5918. It would be helpful, to have the general ID as additional column in the Equation-sheet.**

>> An addition column "equation number" was added that would follow original ID. The combination of ID

and equation number will label each equation. The same ID as sheet "General" has been added in sheet "Equation", meanwhile, column "Tree species (group)" in sheet "General" has been moved into sheet "Equation".

**2.2.4 Formulas: saving the dataset as .csv or .txt to import it into other programs results in e.g. $W=a \cdot D^b$. It might be better to write $W=a \cdot D^{\wedge}b$, which is accepted as power- function in a number of programming languages.**

>> Change as you suggested.

**2.2.5 Avoid merging cells as these might be unreadable for other programs.**

>> Revised as suggested. Merged cells have been canceled in this version of dataset.

**2.2.6 Would it be possible to provide the dataset as .txt or .csv file in general?**

>> In order to maintain the integrity of the dataset, the dataset is still given in commonly used Excel file. Because special formats (e.g. merged cells, and superscript) have been removed or canceled in the revised version, this Excel file can be easy to save as .txt or .csv file, depending on the users' purposes.

[revised manuscript text omitted]

---

## Referee Comment (RC2) · Anonymous Referee #1 · 28 May 2019

Thank you very much for the updates, I scanned the new dataset and manuscript and it looks much better now. I will be happy to do a further review, once a final revised version based on RC1 and RC2 is available.

---

## Short Comment (SC3) · 29 Jun 2019

In this manuscript, the authors have collected and sorted out a large number of published literatures on biomass equation. It has very important reference value for multi-scale forest biomass estimation. The dataset can be used for forest biomass or carbon storage assessment at the forest stand level, project level, and regional or national scale. In view of the scientific, accurate and transparent requirements of the forest biomass or forest biomass carbon stock assessment, China is currently carrying out voluntary greenhouse gas emission reduction by afforestation and forest management projects, national and provincial land use, land use change and forestry greenhouse gas inventory compilation, all of which urgently need such a biomass equation database.

---

## Editor Comment (EC1) · Birgit Heim (Editor) · 11 Oct 2019

Dear Authors and Colleagues

Thank you all for your contributions. thanks for the authors of 'ChinAllomeTree 1.0: China's normalized tree biomass equation dataset' for the replies to the review of your paper and for the revision of the manuscript and the published data set on PANGAEA. Your suggestions for the revision of the manuscript and the revised published data set are convincing. Your manuscript is well structured and well written and the detailed data set is published with all meta data available for users. We look forward to your final manuscript, Best wishes, Birgit Heim

2019.

---

## Short Comment (SC4) · 12 Oct 2019

Dear Editor Birgit Heim, Thank you very much for your positive evaluation of our manuscript. Next, we will completely recheck our manuscript, and polish it with the aid of a professional English editing company. The final version will be submitted as soon as possible. We sincerely hope that our final version will be appropriate for your journal. If any questions about our manuscript, please contact me. Best wishes from China. Yours sincerely, Yunjian Luo

---

## Author Response (AR1)

**Dear Topical Editor Birgit Heim,**

Sincere thanks to you and referees for positive, constructive comments on our manuscript (ms no. essd-2019-1), which are very valuable in improving its quality.

According to your and the referees' comments, we have made substantial modification on our original manuscript and related dataset. Moreover, we have carefully checked the grammar, phrasing, punctuation and clarity in the revised version. The revised version has been polished with the aid of a professional English editing company *Springer Nature Author Service* (The editing certificate is attached).

Your and the Referees' comments are reported in *Italic*, and point-by-point responses are listed in the following Responses to Topical Editor and Referees. A marked-up manuscript version with track changes is provided so that you clearly identify what changes have been made. Furthermore, our revised dataset in PANGAEA Data Publisher (https://doi.pangaea.de/10.1594/PANGAEA.895244) will be soon accessible to the public after inspected by PANGAEA assignee (Stefanie Schumacher).

We sincerely hope that our revised version will be appropriate for your journal. Thank you very much for your kind consideration.

Kind regards,
Yunjian Luo on behalf of the Authors

**Responses to Topical Editor and Referees**

**Topical Editor**

*Dear Authors and Colleagues, Thank you all for your contributions. thanks for the authors of 'ChinAllomeTree 1.0: China's normalized tree biomass equation dataset' for the replies to the review of your paper and for the revision of the manuscript and the published data set on PANGAEA. Your suggestions for the revision of the manuscript and the revised published data set are convincing. Your manuscript is well structured and well written and the detailed data set is published with all meta data available for users. We look forward to your final manuscript, Best wishes, Birgit Heim*

> >> Thank you very much for your positive evaluation of our manuscript. Point-by-point responses are listed in the following Responses to Referees. In order to further improve the flow and readability of our manuscript, we have polished it with the aid of a professional English editing company *Springer Nature Author Service* (The editing certificate is attached).

**Referee Comment: Anonymous Referee #1**

*The present study describes a dataset with tree allometric equations for China, which is - as the authors state - up-to-date missing. The dataset comprises an extensive amount of allometric equations, gathered from literature between 1978 and 2013. The authors describe in detail, how they dealt with missing data and how they rate the quality of the applicable range. The dataset itself is available in Pangaea as Excel-file with two data sheets. However, I have some concerns on the manuscript and especially the dataset, which I will explain in the following.*

**1. MANUSCRIPT**

**1.1** *The authors state, that allometric equations for China are missing (e.g. page 3, line1-2). However, the cited reference (Henry et al. 2015) uses data from 2014. In a quick internet search, I found the http://www.globallometree.org database containing more than 1000 allometric equations for China. Also the name of the new dataset (ChinAllomeTree) is quite near to that of the global database (GlobAllomeTree). Please explain in the manuscript and work out the differences. Will this new collection be introduced into GlobAllomeTree?*

> >> The state that allometric equation for China are missing has been deleted, and the GlobAllomeTree (http://www.globallometree.org) database and China's equations within the database have been briefed in Introduction (P2, L17-19; P3, L8-10). In the database GlobAllomeTree, only 1145 tree biomass equations were listed from China, and very limited in scope, such as data sources (23 scientific articles), spatial coverage (39 sites)

and tree species (*ca.* 50 species) (accessed on November 1, 2019). Importance of our manuscript has been highlighted in the revised version (P3, L11-14). Our dataset showed in our manuscript is more complete, containing 5924 equations of nearly 200 tree species that were derived from 518 references (journals, reports and books). If our manuscript proceeds through rigorous peer review and finally published in ESSD journal, it will benefit more readers and stakeholders worldwide.

>> Because no direct link is between our dataset and GlobAllomeTree, the abbreviation (ChinAllomeTree) of our dataset has been deleted in this version in order to distinguish our dataset from GlobAllomeTree. Thus, the title of our manuscript has been revised as "A review of biomass equations for China's tree species".

**1.2** *Page 3, line 3-4. Please provide a citation.*

>> A citation (Zhang, 2007) has been added in this revised version (P3, L2; P15, L15-16).

Zhang, X.S.: Vegetation Map of China and Its Geographic Patterns, Geological Publishing House, Beijing, China, 91-124pp., 2007 (In Chinese).

**1.3** *Page 9, line 3-4. "As it is often the case" – I would just skip that part of the sentence, as it is rather discussion than result.*

>> Revised as suggested. "As it is often the case" has been deleted (P9, L10).

**1.4** *Table 1. (i) Stand density - Unclear to me, if it refers to the whole number of trees/ha or only those given in "Tree species (group)". (ii) Biomass component - How can a biomass component (after all it is said to be a string) be given in units of kilogram? I would say it's a unitless name.*

>> In revised Table 1, (i) "Stand density" has been revised as "tree spacing" with unit of trees/ha; (ii) "biomass component" has been revised as "tree component", whose unit has been revised as unitless.

**1.5** *Table 3. (i) It might be sufficient to put it in the Appendix? (ii) Please make two sentences out of the first one in the caption. I further do not understand the second part of the sentence (". . ., and mixed species in Column 'Species name' does two or more tree species that equations are developed for. "). Please rephrase (iii) I would appreciate to have the authority after each species name.*

>> (i) Table 3 has been put as Appendix D in the revised version. (ii) Column "species name" has been changed as "tree species". Mixed forest in column "Tree species" refer to tree species pooled (e.g., deciduous broadleaved trees, a certain diameter-class mixed species, even generalized) that equations are developed for. (iii) The authority for each tree species has been added in the revised version (Appendix D).

**1.6** *Figure 1a. (i) The figure does not work when printed in greyscale (dots and height classes). (ii) In the colored version, red dots on green background are a potential problem for color-blind people. (iii) I do not understand how the map and the small 'overview' in the lower right belong together. What is the small rectangle in the lower right within that 'overview'? It might be helpful to provide Longitude and Latitude also for the 'overview' and to describe its function in the figure caption.*

>> From the referee's comment, we think that the referee referred to Figure 3a rather than Figure 1a in the original version. (i) and (ii): To highlight the spatial distribution of study sites, the base map of altitude has been removed, and study sites have been marked with black dots in this version (Figure 2 in the revised version). (iii) The map has been redesigned and geographical coordinates (longitude and latitude) have been added in the small 'overview' in the revised version. The small 'overview' denotes the part of south China.

**1.7** *Write rather "Tree-biomass equations" or "Tree-allometric equations" than "The tree biomass equation" or "The tree allometric equation", as there is not only one equation, but several different ones.*

>> Revised as suggested (P1, L10; P2, L7 and L10).

**1.8** *Replace ∼ with –, as – is the usual from-to sign, while ∼ is rather used for approximations.*

>> Revised as suggested.

**1.9** *Page 1, line 12 and page 2, line 4. "spatio-temporal scales" or "spatial and temporal scales".*

>> Revised as suggested. "spatial-temporal scales" has been revised to "spatial and temporal scales" in this version of manuscript (P1, L11; P2, L4).

**1.10** *Page 7, line 13. Shift "(Fig. 2)" to the end of the sentence.*

>> Revised as suggested. Considering the below comment 1.12, Figure 2 has been put as Appendix C in the revised version. Thus, the citation of Appendix C has shown at the end of the sentence (P7, L21).

**1.11** *Page 9, line 5. Ranges.*

>> Revised as suggested (P9, L13).

**1.12** *Is Figure 2 really necessary? I think it is sufficient, that the values are presented in the text.*

>> Figure 2 showed temporal change of compiled studies during the period 1978-2013. We think that this figure can support the text. However, Figure 2 has been put as Appendix C. The rest figures and appendices have been rearranged in the revised version.

**1.13** *Table 2, I. "For former two forms. . .." – change to "For texts and tables. . ."*

>> Revised as suggested (P69, Table 2).

**1.14** *Table 3, header: "Number of biomass equations" instead of "The number. . ."*

>> Revised as suggested (P78, Appendix D in the revised version).

**1.15** *Table 3, caption: "column" instead of "Column".*

>> Revised as suggested (P78, Appendix D in the revised version).

**2. DATASET**

*Please provide explanations of the dataset (like abbreviations) on Pangaea or within the Excel-file. The dataset should be understandable without having to read the paper, which is – at least at the moment – not linked on the Pangaea page.*

>> Good suggestion. The Description sheet has been added within the revised dataset, including the descriptions of all variables used in the dataset.

*In Table 1 of the manuscript, it is described how parameters like e.g. Latitude, Altitude or MAT are retrieved (from original studies or other sources). It would be helpful to add the information on data-origin within the "General" sheet as a new column to give the user the ability to rate its quality.*

>> Column "Data origin" has been added within both Table 1 and newly added the Description sheet, which describes data origins (e.g., original studies, Google Earth, author defined, or author estimated).

**2.1 Sheet "General"**

**2.1.1** *What is the difference between dominant species and tree species? What are MAT, MAP? I know it is described in the manuscript, but it should be clear from the dataset as well.*

>> The Description sheet has been added within the revised dataset, including the descriptions of all variables used in the dataset. Dominant species indicates that the tree species play the most important roles in the investigated forest (or community, stand); tree species is the specified tree species whose biomass equations were developed in original studies.

**2.1.2** *I am not sure about how equations were pooled or separated and stumbled over this example: ID 268, Li et al. 2013a, has a stand age from 16 to 68 years, while in the original publication, values are given separately per age class. ID 286-289, Li et al. 2010a, give different equations for different age classes. Unfortunately, the original publication is given in Chinese and I thus cannot have a look to see, if I understand the different splitting of age classes in the dataset.*

>> In the revised dataset, the equations have not been pooled or separated but shown the same as the original studies. We have rechecked stand ages in the dataset in order to keep consistent with ones reported in the original studies. Discrete ages, age ranges (i.e., continuous ages) or age classes can be entered into our dataset as determined by the original studies. Discrete ages or age ranges are entered when equations were specific to ages or age ranges in the original studies; otherwise, age classes (young, middle-aged, premature, mature and

overmature) are given according to stand descriptions. We have added a standard categorization of age classes (Appendix A) used by Chinese scientists for readers to best select equations from our dataset.

Li et al (2013a) reported the values of biomass at several ages by using the age-free biomass equations from another simultaneous study (Li et al., 2014; which was online in 2013, but published in 2014). Thus, to clarify the stand ages and associated equations, Li et al. (2013a) has been replaced by Li et al. (2014) in this revised version. In the study (Li et al., 2014), the age-free biomass equations were developed based on the pooled sample trees at discrete ages (16, 35, 50 and 68 years) but not for each age. So in record ID 268, discrete ages were entered in the revised version.

However, Li et al. (2010a) is another team's study. In the study, biomass equations for each age were built, and thus specific ages were entered into our dataset.

Li, H., Li, C.Y., Zha, T.S., Liu, J.L., Jia, X., Wang, X.P., Chen, W.J., and He, G.M.: Patterns of biomass allocation in an age-sequence of secondary *Pinus bungeana* forests in China, The Forestry Chronicle, 90(2), 169-176, 2014. https://doi.org/10.5558/tfc2014-034.

**2.1.3** *A further question concerning stand age: ID 268, Li et al. 2013a, has a stand age from 16 to 68 years, the publication gives values for stands of 16, 35, 50 and 68 years. Your dataset states "16~68" as stand age. In other cases, e.g. ID 508, Wang and Shi 1990, stand age is given as "6, 12, 22, 40". What is the difference? Unfortunately, the original publication of Wang and Shi is in Chinese and I thus cannot have a look to see, if I would understand the difference.*

>> For ID 268, the range '16-68' has been revised as the discrete ages (16, 35, 50 and 68 years) in the revised version. The detailed, please see the explanation for the above comment 2.1.2. Similarly, For ID 508, the age-free biomass equations were developed based on the pooled sample trees at discrete ages (6, 12, 22, and 40 years) but not for an age range (i.e., continuous ages), and thus discrete ages were entered in our dataset. We always consistently follow the original report.

**2.1.4** *Replace "~" with "to"*

>> Revised as suggested.

**2.1.5** *Provide complete citations (Appendix B) within or together with the dataset*

>> Good suggestion. The Sources sheet has been added within the revised dataset.

**2.1.6** *Please avoid formulas within the cells, as these can easily and unwittingly be changed by clicking in the cell (e.g. clicking in the first cell of the column "Equations included" gives'=Equation!B3&"~"&Equation!B14'). This should be changed into plain text.*

>> Column "Equations included" has been deleted in the revised version due to the poor practicability of equation number in Equation sheet. Please refer to the below response to comment 2.2.3.

**2.2 Sheet "Equation"**

**2.2.1** *What do the variables and coefficients (W, D, H, a, b, c, d) stand for? What are Methods and applicable ranges? Again, I know it is described in the manuscript, but it should be clear from the dataset as well.*

>> The Description sheet has been added within the revised dataset, including the descriptions of all variables used in the dataset.

**2.2.2** *Applicable ranges Height and Diameter: Why are they sometimes "/", sometimes "na"? What is the difference?*

>> In our early version, differences between "/" and "na" were not clearly explained indeed. They have been normalized in the revised version of manuscript and dataset, and also explained in the newly added sheet "Description". "NA" refers to not available; "/" does not necessary or applicable.

**2.2.3** *It is to some part impractical to search for equations belonging to a specific ID as they are given e.g. as 5911~5918. It would be helpful, to have the general ID as additional column in the Equation-sheet.*

>> In the revised version , a new ID system has been designed. Each study has the unique ID that corresponds with one or more tree species and their specific equations. The previous continuous coding scheme of equations (e.g. 5911-5918) has been removed in the revised dataset. Identification number of each equation within a study has been given as the equation number. Three new columns 'ID', 'Equation number' and 'Tree species' have been added in the Equation sheet. The combination of ID and equation number can label each biomass equation.

**2.2.4** *Avoid merging cells as these might be unreadable for other programs.*

>> Revised as suggested. Merged cells have been canceled in the revised dataset.

**2.2.5** *Would it be possible to provide the dataset as .txt or .csv file in general?*

>> In order to maintain the integrity of the dataset, the dataset is still given in commonly used Excel file. Because special formats (e.g., merged cells and superscript) have been removed or canceled in the revised version, the Excel file can be easy to save as .txt or .csv file, depending on the users' purposes.

**2.2.6** *Formulas: saving the dataset as .csv or .txt to import it into other programs results in e.g. $W=a\,D^b$. It might be better to write $W=a \cdot D\hat{}b$, which is accepted as power- function in a number of programming languages.*

>> Revised as suggested.

**Short Comment: Xiao Yang**

*Dear Authors, General speaking, this manuscript is well organized and concise, with very strong logic. The data information are huge and useful, especially for studying China, and can be of use for the scientific community. Although the flaws of this paper still need further polish and perfect, but the value and significance of the article is still very prominent. Thus, I propose that the review article can be moved ahead for further process. Kind regards, Xiao Yang*

>> Many thanks to your high appreciation! The revised version of our manuscript has been carefully proofread to further improve the flow and readability of the text. The revised version is polished with the aid of a professional English editing company *Springer Nature Author Service* (The editing certificate is attached).

**Short Comment: Jianhua Zhu**

*In this manuscript, the authors have collected and sorted out a large number of published literatures on biomass equation. It has very important reference value for multiscale forest biomass estimation. The dataset can be used for forest biomass or carbon storage assessment at the forest stand level, project level, and regional or national scale. In view of the scientific, accurate and transparent requirements of the forest biomass or forest biomass carbon stock assessment, China is currently carrying out voluntary greenhouse gas emission reduction by afforestation and forest management projects, national and provincial land use, land use change and forestry greenhouse gas inventory compilation, all of which urgently need such a biomass equation database.*

>> We are very pleased with your high recognition and evaluation of our manuscript.

**SPRINGER NATURE**
**Author Services**

**Editing Certificate**

This document certifies that the manuscript

**A review of biomass equations for China's tree species**

prepared by the authors

**Yunjian Luo, Xiaoke Wang, Zhiyun Ouyang, Fei Lu, Liguo Feng, Jun Tao**

was edited for proper English language, grammar, punctuation, spelling, and overall style
by one or more of the highly qualified native English speaking editors at SNAS.

This certificate was issued on **October 21, 2019** and may be verfed
on the SNAS website using the verification code **BA14-E404-4946-5996-34DP**.

Neither the research content nor the authors' intentions were altered in any way during the editing process. Documents receiving this certification should be English-ready for publication; however, the author has the ability to accept or reject our suggestions and changes. To verify the final SNAS edited version, please visit our verification page at secure.authorservices.springernature.com/certificate/verify. If you have any questions or concerns about this edited document, please contact SNAS at support@as.springernature.com.

SNAS provides a range of editing, translation, and manuscript services for researchers and publishers around the world. For more information about our company, services, and partner discounts, please visit authorservices.springernature.com.

**A review of biomass equations for China's tree species**

[revised manuscript text omitted]